# Disease-associated polyalanine expansion mutations impair UBA6-dependent ubiquitination

Fatima Amer-Sarsour [1,11], Daniel Falik [2,3,4,11], Yevgeny Berdichevsky[1], Alina Kordonsky[5], Sharbel Eid[6], Tatiana Rabinski [2,3], Hasan Ishtayeh[1], Stav Cohen-Adiv[1], Itzhak Braverman[7,8], Sergiu C Blumen[8,9], Tal Laviv[6,10], Gali Prag [5,10], Gad D Vatine [2,3,4,12✉] & Avraham Ashkenazi [1,10,12✉]

## Abstract

**Expansion mutations in polyalanine stretches are associated with a growing number of diseases sharing a high degree of genotypic and phenotypic commonality. These similarities prompted us to query the normal function of physiological polyalanine stretches and to investigate whether a common molecular mechanism is involved in these diseases. Here, we show that UBA6, an E1 ubiquitin-activating enzyme, recognizes a polyalanine stretch within its cognate E2 ubiquitin-conjugating enzyme USE1. Aberrations in this polyalanine stretch reduce ubiquitin transfer to USE1 and, subsequently, polyubiquitination and degradation of its target, the ubiquitin ligase E6AP. Furthermore, we identify competition for the UBA6-USE1 interaction by various proteins with polyalanine expansion mutations in the disease state. The deleterious interactions of expanded polyalanine tract proteins with UBA6 in mouse primary neurons alter the levels and ubiquitination-dependent degradation of E6AP, which in turn affects the levels of the synaptic protein Arc. These effects are also observed in induced pluripotent stem cell-derived autonomic neurons from patients with polyalanine expansion mutations, where UBA6 overexpression increases neuronal resilience to cell death. Our results suggest a shared mechanism for such mutations that may contribute to the congenital malformations seen in polyalanine tract diseases.**

**Keywords** Autonomic Nervous System; Congenital Central Hypoventilation Syndrome; Trinucleotide Repeats; Ubiquitin-Activating Enzyme; Ubiquitin Transfer System
**Subject Categories** Molecular Biology of Disease; Neuroscience; Post-translational Modifications & Proteolysis

## Introduction

Trinucleotide repeats present within coding regions of the genome encode stretches of the same amino acid (Usdin, 2008), and expansion mutations of such amino acid sequences have now been linked to a growing number of human diseases (Brown and Brown, 2004). The nine diseases caused by expansions of polyalanine stretches are due to the expression of aberrant nuclear proteins (eight of the nine are transcription factors). Examples include mutant paired like homeobox 2B (mutant PHOX2B) in congenital central hypoventilation syndrome (CCHS) (Amiel et al, 2003), mutant RUNX family transcription factor 2 (mutant RUNX2) in cleidocranial dysplasia (Mundlos et al, 1997), mutant homeobox D13 (mutant HOXD13) in synpolydactyly (Muragaki et al, 1996), and mutant poly(A) binding protein nuclear 1 (mutant PABPN1) in oculopharyngeal muscular dystrophy (OPMD) (Richard et al, 2017). The vast majority of cases are caused by de novo mutations and often involve congenital neurological abnormalities (Brown and Brown, 2004; Brown et al, 2001; Laumonnier et al, 2002). The disease-causing expansions vary in length, according to the gene in question, with the severity of the associated clinical phenotype generally increasing with the length of the expanded tract (Albrecht and Mundlos, 2005; Amiel et al, 2004; Usdin, 2008). Physiological polyalanine stretches are commonly found in proteins, although they usually do not exceed 20 residues. In contrast, disease-causing expansion mutations may contain up to 13 additional alanine residues and can result in protein misfolding (Albrecht and Mundlos, 2005; Amiel et al, 2004; Usdin, 2008). The exact functional or structural role of the polyalanine stretches in normal proteins is unclear. One hypothesis is that they are simply protein spacer elements between functional domains. Another suggests that polyalanine stretches play roles in protein-protein recognition, but this remains unknown.

Perturbations of ubiquitin activation, conjugation, and transfer to target proteins via the E1-E2-E3 enzymatic cascade, have been linked to dysregulation of neuronal homeostasis and to a growing number of neurodevelopmental disorders (Amer-Sarsour et al,

[1]Department of Cell and Developmental Biology, Faculty of Medicine, Tel Aviv University, 6997801 Tel Aviv, Israel. [2]Department of Physiology and Cell Biology, Faculty of Health Sciences, Ben-Gurion University of the Negev, 8410501 Beer Sheva, Israel. [3]The Regenerative Medicine and Stem Cell (RMSC) Research Center, Ben-Gurion University of the Negev, 8410501 Beer Sheva, Israel. [4]The Zelman Center for Neuroscience, Ben-Gurion University of the Negev, 8410501 Beer Sheva, Israel. [5]School of Neurobiology, Biochemistry and Biophysics, the George S. Wise Faculty of Life Sciences, Tel Aviv University, 6997801 Tel Aviv, Israel. [6]Department of Physiology and Pharmacology, Faculty of Medicine, Tel Aviv University, 6997801 Tel Aviv, Israel. [7]Department of Otolaryngology, Head and Neck Surgery, Hillel Yaffe Medical Center, Hadera, Israel. [8]Rappaport Faculty of Medicine, Technion, Haifa, Israel. [9]Department of Neurology, Hillel Yaffe Medical Center, Hadera, Israel. [10]Sagol School of Neuroscience, Tel Aviv University, 6997801 Tel Aviv, Israel. [11]These authors contributed equally: Fatima Amer-Sarsour, Daniel Falik. [12]These authors jointly supervised this work: Gad D Vatine, Avraham Ashkenazi. ✉E-mail: vatineg@bgu.ac.il; ashkenaziavi@tauex.tau.ac.il

2021; Lee et al, 2013; Matsuura et al, 1997). One example is the UBA6-UBE2Z/USE1 ubiquitin transfer system that is important for neuronal homeostasis and development (Jin et al, 2007, Lee et al, 2013; Pelzer et al, 2007). Here, we describe a polyalanine motif in the ubiquitin-conjugating E2 enzyme USE1, which contributes to USE1 ubiquitin loading by the E1 ubiquitin activating enzyme, UBA6. In addition, our results identify a domain in UBA6 that recognizes polyalanine-containing proteins and demonstrate that, under disease conditions, UBA6 preferentially interacts with different polyalanine-expanded proteins, thereby competing with USE1 binding. Since similar effects could be confirmed in neurons derived from patients with disease-causing mutations, we suggest that this represents a previously undescribed vulnerability caused by polyalanine expansion mutations.

# Results

## The polyalanine domain of USE1 regulates ubiquitin transfer

A search for proteins containing polyalanine stretches in the ubiquitin cascades identified a small subset of E3 ubiquitin ligases and the E2 ubiquitin-conjugating enzyme, USE1 (Appendix Fig. S1A). USE1 has both N- and C-terminal extensions as well as the ubiquitin-conjugating (UBC) core domain (Fig. 1A), classifying it as a class IV E2 enzyme (Ye and Rape, 2009). USE1 can be specifically loaded with ubiquitin or ubiquitin-like FAT10 by the E1 ubiquitin activating enzyme, UBA6, via a transthiolation reaction (Aichem et al, 2010; Jin et al, 2007; Lee et al, 2011; Pelzer et al, 2007). The polyalanine stretch of USE1 is located in the N-terminal extension and is well conserved in primates, several other mammals, and reptiles (Fig. 1A, Appendix Fig. S1B). In order to examine the role of the alanine stretch in USE1 ubiquitin loading, we replaced two alanine residues in the stretch with arginine residues. When transfected into HEK293T cells, the ubiquitin loading of the USE1 2 A > 2 R mutant was significantly reduced in comparison to wild-type USE1 (Fig. 1B). Exposure of cell lysates to the reducing agent β-mercaptoethanol, abolished USE1 loading under all examined conditions, suggesting that the additional band observed is ubiquitin conjugated via a thiolation reaction, similarly to a catalytic dead enzyme (USE1 C188A) that cannot be loaded with ubiquitin (Fig. 1B). Under non-reducing conditions, the ubiquitin loading of wild-type USE1 and USE1 2 A > 2 R was abolished in UBA6-siRNA-depleted cells (Fig. 1B) suggesting that the difference in ubiquitin loading between the wild type and 2 A > 2 R USE1 is dependent on UBA6 activity.

We constructed additional USE1 mutants including one with a deletion of the polyalanine stretch (USE1 ΔPolyAla), and a deletion of Loop B (USE1 ΔLB), which is hyperactive (Schelpe et al, 2016) (Fig. EV1A). The USE1 ΔPolyAla mutant exhibits reduced ubiquitin loading compared to wild type USE1 or to USE1 ΔLB both in cells and in vitro with purified recombinant UBA6 (Figs. 1C and EV1A,B). We next mutated the endogenous polyalanine stretch of USE1 by generating a knockout HEK293T cell line harboring a deletion of the polyalanine stretch in the UBE2Z alleles (USE1 ΔPolyAla KO). The endogenous USE1-UBA6 interaction in these cells was detected with Förster resonance energy transfer (FRET) based fluorescence lifetime imaging microscopy (FLIM) and with

immunoprecipitation. The results indicated less binding between UBA6 and USE1 in the USE1 ΔPolyAla KO cells compared to control cells (Figs. 1D and EV1C).

E6AP/UBE3A is a highly potent E3 ubiquitin ligase whose regulation is critical for proper development of the nervous system. Indeed, decreased activity of the ligase results in Angelman syndrome while increased activity causes autism spectrum disorders (Glessner et al, 2009; Kishino et al, 1997; Matsuura et al, 1997). Harper and co-workers identified a unique regulatory cascade by which UBA6-USE1 ubiquitinates E6AP for degradation in the proteasome (Lee et al, 2013). We therefore examined the stability of E6AP in the USE1 ΔPolyAla KO cells. In control HEK293T cells, a cycloheximide chase experiment revealed that E6AP has an apparent half-life of approximately 10–14 h (Fig. 1E), which is consistent with previous reports in cultured cells (Lee et al, 2013). In contrast, E6AP was stabilized in the USE1 ΔPolyAla KO cells and showed a decrease in Lys48-linked polyubiquitination (Fig. 1E,F), suggesting that proteasome-mediated degradation of E6AP is inhibited in the USE1 ΔPolyAla KO cells. Since E6AP is monoubiquitinated and polyubiquitinated by the UBA6-USE1 cascade (Lee et al, 2013), we further investigated the role of the polyalanine stretch of USE1 in E6AP ubiquitination by incubating purified E6AP with UBA6, USE1 WT and USE1 mutants (ΔPolyAla and C188A) in vitro. The results indicated that the incubation with the USE1 ΔPolyAla decreased the polyubiquitination of E6AP but not the monoubiquitin conjugate, similarly to the effects of the USE1 C188A catalytically dead mutant (Fig. 1G). It is plausible that under these experimental conditions, E6AP undergoes direct monoubiquitination by E1 as demonstrated previously for other ubiquitin binding domain containing proteins (Hoeller et al, 2007). These findings provide further support that the polyalanine stretch contributes to the recognition of USE1 by UBA6.

## Isolated polyalanine stretches compete with USE1

Mutations in the polyalanine domain of USE1 may indirectly affect the structural integrity and compromise binding of UBA6 to another part of USE1. We therefore examined whether isolated polyalanine stretches (19 Ala residues) interact with UBA6. We transfected HEK293T cells with GFP-19Ala, and monitored the binding to UBA6, or the possible competition with USE1. The results indicated that the polyAla stretch binds UBA6 and that this interaction decreases the USE1-UBA6 binding (Fig. EV1D). The ubiquitin-activating E1 enzymes, UBA1 and UBA6 share 40% identity of protein sequence and a strong specificity for their cognate ubiquitin-like proteins (Bohnsack and Haas, 2003; Schulman and Harper, 2009; Whitby et al, 1998). The ubiquitin fold domain of E1 interacts with the α1 helix of E2 and is responsible for determining the selectivity of E2s for different ubiquitin-like proteins (Lee and Schindelin, 2008). UBA1 and UBA6 assume the same domain architecture and folds. Therefore, to identify additional regions in UBA6 that are likely to determine the specificity to USE1, we sought to compare the physico-chemical properties of UBA6 to those of UBA1 (Fig. 2A). Using AlphaFold we constructed a model of UBA6, and intriguingly, during the revision of the paper, the crystal structures of UBA6 have been published (Truongvan et al, 2022; Yuan et al, 2022). These experimental data confirmed the AlphaFold model with very minor differences. Comparison of the models for the general structure containing 698 Cα atoms yielded an RMSD of 1.2 Å. For the second catalytic cysteine half (SCCH)

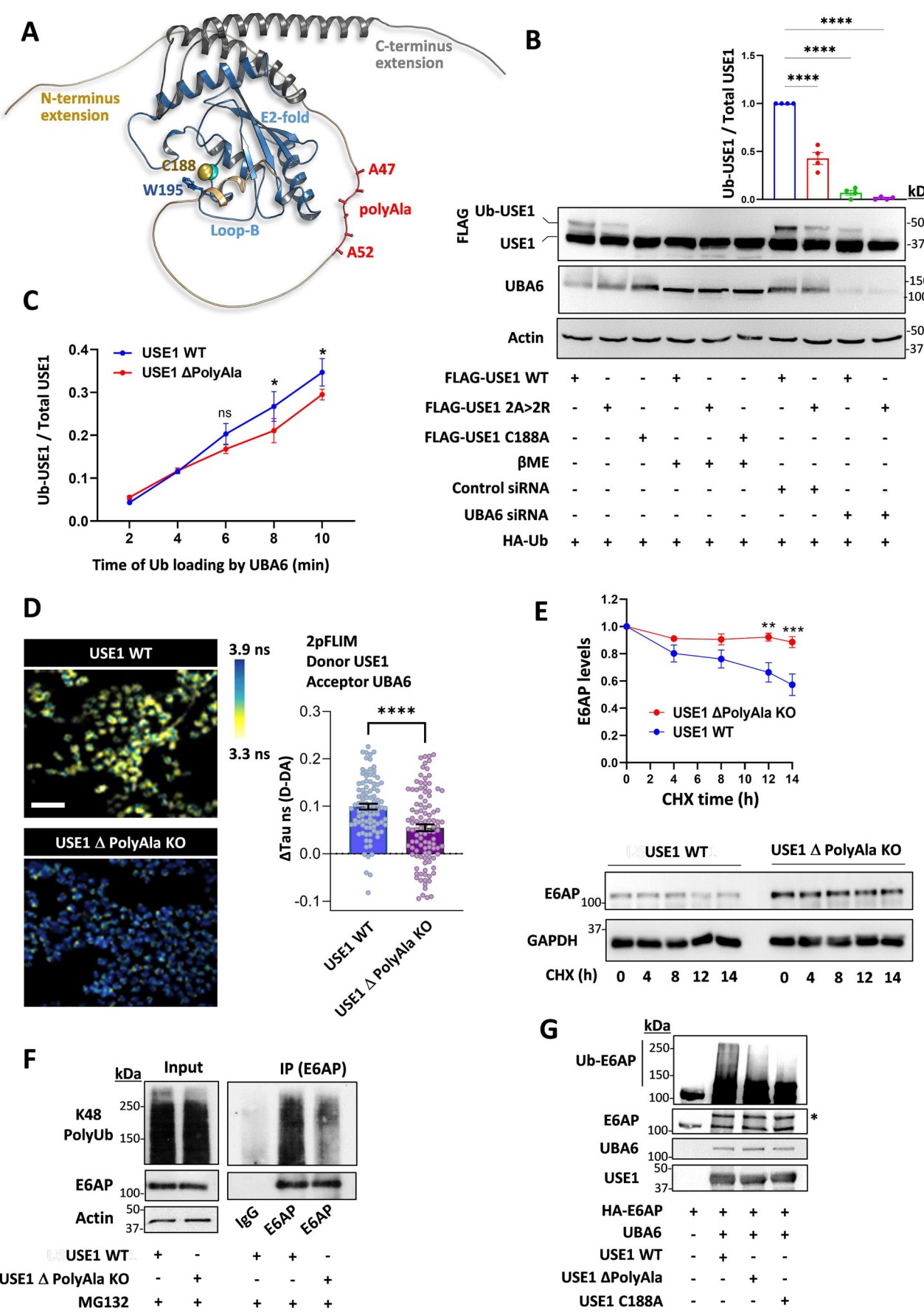

Figure 1.   A polyalanine stretch in USE1 regulates UBA6-USE1 ubiquitin transfer.

(A) Structure of the USE1 enzyme (PDB 5A4P) indicating the catalytic cysteine (Cys188), loop B (LB) with Trp195 masking Cys188, and model of the C-and N-terminal extensions including the alanine repeats (created by AlphaFold). (B) FLAG-wild type (WT) USE1, FLAG-USE1 mutant with aberrations in the polyalanine (2 A > 2 R), and FLAG-USE1 catalytic dead mutant (C188A) were co-expressed with HA-Ub in control or UBA6-depleted (*Uba6* small inhibitory RNA, siRNA) HEK293T cells. Cell lysates were incubated with or without β mercaptoethanol (βME) and analyzed for ubiquitin loading. Results are normalized to control WT USE1. n = 4 biological replicates. (C) Time-dependent in vitro ubiquitin loading of WT and ΔPolyAla USE1 by UBA6. A Representative blot is shown in Figure EV1B. Quantification of USE1 ubiquitin loading is presented for n = 3 biological replicates. (D) Quantification of UBA6-USE1 interaction in WT and USE1 ΔPolyAla knockout (KO) HEK293T cells using FLIM-FRET. Representative 2pFLIM pseudo-colored images of WT cells and ΔPolyAla KO cells, stained for USE1 and UBA6 using secondary antibodies as donor (Alexa 488) and acceptor (Alexa 555), respectively. Scale bar is 200 μm. The comparison of the difference in lifetime (donor only to donor and acceptor lifetime) in nano seconds (ns) of WT and ΔPolyAla KO cells is shown (Control 0.099 ± 0.006, KO 0.055 ± 0.007) for n = 97 and 108 cells, respectively. (E) WT and ΔPolyAla KO cells were treated with cycloheximide (CHX) at the indicated times, and were analyzed for E6AP levels. Results are normalized to t = 0 for each type of cells. n = 4 biological replicates. (F) WT and ΔPolyAla KO cells were incubated for the last 6 h with the proteasome inhibitor MG132 (10 μM), and E6AP was immunoprecipitated from cell lysates for ubiquitination analysis (under reducing conditions with βME). (G) E6AP purified from HEK293T cells was incubated with bacterially produced UBA6, USE1 WT and USE1 mutants (ΔPolyAla and C188A) for an in vitro E6AP ubiquitination assay. E6AP ubiquitin conjugates (under reducing conditions with βME) was resolved by SDS-page (Asterisk (*) indicates the mono-ubiquitin conjugate). Data information: Data points in (B–E) represent mean ± s.e.m. P values were calculated by One-way ANOVA Dunnett's test (B), Two-way ANOVA Sidak's test (C, E), Unpaired 2-tailed t test (D). *P < 0.05, **P < 0.01, ****P < 0.0001, ns non-significant. Source data are available online for this figure.

domain alone (168 Cα), which assumed a slightly different orientation compared with the enzyme core, the RMSD was 0.78 Å. Interestingly, comparison of the structures of UBA1 and UBA6 at the interface with the E2-α1 as well as the E2-α1 helices themselves appear to be very similar. However, a projection of calculated electro-potential properties on the surfaces of the two enzymes revealed significant differences within the SCCH domains. In place of the negative groove seen in UBA1, UBA6 contains a large positively charged groove that is formed by the key residues Lys628, Arg691, Lys709, and Lys714 (Fig. 2A). Similarly, a structural comparison of the canonical E1 enzymes UBA1, 2, 3, 6, and 7 (Huang et al, 2007; Lee and Schindelin, 2008; Lois and Lima, 2005; Schulman and Harper, 2009) reveals significant differences in the SCCH domains (Fig. EV2A). Moreover, a recent study demonstrated that replacing the SCCH domain of UBA1 with the one of UBA2 allows the engineered UBA1 to load ubiquitin onto the E2 of SUMO (UBC9) (Akimoto et al, 2022). Consequently, we hypothesize that the SCCH domain of UBA6 may determine the specificity for USE1. Indeed, a biophysical analysis revealed that a peptide of 7 alanine residues interacts directly with the SCCH domain of UBA6, but has significantly weaker binding to the SCCH domain of UBA1 (Fig. EV2B). To further test this hypothesis, we constructed UBA6 mutants with the positive residues in the SCCH domain replaced by Ala (UBA6 mut 4Ala) or Asp (UBA6 mut 4Asp). The USE1 binding of both these mutants in HEK293T cells is indeed significantly reduced compared to wild type UBA6 (Fig. 2B) as is the interaction of both mutants with isolated polyalanine stretches (19Ala) (Fig. 2C). In vitro, the ability of both recombinant UBA6 mut 4Ala and UBA6 mut 4Asp to load USE1 with ubiquitin is lower than that displayed by wild type UBA6 (Fig. 2D). Interestingly, neither mutations in the positive groove of UBA6 nor aberrations in the polyAla sequence in USE1 are sufficient to abrogate ubiquitin transfer from UBA6 to USE1. This suggests that while these two regions are important for specificity they are not essential for the general mechanism of ubiquitin activation (i.e., adenylation) or transthiolation.

## Disease-causing proteins with polyalanine expansions interact with UBA6

Since UBA6 is mainly localized in the cytoplasm in HEK293T cells (Appendix Fig. S2A), we next expressed the GFP-19Ala isolated stretches with or without a nuclear localization sequence (NLS),

and monitored their binding to UBA6. As expected, polyalanine stretches bind UBA6 effectively when expressed without an NLS (Fig. 3A) and reduce the ubiquitin loading of USE1 (Appendix Fig. S3A). However, in contrast, isolated polyalanine stretches with an NLS do not bind UBA6 and have no apparent effect on USE1 loading (Fig. 3A, Appendix Fig. S3A). Biochemical analysis of the GFP-19Ala could not detect a sarkosyl-insoluble fraction, suggesting that the soluble polyalanine stretches can interact with UBA6 (Appendix Fig. S3B). To explore pathological UBA6 interactions, we expressed various disease-causing proteins with polyalanine expansion mutations in cells (Appendix Fig. S2B–E) and compared their interactions with UBA6 to those of the wild type protein (Fig. 3B–E). This includes PHOX2B (WT, +13 Ala, Fig. 3B), RUNX2 (WT, +6 Ala, +12 Ala, Fig. 3C), HOXD13 (WT, +10 Ala, Fig. 3D), and PABPN1 (WT, +7 Ala, Fig. 3E).

These pathological interactions are measurable because although UBA6 is mainly localized to the cytoplasm and, as expected, wild type PHOX2B, RUNX2, HOXD13, and PABPN1 are primarily localized to the nucleus, the mutant proteins with polyalanine expansions are partially mislocalized to the cytoplasm where they can interact with endogenous UBA6 (Appendix Fig. S2B–E, Fig. 3B–E). Consequently, the interactions between UBA6 and the mutant proteins (PHOX2B + 13 Ala, RUNX2 + 12 Ala, HOXD13 + 10 Ala, and PABPN1 + 7 Ala) are measurable in the cytoplasmic but not in the nuclear fraction of the cells (Fig. EV3A–D), which reduces the USE1 ubiquitin loading (Fig. EV3E).

## Inhibition of UBA6-depndent E6AP degradation by polyalanine expansions

Our results indicate that recombinant mutant PHOX2B (+13 Ala) competes with USE1 for binding to UBA6 in vitro (Fig. 3F). To test the effects on E6AP degradation, we measured the stability and ubiquitination of E6AP in mutant PHOX2B expressing HEK293T cells. Mutant PHOX2B increases the levels of E6AP due to its stabilization in HEK293T cells (Fig. 3G, Appendix Fig. S3C). However, mutant PHOX2B does not increase E6AP mRNA levels, indicating that the increased E6AP levels are not related to transcriptional effects (Fig. 3H). This correlates with a decrease in Lys48-linked polyubiquitination of E6AP (Figs. 3I and EV3F,G), suggesting that proteasome-mediated degradation of E6AP is inhibited by mutant PHOX2B in the cells. Moreover, the increase

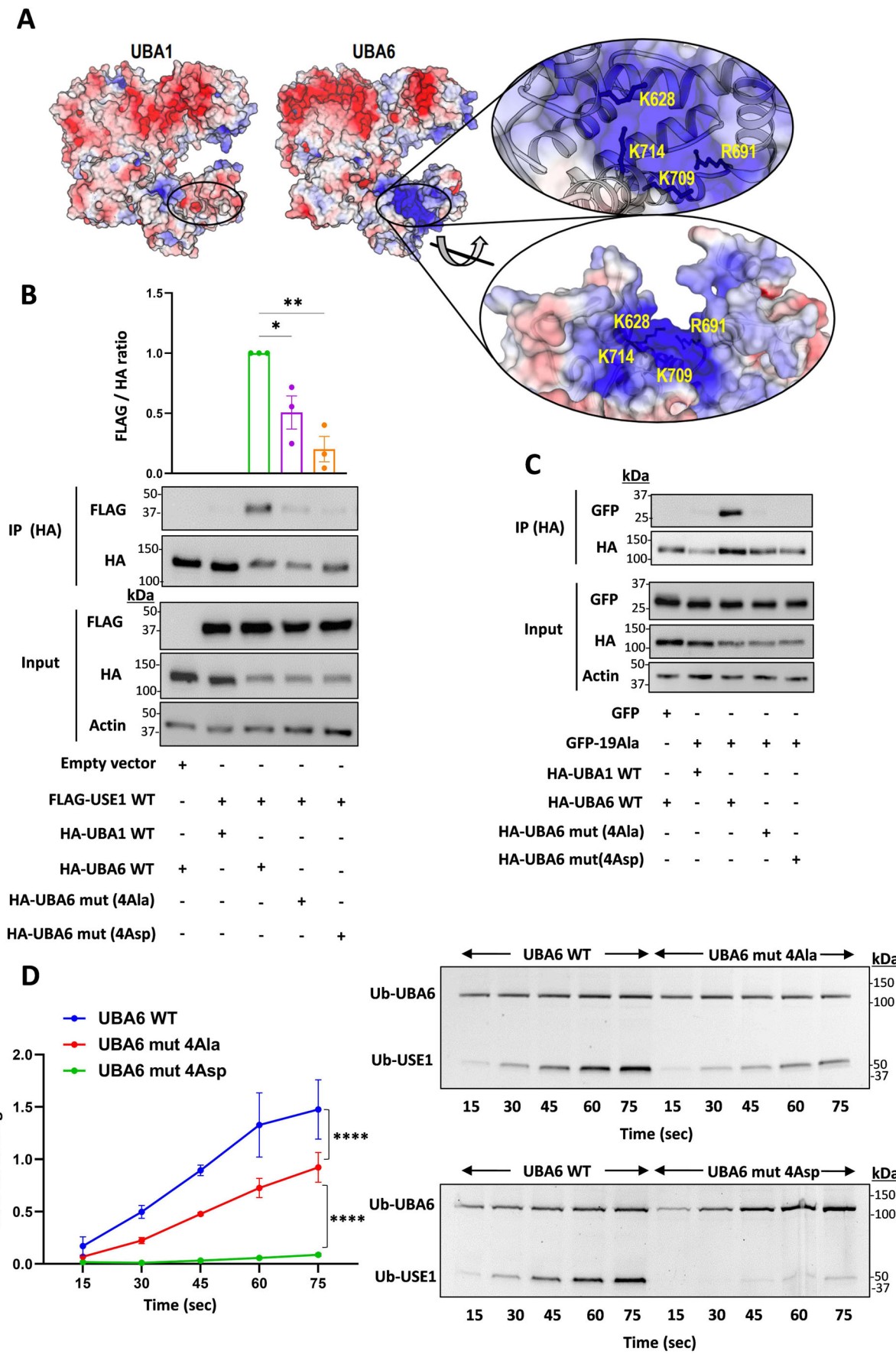

**Figure 2. UBA6 interacts with polyalanine stretches.**

(A) Electrostatic surface representation of the UBA6 structural model in comparison to UBA1. The location of key Arg and Lys residues forming the positively charged patch in UBA6 is presented. (B, C) HA-tagged constructs of WT UBA1, WT UBA6, and UBA6 mutants with Ala or Asp substitution mutations in Lys628, Arg691, Lys709, and Lys714 (UBA6 mut 4Ala or UBA6 mut 4Asp) were co-transfected with FLAG-USE1 (B) or empty GFP and GFP-polyAla (19Ala) (C) into HEK293T cells. Cell lysates were immunoprecipitated with anti-HA antibodies and the immunocomplexes were analyzed with anti-FLAG, anti-HA, and anti-GFP antibodies. (D) Representative gel of time-dependent in vitro ubiquitin loading of USE1 by WT UBA6, UBA6 mut 4Ala, and UBA6 mut 4Asp in the presence of fluorescein-labeled ubiquitin. Imaging of fluorescent ubiquitin conjugates was carried out with a laser scanner at 488 nm. Quantification of USE1 ubiquitin loading is presented. Data information: Data points in (B–D) represent mean ± s.e.m. $n = 3$ biological replicates. $P$ values were calculated by one-way ANOVA Dunnett's test (B, C) or two-way ANOVA Tukey's test (D). *$P < 0.05$, **$P < 0.01$, ****$P < 0.0001$. Source data are available online for this figure.

in E6AP levels seen in the polyalanine-expressing cells, could be reversed by UBA6 overexpression (Appendix Figure S3D), which is compatible with a model in which the polyalanine stretch impairs the regulation of E6AP levels by competing with USE1 on UBA6.

In order to test our model in neuronal cells, we next used mouse primary cortical neurons. Similarly, UBA6 was predominantly detected in the cell body and in neurites, with only a small fraction located in the nucleus (Fig. 4A). Transduction of the neurons with lentiviruses expressing GFP-tagged wild type or mutant PHOX2B (+13 Ala) resulted in a significant difference in subcellular localization and expression pattern. While most of the wild type PHOX2B could be detected in the nucleus, mutant PHOX2B (+13 Ala) is also present in the cell body and in neurites, where the GFP fluorescence imaging and biochemical analysis revealed both non-aggregated and aggregated patterns (Figs. 4B,C and EV4A). Neurons expressing mutant PHOX2B exhibit increased colocalization with UBA6, which correlates with increasing length of the polyalanine stretch (Fig. 4D). An analysis of UBA6 abundance revealed that UBA6 is predominantly associated with the non-aggregated forms of PHOX2B (Fig. 4E). Moreover, expression of mutant PHOX2B with +7 Ala and +13 Ala increase the levels of E6AP in the primary neurons by 1.3 fold (Fig. EV4B), which is consistent with physiological induction rates of E6AP protein levels in cultured neurons (Greer et al, 2010). The synaptic activity-regulated cytoskeleton-associated protein (Arc) is negatively regulated by E6AP (Greer et al, 2010; Kuhnle et al, 2013), and accordingly, the increase in E6AP levels due to ectopic expression of the PHOX2B mutants with +7 Ala and +13 Ala decrease Arc levels in primary neurons (Fig. EV4B,C). To measure the effects of mutant PHOX2B on the stability of E6AP and Arc, we performed cycloheximide chase experiments in transduced neurons expressing WT or mutant PHOX2B (+13Ala). While E6AP levels remained stable in the mutant PHOX2B-expressing neurons, Arc was more rapidly degraded (Fig. 4F,G). These alterations appear to be dependent on UBA6, because overexpression of UBA6 in the mutant PHOX2B-expressing neurons decreases E6AP and increases Arc levels (Fig. EV4D–F).

## UBA6 dysregulation in fibroblasts and neurons of patients with polyalanine expansions

In order to investigate the mechanisms underlying cellular vulnerabilities caused by the polyalanine expansion mutations in humans, we sought to assess the associations between endogenous polyalanine expansions and UBA6. The cricopharyngeal muscle of OPMD patients is vulnerable to the polyalanine expansion mutations in the *PABPN1* gene (Brais et al, 1998). OMPD patient-derived cricopharyngeal myotubes and primary fibroblasts as well as non-affected control

fibroblasts, express PABPN1 both in the cytoplasm and in the nucleus (Appendix Fig. S4A,B), which could be related to the ability of PABPN1 to shuttle between the nucleus and the cytoplasm (Calado et al, 2000). Accordingly, OPMD patient-derived fibroblasts have less interaction between UBA6 and USE1 than control fibroblasts (Appendix Fig. S4C–E).

Endogenous PHOX2B is expressed in specific neuronal populations of the autonomic nervous system (Dauger et al, 2003; Pattyn et al, 1999). In humans, polyalanine expansion mutations within the *PHOX2B* gene cause CCHS, a rare and life-threatening condition with autonomic nervous system dysfunction (Trang et al, 2014; Vanderlaan et al, 2004). We have previously described the generation of two induced pluripotent stem cells (iPSCs) from identical twins carrying a heterozygous PHOX2B + 5 Ala expansion (101iCCHS 20/25 and 102iCCHS 20/25) (Falik et al, 2020). Here, we obtained skin punch biopsies from an additional CCHS patient bearing a heterozygous +7 Ala expansion (104iCCHS 20/27), as well as from their sex-matched healthy family relatives, which serve as controls (103iCTR 20/20 and 105iCTR 20/20) (Appendix Fig. S5). These patients suffer from central sleep apnea with a more severe peripheral autonomic presentation in 104iCCHS 20/27, including Hirschsprung disease (Sharabi et al, 2022). Patient-derived fibroblasts were reprogrammed using non-integrating episomal plasmids as previously described (Falik et al, 2020). All generated iPSC lines expressed the pluripotency markers NANOG, SOX2, OCT3/4 TRA-1-60, and SSEA4 (Appendix Fig. S5A–J), and spontaneously differentiated into the three germ layers (Appendix Fig. S5L–N). All lines possess a normal karyotype, and genetic analysis confirms the presence of a heterozygous expansion of seven alanine residues resulting in a 27 polyalanine stretch in PHOX2B (Appendix Fig. S5K,O). In order to obtain endogenous PHOX2B-expressing cells, we differentiated the iPSCs to neural crest progenitor cells that were further differentiated into autonomic neuroblast cells, and finally to peripheral autonomic neurons (Fig. 5A). Characterization of the human autonomic neurons reveals expression of PHOX2B, the pan-neuronal marker βIII-tubulin, the catecholaminergic marker tyrosine hydroxylase (TH), the peripheral neuronal marker peripherin, and atonal BHLH transcription factor 1 (ATOH1), which is associated with breathing and digestion (Fig. 5A).

Analysis of the PHOX2B localization revealed a decrease in the nuclear fraction of PHOX2B in the mutant 20/25 and 20/27 neurons, as compared to healthy control neurons (Fig. EV5A,B). The cytoplasmic PHOX2B is soluble, and predominantly perinuclear without visible aggregates (Fig. EV5C,D). In addition, there was an increased association between UBA6 and PHOX2B in the CCHS patient neurons (Fig. 5B), accompanied by a decrease in the interaction between UBA6 and USE1 (Fig. 5C). Some CCHS

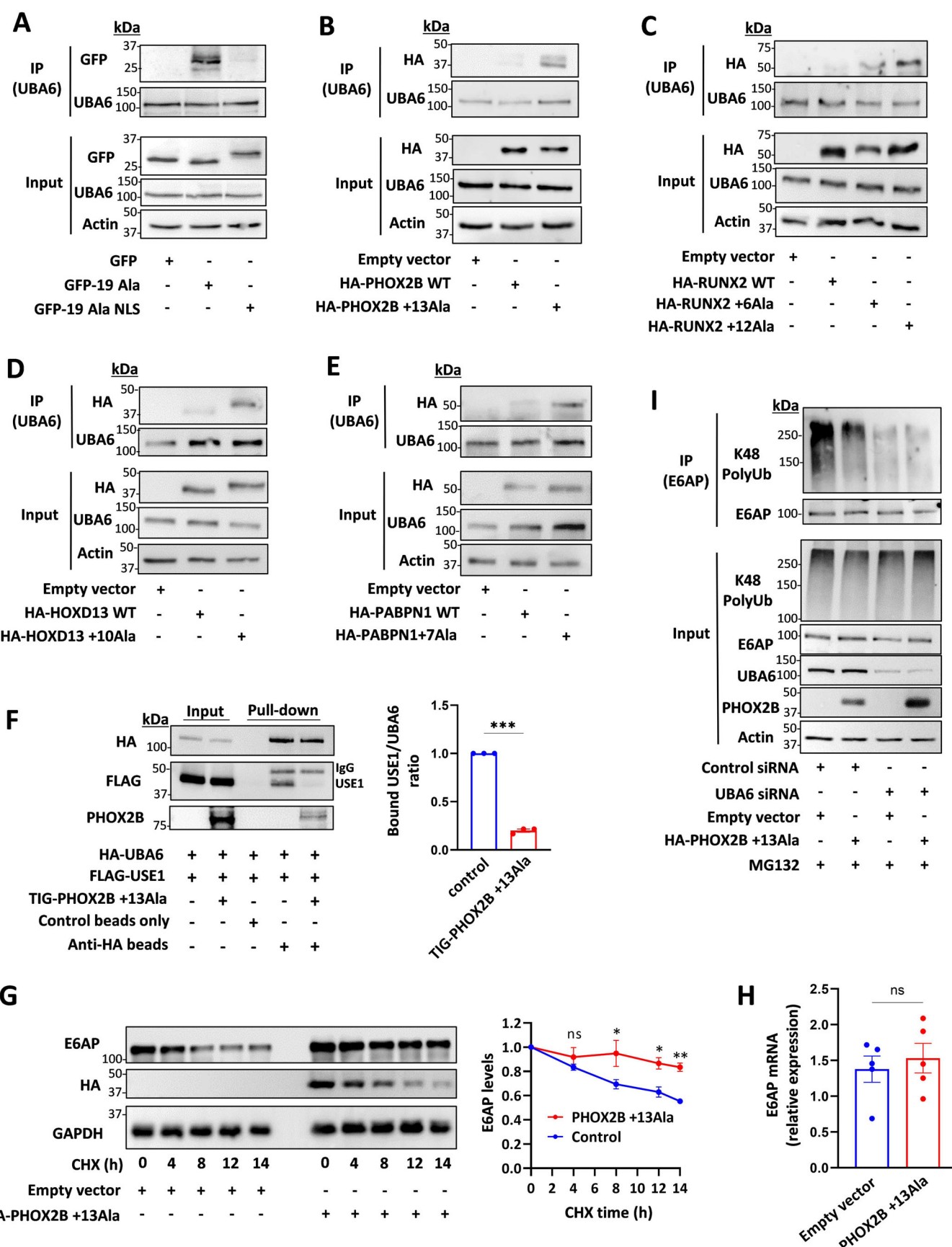

**Figure 3. Polyalanine-expanded disease proteins interact with UBA6 and inhibit E6AP degradation.**

(A–E) HEK293T cells were transfected with the indicated constructs and immunoprecipitated for endogenous UBA6. (A) Empty GFP, and GFP-polyAla with or without a nuclear localization sequence (NLS). (B) WT and mutant PHOX2B (+13Ala). (C) WT and mutant RUNX2 (+6Ala and +12Ala). (D) WT and mutant HOXD13 (+10Ala). (E) WT and mutant PABPN1 (+7 Ala). (F) Cell lysates of HEK293T cells expressing HA-UBA6 and FLAG-USE1 were incubated with recombinant TIG-mutant PHOX2B. Anti-HA beads were used to pulldown UBA6 (beads only were used as control). The bound USE1/UBA6 ratio is shown. $n = 3$ biological replicates. (G) Cells were transfected with mutant PHOX2B or empty vector. The cells were treated with cycloheximide (CHX) at the indicated times, and were analyzed for E6AP levels. Results are normalized to $t = 0$. $n = 3$ biological replicates. (H) Quantification of E6AP mRNA in the cells transfected with empty vector or mutant PHOX2B. $n = 5$ biological replicates. (I) Control and UBA6-depleted HEK293T cells were transfected with mutant PHOX2B or empty vector and incubated for the last 6 h with the proteasome inhibitor MG132 (10 μM). Endogenous E6AP was immunoprecipitated from cell lysates for ubiquitination analysis. Data information: Data points in (F–H) represent mean ± s.e.m. $P$ values were calculated by paired 2-tailed $t$ test (F), Two-way ANOVA Sidak's test (G) and unpaired 2-tailed $t$ test (H). *$P < 0.05$, **$P < 0.01$, ***$P < 0.001$, ns non-significant. Source data are available online for this figure.

neurons exhibited severe PHOX2B cytoplasmic mislocalization, with apoptotic nuclear morphology (1.9% and 3.9% of the 20/25 and the 20/27 mutations, respectively) (Fig. EV5D). In accordance with the mouse data, the levels of E6AP protein are higher in the patient neurons (Fig. 5D), although no change in E6AP mRNA was detected (Fig. 5E). Moreover, Arc levels were lower in the CCHS neurons with the 20/25 mutation than in control cells with the levels further reduced in CCHS neurons with the 20/27 mutation (Fig. EV5E,G). In addition to E6AP, the UBA6-USE1 cascade has been shown to regulate the levels of the synaptic protein shank3 (Lee et al, 2013). We could not detect endogenous expression of shank3 in the human autonomic neurons (Fig. EV5H), thereby precluding the analysis of UBA6-USE1 effects on shank3 in these neurons.

Finally, we asked how does UBA6 affect polyalanine expanded protein-associated toxicity? Because we observed apoptotic nuclear morphology in the CCHS patient neurons (Fig. EV5D), we attempted to rescue the cell death in CCHS neurons by their transduction with lentiviruses expressing the cDNA of UBA6 (Fig. 5F,G), which resulted in robust UBA6 expression in both cell body and neurites (Fig. EV5I). Interestingly, the overexpression of UBA6 significantly reduced the number of CCHS neurons positive to terminal deoxynucleotidyl transferase dUTP nick end labeling (TUNEL) (Fig. 5G), and also decreased the amounts of phosphatidylserine on the extracellular surface as detected by Annexin V binding (Fig. 5F). We can therefore conclude that overexpression of UBA6 in the CCHS neurons rescues them from neuronal death.

## Discussion

Expansion mutations in stretches of repetitive DNA sequences that encode poly amino acids such as polyglutamine or polyalanine can cause various diseases. Although much progress has been made on determining the molecular mechanisms of polyglutamine diseases (Ashkenazi et al, 2017; Paulson et al, 2017; Usdin, 2008), the consequences of polyalanine expansions remain more of an enigma. Here, we demonstrate how expanded polyalanine stretches can compete with the normal function of a shorter stretch to disrupt the specific interaction between the E2 ubiquitin-conjugating enzyme USE1 and the E1 ubiquitin-activating enzyme, UBA6. Our results indicate that UBA6 interacts with polyalanine stretches, and that this interaction contributes to UBA6 recognition of USE1, but can be altered by different polyalanine disease-causing proteins.

The correlation between polyalanine expansion mutations and a possible harmful gain-of-function is supported by the observation that the respiratory abnormalities of $phox2b^{+/-}$ mouse mutants are

transient and milder than those exhibited by $phox2b^{+/27Ala}$ mouse mutants or those of CCHS patients (Dubreuil et al, 2008; Ramanantsoa et al, 2006). These genetic studies suggest the possibility of additional disease contributing mechanisms besides partial loss of PHOX2B function in the nucleus and alterations in transcriptional programs by polyalanine expansion mutations, as was recently described for HOXD13 (Basu et al, 2020). UBA6 plays an important role in mouse embryonic development, neuronal function, and survival (Chen et al, 2020; Lee et al, 2013). Thus, inhibition of UBA6 by sequestering the enzyme into soluble cytoplasmic polyalanine expansions may represent a deleterious mechanism for such mutations. The correlation between disease severity and the length of the polyalanine expansion may be related to the greater tendency of mutants with longer polyalanine stretches to mislocalize to the cytoplasm and interact there with UBA6. Indeed, the expanded polyalanine domain itself can serve as a nuclear export signal (Li et al, 2017). This tendency of polyalanine-expanded proteins to mislocalize to the cytoplasm has been previously detected in overexpression systems (Albrecht and Mundlos, 2005), but the relevance to human disease was questioned. Our results provide evidence that cytoplasmic mislocalization of endogenous PHOX2B can occur in the neurons affected in CCHS, which may contribute to the neuronal dysfunction seen in this disease.

In summary, our study provides mechanistic insights into the long-standing question of the functional role of polyalanine stretches and demonstrates that they are relevant to the ubiquitin system and its dysregulation in disease states.

## Methods

### Ethics statement

All mouse experiments were reviewed and approved by the Institutional Animal Care and Use Committee of Tel Aviv University. All cell lines and protocols related to human stem cell research in the present study, and the analysis of patient biopsies were used in accordance with guidelines approved by the Institutional Review Board (Sheba Medical Center, Hilel Yafe Medical Center and Tel Aviv University). Informed consent was obtained from all donors.

### Antibodies

The following antibodies were used in this study: mouse anti-HA (Biolegend 901501); rabbit anti-HA (Cell signaling 3724); rabbit

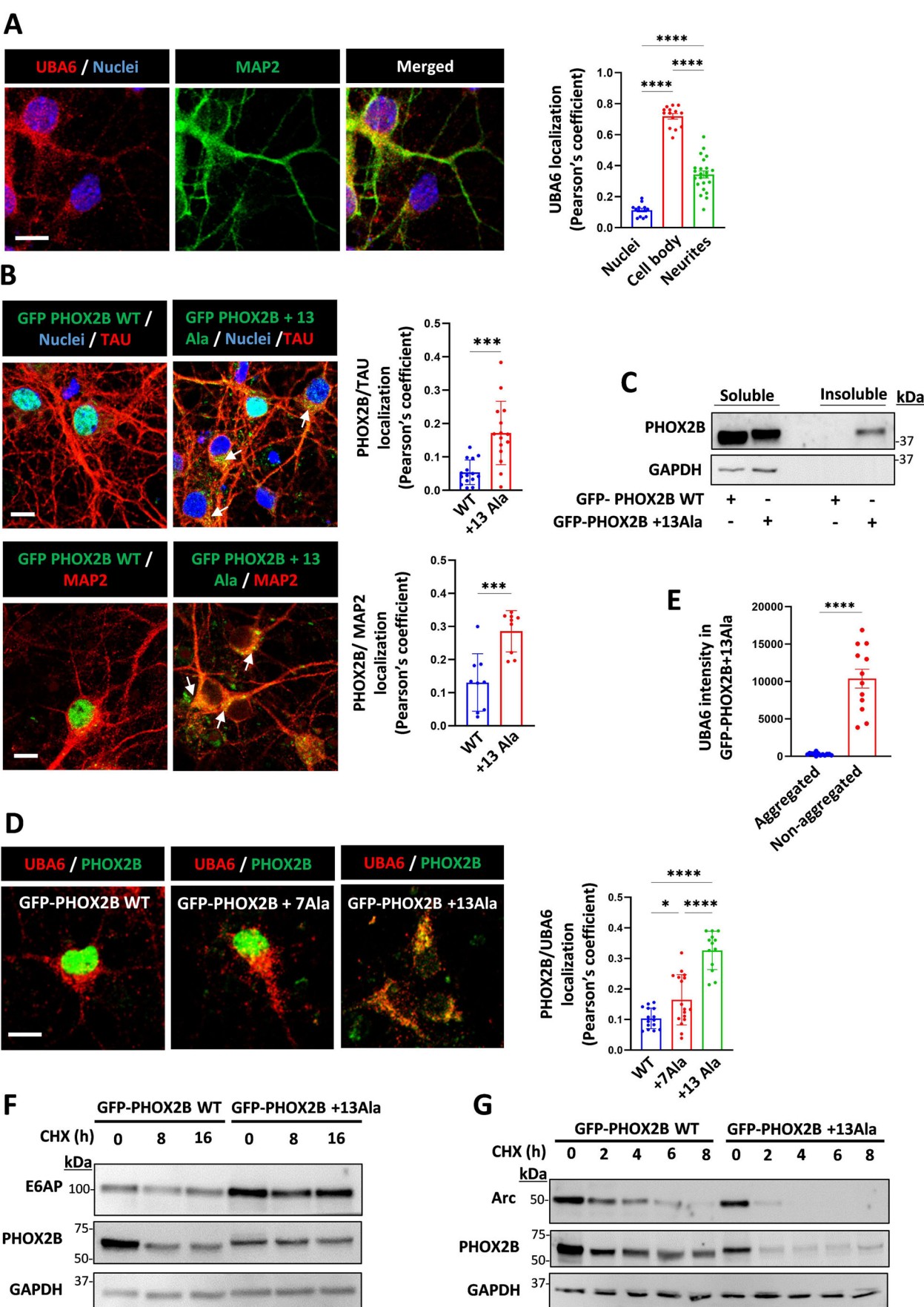

**Figure 4. Cytoplasmic mutant PHOX2B interacts with UBA6 and alters the levels of E6AP and Arc in primary neurons.**

(A) Mouse primary cortical neurons were labeled for endogenous UBA6 and MAP2. Scale bar 10 μm. Quantification of the association of UBA6 with the neuronal nucleus, cell body, and neurites is shown (Pearson's coefficient). $n = 50$ neurons analyzed. (B) The neurons were transduced with lentiviral vectors expressing GFP-tagged WT PHOX2B or mutant PHOX2B (+13Ala), and were labeled for endogenous MAP2 and TAU (non-nuclear fraction of mutant PHOX2B marked with arrows). Scale bar 10 μm. The quantification presents the association of GFP-PHOX2B with MAP2 and TAU (Pearson's coefficient). $n = 30$-50 neurons analyzed. (C) The PHOX2B transduced neurons (WT or +13Ala) were analyzed for the levels of PHOX2B in the soluble and sarkosyl-insoluble fractions. (D) The neurons were transduced with lentiviral vectors expressing GFP-tagged WT PHOX2B or mutant PHOX2B (+7 Ala and +13Ala) and were labeled for endogenous UBA6. The quantification of the association of GFP-PHOX2B with UBA6 (Pearson's coefficient) is presented. Scale bar 10 μm. $n = 30$, $n = 60$, and $n = 90$ neurons analyzed for WT PHOX2B, mutant PHOX2B +7Ala and +13Ala, respectively. (E) Additional analysis of the images in Fig. 4D for the intensity of UBA6 in PHOX2B+13Ala aggregated or non-aggregated forms as detected by GFP fluorescence. $n = 90$ neurons analyzed. (F, G) The neurons were transduced with lentiviral vectors expressing GFP-WT PHOX2B or GFP-mutant PHOX2B +13Ala and were treated with cycloheximide (CHX) at the indicated times. Representative blots are presented for the levels of E6AP (F) and Arc (G). Data information: Data points in (A, B, D, E) represent mean ± s.e.m. $P$ values were calculated by one–way ANOVA Tukey's test (A, D), unpaired 2-tailed $t$ test (B, E). $*P < 0.05$, $***P < 0.001$, $****P < 0.0001$. Source data are available online for this figure.

anti-Actin (Merck A2066); mouse anti- E6-AP (Santa Cruz 166689); mouse anti-FLAG (Merck F1804); rabbit anti-UBA6 (Cell signaling 13386); rabbit anti-K48 polyUb (Cell signaling 8081 S); rabbit anti-UBE2Z (USE1, (Abcam 254700); rabbit anti-GFP (Abcam 6556), mouse anti-PHOX2B (Santa Cruz 376997); mouse anti-mCherry (Abcam 125096); mouse anti-Tau (Santa Cruz 58860); rabbit anti-MAP2ab (synaptic systems 188003); mouse anti-Arc (BD Transduction Laboratories 612602); rabbit anti-Arc (Abcam 183183); Alexa Fluor 555 (Abcam 150114) conjugated goat anti-mouse secondary antibody, Alexa Fluor® 555 (Abcam 150078) conjugated goat anti-rabbit secondary antibody, goat anti-mouse (Abcam 6789) and goat anti-rabbit (Abcam 6721); horseradish peroxidase (HRP)-conjugated secondary antibodies, mouse anti-MATH-1 (Santa Cruz 136173), mouse anti-PRPH (Santa Cruz 377093); mouse anti-TH (Santa Cruz 25269); mouse anti-SOX10 (Abcam 155279); rabbit anti-β-Tubulin III (Merck T2200); mouse anti-OCT3/4 (Santa Cruz 5279); mouse anti-TRA 1-60 (R & D MAB4770); mouse anti-SSEA-4 (Santa Cruz 21740); mouse anti-IgG3 Isotype (Bio-legend 330405); rabbit anti-SOX2 (Abcam 97959); rabbit anti-NANOG (Abcam 21624); rabbit anti- 68 kDa Neurofilament (Abcam 52989); rabbit anti-α-Fetoprotein (ScyTek A00058); rabbit anti- α-smooth muscle actin (Abcam 32575); mouse anti-myosin heavy chain (BioTest MAB4470-SP); Alexa Fluor 488-conjugated anti-mouse secondary antibody (Jackson ImmunoResearch 715545150); Alexa Fluor 488-conjugated anti-mouse secondary antibody (Thermo Fisher Scientific, A-11034); Alexa Fluor 488-conjugated anti-rabbit secondary antibody (Molecular Probes, A-11029); Alexa Fluor 594-conjugated anti-rabbit secondary antibody (Jackson ImmunoResearch 711585152); Alexa Fluor 568-conjugated anti-rabbit secondary antibody (Abcam 175471); and Alexa Fluor 647-conjugated anti-rabbit secondary antibody (Abcam 150079).

## Cloning and constructs

E1 constructs for mammalian cell expression, namely pcDNA3.1-HA-tagged UBA1 wild type and pcDNA3.1-HA-tagged UBA6 (UBE1L2) wild type, were kindly provided by Dr. Marcus Groettrup. pcDNA3.1-HA-UBA6 mut (4Ala) and pcDNA3.1-HA-UBA6 mut (4Asp) where Lysine 628, 709,714 and Arginine 691 were changed to amino acids Alanine and Aspartate respectively, were constructed by Gibson assembly using the DNA gene blocks. The SCCH domains of UBA6 or UBA1 were constructed by introduction of PCR fragments containing amino acids 623-889 (UBA6) and 624-891 (UBA1) into plasmid pYB50 by Gibson

assembly (lab collection). For *E. coli* expression as a His$_6$-tagged proteins, the wild type and mutant UBA1 and UBA6 genes where subcloned into modified and improved vector pET28a (Shilling et al, 2020).

E2 USE1 (UBE2Z) expressing plasmids pcDNA3.1-His$_6$-3xFlag-USE1 was kindly provided by Dr. Annette Aichem. For expression in mammalian cells, the putative Kozak sequence was first added to the USE1 gene, which was subcloned into pEGFP-N1 based plasmid resulting in plasmid pCMV-His$_6$-3xFlag-USE1 new Kozak used in this study. The ΔPolyAla region mutant, where amino acids 47-56 of the USE1 gene were deleted, the 2 A > 2 R mutant where Alanine 49 and Alanine 52 were changed to Arginine respectively, the ΔLB mutant where amino acids 194–197 comprising a Loop B (Schelpe et al, 2016) were deleted, and the C188A mutant were constructed by site-directed mutagenesis and Gibson assembly applying Q5® Site-Directed Mutagenesis Kit and NEBuilder® HiFi DNA Assembly respectively. For *E.coli* expression as biotinylated His$_6$-tagged proteins, the USE1 wild type and mutant genes were subcloned into modified plasmid pET30-His$_6$-N-AviTag (Laboratory collection).

His$_6$-Ub plasmid for expression in *E. coli* of the His-tagged ubiquitin gene was described previously (Keren-Kaplan et al, 2012). The HA-ubiquitin plasmid for mammalian expression was a gift from Dr. Edward Yeh (Addgene plasmid # 18712 (Kamitani et al, 1997)). The p4054 HA-E6AP isoform II was a gift from Dr. Peter Howley (Addgene plasmid # 8658 (Kao et al, 2000)). pEGFP-C1 19 Alanines and pEGFP-C1 19 Alanines with nuclear localization sequence (NLS) constructs for expression in mammalian cells as well as HA-bovine PABPN1 wild type and HA-bovine PABPN1 mut +7 Ala mutant constructs were a gift of Dr. David Rubinsztein. pEGFP-C119 Alanines and pEGFP-C119 Alanines with nuclear localization sequence (NLS) constructs for expression in mammalian cells as well as HA-bovine PABPN1 wild type and HA-bovine PABPN1 mut +7 Ala mutant constructs were a gift of Dr. David Rubinsztein. As a first step toward deletion of the EGFP gene and insertion of the C-terminal HA tag, we deleted the PolyA region from the wild type gene performing a site-directed mutagenesis applying Inverse PCR and Q5® Site-Directed Mutagenesis Kit (New England Biolabs, NEB). Then the HA-tagged delta Ala mutant bovine gene was humanized by changing amino acids Asp 95 and Ser 102 to Ser 95 and Pro 102, respectively, applying Q5® Site-Directed Mutagenesis Kit (NEB). Following humanization, 10 Ala (wild type) and 17 Ala (+7 mutant) regions were added using gene blocks (IDT) and Gibson assembly (NEBuilder® HiFi DNA Assembly kit, NEB).

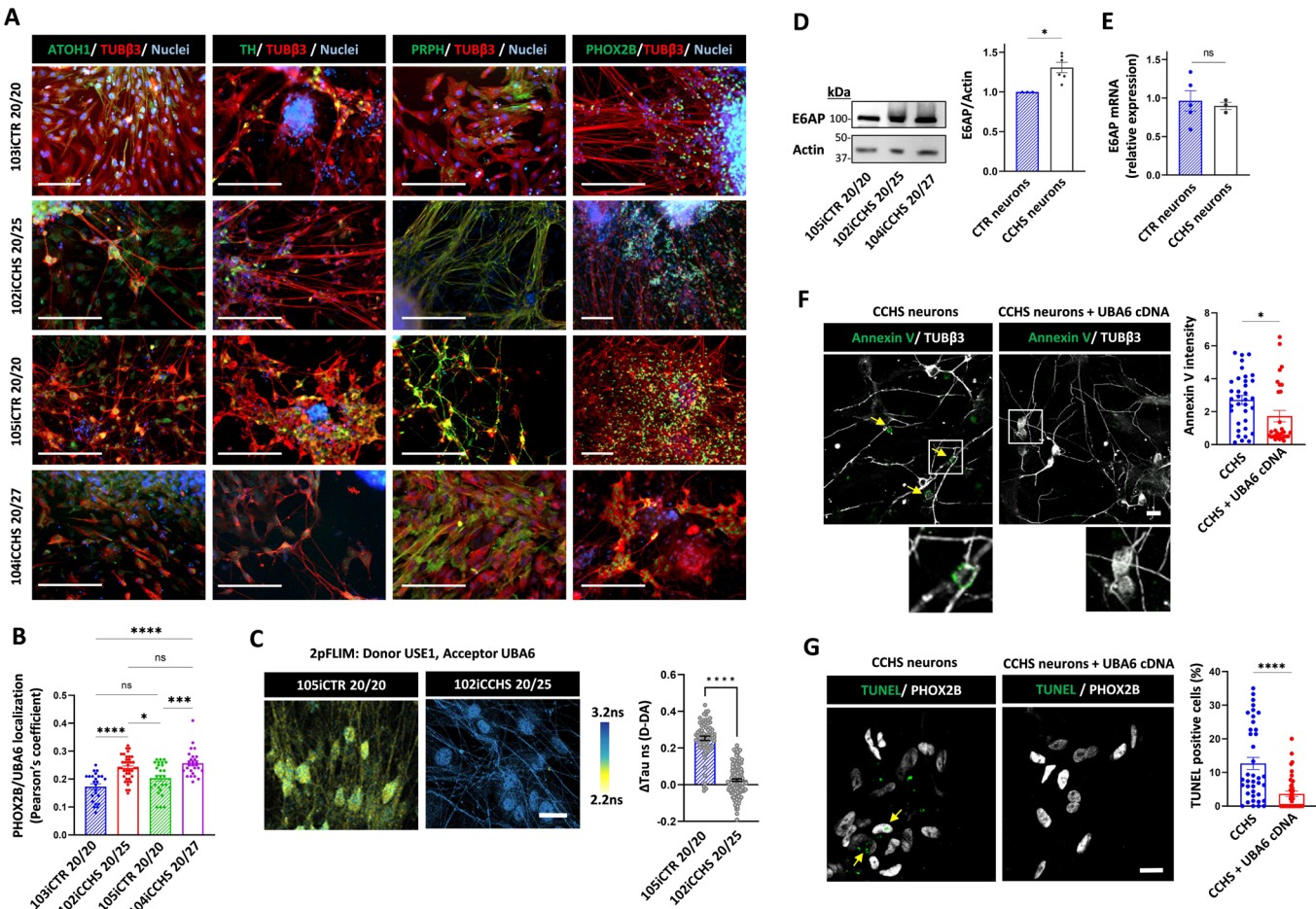

**Figure 5. UBA6 overexpression rescues CCHS patient-derived autonomic neurons from neuronal death.**

(A) Characterization of iPSC-derived autonomic neurons at day 31 of differentiation from healthy controls and CCHS patients. Immunocytochemistry of PHOX2B, βIII-tubulin (TUBβ3), tyrosine hydroxylase (TH) peripherin (PRPH), and atonal BHLH Transcription Factor 1 (ATOH1). Scale bar 200 µm. (B) Quantification of the association of endogenous UBA6 with endogenous PHOX2B (Pearson's coefficient) in autonomic neurons from control and CCHS patients. Total number of neurons analyzed in different imaged fields was 160 for 103iCTR 20/20, 350 for 102iCCHS 20/25, 380 for 105iCTR 20/20 and 550 for 104iCCHS 20/27. Images are shown in Figure EV5D. (C) Representative 2pFLIM pseudo-colored images of control and CCHS patient-derived neurons, stained for USE1 and UBA6 using secondary antibodies as donor (Alexa 488) and acceptor (Alexa 555), respectively. Scale bar is 20 µm. Comparison of the difference in lifetime for each group, for the subtraction of donor only to donor and acceptor fluorescence lifetime. Control neurons 0.253 ± 0.012, patient neurons 0.025 ± 0.007 (n = 67 and 153 cells, respectively). (D) Analysis of E6AP levels in control (105iCTR 20/20) and CCHS patients (102iCCHS 20/25 and 104iCCHS 20/27). Results are normalized to control. n = 3–6 biological replicates. (E) Quantification of E6AP mRNA in the control and patient-derived autonomic neurons. n = 3–5 biological replicates. (F, G) Patient-derived autonomic neurons (102iCCHS 20/25) were transduced with mCherry-UBA6 cDNA lentiviral vectors. (F) Representative images of Annexin V signal (colored green and marked with arrows) of the transduced and non-transduced patient neurons. Scale bar 20 µm. Quantification of cell surface Annexin V intensity in different image fields (transduced n = 80 cells, non-transduced n = 92). (G) Representative images of TUNEL staining (colored green marked with arrows) of the transduced and non-transduced patient neurons. The percentage of the TUNEL positive PHOX2B expressing cells is shown. At least 1000 PHOX2B expressing cells were analyzed. Data information: Data points in (B–G) represent mean ± s.e.m. P values were calculated by one-way ANOVA Tukey's test (B), unpaired 2-tailed t test (C, E–G) and paired 2-tailed t test (D). *P < 0.05, ***P < 0.001, ****P < 0.0001, ns non-significant. Source data are available online for this figure.

Human RUNX2 carrying plasmids pcDNA3.2/GW/D-TOPO RUNX2 wild type, pcDNA 6.2/C-EmGFP-DEST-RUNX2 mut (+6 Ala) and pcDNA 6.2/C-EmGFP-DEST- RUNX2 mut (+12 Ala) were kindly provided by Dr. Yoshihito Tokita. In this study, in all aforementioned plasmids, the EmGFP gene was deleted and HA-tag was introduced onto the C-terminus of the RUNX2 genes. The mouse HOXD13 bearing plasmids pcDNA3.1-Hoxd13 wild type and pcDNA3.1-Hoxd13 mut +10Ala were a gift of Dr. Denes Hnisz. First, the Valine259 to Glutamate mutation was corrected in both constructs and then the wild type and the mutant Hoxd13 genes were subcloned

into pEGFP-N1-derived vector while adding a putative Kozak sequence and the C-terminal HA-tag.

The PHOX2B carrying plasmids pcDNA3.0-HA-PHOX2B wild type and pcDNA3.0-HA-PHOX2B mut (+13 Ala) were kindly provided Dr. Diego Fornasari. To express the PHOX2B genes alone and as fusion with EGFP in neuronal cells under control of the human Synapsin I promoter, the PHOX2B genes were subcloned into lentiviral pLL3.7 vector bearing the Synapsin I promoter. Plasmids bearing the HA-PHOX2B mut (+7 Ala) gene were constructed applying Gibson assembly and resulted in plasmids

pcDNA3.0-HA-PHOX2B mut (+7 Ala), pLL3.7-hSyn-HA-PHOX2B mut (+7 Ala) and pLL3.7-hSyn-HA-PHOX2B mut (+7 Ala)-EGFP.

A lentiviral target vector bearing wild type UBA6-mCherry fusion protein under the control of pCMV promoter was constructed by subcloning of the *Nhe*I-*Kpn*I fragment from plasmid pLL3.7-hSyn-UBA6-mCherry into plasmid pLL3.7-pCMV-Kozak-HA-linker-EGFP (Lab collection). For *E. coli* expression of the His-tagged trigger factor (TIG)-TEV site-PHOX2B mut (+13 Ala) fusion protein, the PHOX2B mut (+13 Ala) gene was obtained by PCR from plasmid pET30-NAvitag-PHOX2B mut (+13), and introduced into *Nhe*I-*Hin*dIII sites of the plasmid pET43-His₆-TIG-TEV site-PHOX2B wild type.

The integrity of every construct used in this study was verified by the sequencing analysis and the detailed explanations of cloning procedures will be provided upon request.

## Cell lines and transfection

Cell lines used in this study include human embryonic kidney cells, HEK293T (ATCC CRL-1573) and HEK293FT (Invitrogen, R70007). The cells were authenticated by STR profiling and were routinely tested for mycoplasma contamination. The cells were grown in Dulbecco's modified Eagle's medium (01-052-1A, Sartorius) supplemented with 10% heat-inactivated fetal bovine serum (04-007-1A, Sartorius), 10000 units/ml penicillin, and 10 mg/ml streptomycin (03-031-1B Sartorius) and 2 mM L-glutamine (G7513, Merck) at 37 °C with 5% $CO_2$. Cells were seeded and cultured for approximately 24 h until they grew to 50–60% confluence before transfection. In some experiments, the cells were treated with cycloheximide (Merck, 01810) at a final concentration of 50 µg/ml in different incubation times for analysis of protein degradation rates. Transient transfection of indicated plasmids, was accomplished using TransIT-LT1(Mirus, MIR 2300) according to the manufacturer's protocol. Vector and Mirus were mixed in a reduced serum medium (Opti-MEM® 10001865, Gibco) and incubated at room temperature for up to 30 min before being dripped gently onto the cell culture and incubated for 24–48 h. Transfection efficiency was confirmed by western blot analysis. In RNA interference experiments, cells were transfected 24 h after seeding with 50–100 nM *SMART*pool siRNAs (Dharmacon) for gene silencing and Lipofectamine 2000 (1000186, Invitrogen), with two rounds of knockdown for 5 days, according to the manufacturer's instructions (Invitrogen). For this purpose, the siRNAs and Lipofectamine were diluted separately in reduced serum medium Opti-MEM® (10001865, Gibco), then mixed for 15 min at room temperature and dripped gently onto the cell culture, which was then incubated at 37 °C for 4–6 h, before restoration of full medium. The following oligonucleotides (ON-TARGETplus SMARTpool, Dharmacon L-006403-01-0005) were used for UBA6 depletion:

siRNA J-006403-09, UBA6: Target Sequence: GUGUAGAAUU AGCAAGAUU

siRNA J-006403-10, UBA6: Target Sequence: GCAUAGCUGU CCAAGUUAA

siRNA J-006403-11, UBA6: Target Sequence: CAGUGUUGUA GGAGCAAUA

siRNA J-006403-12, UBA6: Target Sequence: GGAAUUUGGU CAGGUUAU

To generate the USE1 ΔPolyAla KO cell line, the GenCRISPR™ gene editing technology was applied using a service from GenScript USA. The gRNAs cleavage efficiency was tested by transient

transfection and a gRNA with the highest cleavage efficiency was chosen (gRNA CCTGCCGGATGTGTGGGCGG) to generate the gene deletion resulted in the deletion of the domain located at position 47-56 of the UBE2Z/USE1 human protein comprising the polyalanine domain. The sequence of donor is designed as below:

ctggtgttgttggcgttagcggcagcggcggcgggttcgggccgcctttcctgccg-gatgtgtgggcccgggagcggcctggctccgctgcccgggct

The editing materials were transfected in the HEK293T cells and the transfected cells were plated in 96-well plates by limiting dilution to generate isogenic single clones. The clones were picked from wells and screened by PCR and Sanger sequencing screening to identify full allelic deletion clones.

## Isolation and culture of mouse primary cortical neurons

Primary cortical neurons were isolated from wild-type C57BL/6J mouse embryos at E17. Brains were harvested and placed in ice-cold HBSS under a dissection microscope. Cerebral cortices were dissected and incubated in HBSS. After mechanical dissociation using sterile micropipette tips, dissociated neurons were resuspended and cultured at 37°C in a humidified incubator with 5% $CO_2$ and 95% $O_2$ in poly-D-lysine coated 6-well plates in neurobasal media (12349015, Gibco) supplemented with 1% GlutaMAX™ Supplement (35050-061, Thermofisher), 1% Sodium pyruvate (11360039, Gibco), 2% B27 supplement (17504044, Gibco) and 1% Penicillin–Streptomycin (03-031-1B, Sartorius). One half of the culture media was changed every 3 days until treatment. Differentiated cortical neurons were infected with indicated lentiviral vectors after 5 days in culture.

## Lentivirus production and infection

Third generation lentiviral vectors pLL3.7 (Addgene, #11795) that express shRNA under the mouse U6 promoter and CMV-EGFP or hSyn-EGFP reporter cassettes were obtained from the TAU Viral Core facility. Helper plasmids pMDLg/pRRE, pRSV-Rev, and pCMV-VSVG that carry HIV regulatory protein genes, as well as the pseudotyped envelope protein gene from vesicular stomatitis virus envelope G (VSV G), were obtained from the same facility.

Briefly, HEK-293FT packaging cells growing in 15 cm dishes were transfected with a mix of 7.8 µg helper vector pMDLg/pRRE, 3 µg helper vector pRSV-Rev, 4.2 µg envelope vector pCMV-VSVG, and 12 µg target vector pLL3.7-hSyn-PHOX2B-EGFP carrying the wild type or mutant PHOX2B gene, or CMV-mCherry UBA6. Polyethylenimine (PEI), (Merck 408727) was used as a transfection reagent. At 16 h of post-transfection, the culture media was removed and replaced with fresh high-serum medium, which was harvested 48 h later and filtered through Amicon Ultra-15 (UFC910024) vials at $1500 \times g$ for 30 min to obtain concentrated and purified lentiviruses for transduction. On the day of transduction, half the culture media was removed from primary cultured cortical neurons and was stored for later use (conditioned media), and then viral particles were added and incubated for 15 h. At the end of this time, the culture medium was replaced by a 1:1 mix of fresh and the conditioned medium that was collected previously. The neurons were then incubated for up to a week before being harvested for either western blot or immunostaining.

## Generation of patient-specific fibroblasts

Skin biopsies were collected from a 2-year-old female CCHS patient who carries a 20/27 Ala PHOX2B genotype as well as from her healthy sister and from the healthy father of two previously reported identical 4-year-old twin males with CCHS who carry 20/25 Ala stretches (Falik et al, 2020). The biopsies were dissected and cultured for two weeks in 6-well plates under a coverslip in DMEM (Sartorius 011701A) with 20% FBS and with half the media replaced every other day.

## Reprogramming of iPSCs

Fibroblast, $10^6$ cells were harvested using TrypLE Express (Gibco, 12604021) and electroporated with non-integrating episomal vectors using a Neon transfection system (Invitrogen, kit MPK10096). The cells were then plated on mouse embryonic fibroblast (MEF)-coated plates and cultured in DMEM with 15% FBS, 5 ng/ml basic fibroblast growth factor (bFGF, Peprotech 10018B) and 5 μM ROCK inhibitor (Enzo, ALX270333). After 2 days, the medium was replaced with NutriStem (BI) supplemented with 5 ng/ml bFGF, with fresh medium added every other day. On day 22, six colonies were transferred to new MEF-coated plates and cultured in NutriStem with 5 ng/ml bFGF. Three colonies were selected from each line and manually transferred to matrigel (Corning)-coated plates and cultured in NutriStem, which was replaced daily and with weekly passage.

## iPSC characterization

iPSCs were assessed for the expression of the pluripotency markers NANOG, SOX2, OCT3/4, TRA 1-60, and SSEA by immunocytochemistry and FACS analysis. The differentiation potential was assessed by harvesting the iPSCs at confluency using TrypLE and resuspending the cells in NutriStem supplemented with 10 ng/ml bFGF and 7 μM ROCK inhibitor (Enzo, ALX270333). Embryoid bodies (EBs) spontaneously formed after 2 days, at which time, the medium was replaced with EB medium (DMEM with 15% FBS, 1% Non-Essential Amino Acids and 0.1 mM β-mercaptoethanol, Gibco 31350010). After 4–7 days, the EBs were plated on 0.1% gelatin-coated plates and cultured for 21 days with EB medium replacement twice weekly. On day 21, the cells were fixed and stained for heavy chain neurofilament, α-SMA, or α-fetoprotein as ectodermal, mesodermal and endodermal markers, respectively. G-banding karyotype analysis was used to exclude any chromosomal abnormalities that may have occurred during the reprogramming process. Briefly, iPSCs were supplemented with 100 ng/ml colcemid (Sartorius 120041D), incubated for 60 min, and harvested in Versene solution (Gibco 15040033). Cells were fixed in 1:3 glacial acetic acid:methanol (Biolabs-chemicals) solution and the G-banding karyotype was determined. Finally, all lines were tested for mycoplasma contamination using the Hy-mycoplasma PCR kit (Hylabs, KI5034I).

## Differentiation of iPSCs into autonomic neurons

The protocol was performed as previously described with modifications (Kirino et al, 2018). iPSCs were cultured in Nutristem (Sartorius, 05-100-1A) to a confluence of 1 million cells/well. At day 0, iPSCs

washed with DPBS, and a single cell suspension was prepared using Versene solution for 2 min (Gibco, 15040033), and then cells were transferred to T25 flasks coated with Poly-Hema (2-hydroxyethyl methacrylate) at a concentration of 350,000 cells/ml in neuromeso-dermal progenitor cell induction (NMP) medium containing Essential 6 medium (Gibco, A1516401), 1.5 mM CHIR (Tocris biotech, 4423), 10 μM SB (Tocris biotech, 431542), Penicillin-Streptomycin-Amphotericin B solution (PSA, Sartorius, 03-033-1B), supplemented with 10 μM ROCK inhibitor (ROCKi Enzo, ALX270333), where they formed aggregates. On the following day, half of the medium was replaced with NMP without ROCKi. At day 3, the medium was replaced with sympathetic (NCi) neural crest induction medium containing Essential 6 medium, 1.5 mM CHIR, 20 ng/ml bFGF (Peprotech 10018B), 50 ng/ml BMP4 (Prospec, Cyt-1093), 100 nM all trans retinoic acid (RA, Merck, R2625), and PSA. On day 10, the culture was dissociated into single cells using Accutase (Gibco, A1110501) for 4 min at 37 °C, and cultured in sympathetic neuroblast induction and propagation (NCC) medium, containing neurobasal medium (Gibco, 21103049), 20 ng/ml bFGF, 50 ng/ml BMP4, 20 ng/ml EGF, 2 μg/ml heparin, B27 (Gibco, 17504044), N2 (Gibco, 17502048), GlutaMAX (Gibco, 35050038), PSA and 10 μM ROCKi. On day 11, half of the NCC medium was replaced with NCC medium without ROCKi and half of the medium was replaced every other day. On day 17, medium was replaced with sympathetic neuronal maturation medium (NMM medium) containing neurobasal medium, B27, N2, 10 ng/ml GDNF (Peprotech, 45010), 10 ng/ml BDNF (Peprotech 45002), 10 ng/ml NGF (R & D 256-GF), GlutaMAX, and PSA. One third of the medium was replaced every other day until day 31.

## Immunocytochemistry for iPSC-derived autonomic neurons

In order to prepare for the culture, coverslips were incubated in poly-L-ornithine solution (Merck, P3655) overnight at 37 °C followed by 3 washes with cell culture grade water and drying for 15 min. On seeding day, the coverslips were incubated in laminin (Merck, L2020) d for 1 h at 37 °C before being washed with PBS. The neurospheres were harvested manually and seeded on the coverslips. After 4–5 days, the cells on the coverslips were washed with Dulbecco's Phosphate-Buffered Saline (DPBS, Sartorius, 020231A) and fixed in 4% paraformaldehyde at room temperature for 15 min. The cells then were washed twice in DPBS, and blocked with DPBS containing 0.1% Triton X-100 (Merck, T8532) and 1% bovine serum albumin (blocking solution, Merck, A7906100G), for 1 h at RT. Primary antibodies were added to the blocking solution and incubated overnight at 4°C. Following three washes with blocking solution, the cells were incubated with fluorescent secondary antibodies for 2 h at room temperature. DRAQ5 was used to stain the cell nuclei.

## Flow cytometry

iPSCs were harvested and dissociated into single cells by incubation with TrypLE for 2 min at 37 °C. For the detection of intracellular markers, samples were incubated in fixation solution (Invitrogen, 00522356, 00512343) for 40 min at room temperature followed by washings with permeabilization solution (Invitrogen, 00833356). For surface markers, the cells were washed once with 3% FBS in

DPBS. In both cases, the samples were incubated with the appropriate primary antibodies for 2 h at room temperature, then were washed twice and incubated for 1 h with the relevant secondary antibody followed by three more washes. Analysis was performed using a NovoCyte flow cytometer (ACEA). The first gating was SSC-H/FSC-H, and the entire cell population was selected (without cell debris) followed by FSC-H/FSC-A gating and single cell selection. The final analysis is presented as counts (%)/FITC-H.

## Analysis of cell death markers in CCHS neurons

On day 31 of differentiation, the neurons were washed twice and detached using Accutase for 4 min at 37 °C. Cells were gently mixed every minute during incubation time. The cells were then washed and were seeded $1–2 \times 10^6$ on coverslips with NMM medium and 10 μM ROCKi. One day post seeding, 0.5 ml of fresh NMM medium was added. On the next day, the neurons were infected with lentivirus expressing CMV mCherry-UBA6 overnight or kept in normal NMM medium. The medium containing the virus was discarded and replaced by conditioned and fresh NMM. Cells were cultured for an additional week and were analyzed for cell death markers.

For TUNEL assay detecting DNA fragmentation, the cells were fixed in 4% PFA for 20 min and then washed twice. TUNEL Assay Kit (Abcam, ab66108) was used according to the manufacturer's instruction. The cells were washed twice with wash buffer followed by adding the DNA labeling solution for 1 h at 37 °C without agitation. The DNA labeling solution was removed, and the cells were washed twice with Rinse buffer followed by adding Propidium Iodide/RNase A solution for 30 min. The Propidium Iodide/RNase A solution was removed and the cells were washed twice with blocking solution and blocked for 1 h at room temperature. The blocking solution was removed and primary anti-PHOX2B antibody was added overnight for immunostaining.

For detecting cell surface exposure of phosphoatidylserine, the Annexin V-FITC apoptosis detection kit (Merck, CBA059) was used according to the manufacturer's protocol. Annexin V-FITC solution containing 1:100 Annexin: calcium buffer was prepared and was added to the neurons for 10 min at room temperature without agitation. Then, the cells were washed with calcium buffer and were fixed using 4% PFA for 20 min, washed with a blocking solution and blocked for 1 h at room temperature. The blocking solution was removed and primary anti-β3-Tubulin antibody was added overnight for immunostaining.

## Generation of OPMD patient-derived cells

The tissue biopsy was obtained from an OPMD patient undergoing cricopharyngeal myotomy (heterozygous polyalanine expansion mutation resulted in +3 Ala in PABPN1). The tissue was submerged by incubation with collagenase II solution (Merck, C0130). After the incubation, the tissue was transferred using a 5% BSA coated pipette tip into a 5% BSA pre-coated 10 cm plate filled with DMEM supplemented with 2.5% pen-strep-Nystatin (PSN). The tissue was further incubated for 30 min, and the muscle was then repeatedly pipetted to dissociate the myofibers. Using a fire-polished Pasteur pipette coated with 5% BSA, all visible myofibers

were transferred to a 6-well plate coated with 5% BSA and filled with DMEM 2.5% PSN and then to another well coated with Matrigel (Corning, 354234) and filled with Bioamf 1% (Sartorius, 01-194-1 A) PSN. The connective tissue was transferred to another well coated with Matrigel and filled with bioamf1% to extract fibroblasts.

## Western blotting assay

Cells were washed with PBS and harvested in Laemmli buffer containing 5% beta-mercaptoethanol. For the ubiquitin loading assays, the cells were lysed on ice in lysis buffer (20 mM Tris-HCl, pH 6.8, 137 mM NaCl, 1 mM EGTA, 1% Triton ×100, 10% glycerol, and a protease inhibitors cocktail), centrifuged to discard the cell pellet and then the supernatant was added to Laemmli buffer at a ratio of 1:1 without using beta-mercaptoethanol. Protein samples were boiled for 5 min at 95 °C, separated by SDS–PAGE, transferred onto PVDF membranes, subjected to western blot analysis, and visualized using the ECL enhanced chemiluminescence reagent (CYANAGEN). Protein levels in each sample were evaluated by normalization to the housekeeping β-actin. The bands were quantified using ImageJ software.

## Immunoprecipitation and ubiquitination assays

Cells were lysed on ice in immunoprecipitation (IP) buffer (20 mM Tris-HCl, pH 7.2, 150 mM NaCl, 2 mM MgCl$_2$, 0.5% NP-40), supplemented with a protease inhibitors cocktail before use. In IP experiments performed in separated cell fractions, cells were lysed in IP buffer containing 0.1% NP40. Supernatant was kept as a cytoplasmic fraction and the nuclear pellet was dispersed with IP buffer and passed through 25 and 27 gauge needle then sonicated to ensure extraction of nuclear proteins. For polyubiquitination experiments, cells were treated with a proteasome inhibitor MG132 (10 μM) during the last 6 h before lysis with the IP buffer supplemented with 1 mM PMSF and 10 mM iodoacetamide.

Whole-cell lysates obtained by centrifugation were incubated with 2–5 μg of antibody overnight at 4 °C, followed by 2 h incubation with Protein A-Sepharose CL-4B (Cytiva, 17-0780-01). The immunocomplexes were then washed three times with IP buffer, and boiled at 95 °C for 5 min in Laemmli sample buffer containing 5% beta-mercaptoethanol, before being separated by SDS–PAGE for western blotting assays. For experiments analyzing endogenous UBA6-USE1 interaction in USE1 ΔPolyAla KO cells, a cross-linking step was performed with formaldehyde prior to cell lysis and IP. For experiments to examine the binding of isolated polyalanine stretches, a pre-clearing step was performed by incubating the whole cell lysates with 25 μl beads for 2 h at 4 °C. The beads were then discarded, and the cell lysates were incubated with antibody overnight as already described. For experiments analyzing ubiquitin load on USE1 under expression of polyalanine disease proteins, the different polyalanine constructs were expressed in HEK293T cells for 72 h while the FLAG-USE1 was expressed in the cells for the last 24 h.

## Protein expression, purification, and labeling

All proteins used in this study were overexpressed in Rosetta (DE3, pLysS) *Escherichia coli* (Merck) cells using 0.4 mM

isopropyl 1-thio-D-galactopyranoside (Inalco Pharmaceuticals) induction overnight at 18 °C. Purification of proteins was performed on Ni Sepharose 6 Fast Flow sepharose (Cytiva) in 50 mM HEPES (pH 7.5), 300 mM NaCl, 1 mM TCEP, 10 mM imidazole and 10% (w/v) glycerol and eluted using 250 mM imidazole (pH 7.5) in the same buffer. UBA6 and E2 variants were further desalted using PD-10 Columns (Cytiva) into Loading buffer containing 20 mM HEPES (pH 7.5), 150 mM NaCl, and 10% (w/v) glycerol. Concentrated protein aliquots were stored at 80 °C. All protein concentrations indicated correspond to total protein and are based on UV absorbance at 280 nm.

Fluorescein-5-Maleimide (AnaSpec) was attached to ubiquitin following the directions as previously described (Keren-Kaplan et al, 2012). Briefly, proteins in 20 mM Tris (pH 7.5), 150 mM NaCl and 1 mM TCEP were incubated for 2 h in the presence of the fluorophore at room temperature such that the label:protein ratio would be 4. To quench the reaction, beta-mercaptoethanol was added at a ratio of 10:1 to fluorescein. Fluorescein-labeled ubiquitin was then separated from free dye on PD10 desalting columns (Cytiva) and was stored at 80 °C.

## E1 and E2 loading assays

All loading assays were performed at 32 °C in a buffer containing 20 mM HEPES (pH 7.5), 150 mM NaCl and 10% (w/v) glycerol. E1 and E2 loading assays were performed using 3 μM fluorescein ubiquitin, a range of 10 nM to 1 μM E1, 1 μM E2, and 2.5 mM concentrations each of ATP and $MgCl_2$. Reactions were stopped using non-reducing SDS-PAGE loading buffer. Samples were separated on 4–20% Tris-Glycine NuPAGE gels (Thermo) in Tris-Glycine buffer. Band detection in E1 and E2 loading assays was performed using an Alliance Q9 imager (Uvitec Cambridge). The different qualitative end-point assays were performed using freshly thawed protein aliquots, and the results obtained were reproducible across at least three different protein batches. All constructs and conditions were carried out in triplicate. For in vitro experiments of E6AP ubiquitination (resolved under reducing conditions), the HA-E6AP isoform II (a gift from Peter Howley (Kao et al, 2000), Addgene plasmid #8658) was purified from HEK293T cells using HA-beads (900801, Biolegend), and IP buffer containing 0.5% NP40. The HA peptide (1 mg/ml) was used for elution in elusion buffer (50 mM Tris-HCL PH 7.2, 50 mM NaCl, 1 mM EDTA) (931401, Biolegend).

## In silico structural analysis

We constructed models for UBA6 using the Protein Homology/ analogY Recognition Engine V 2.0 (PHYRE2) and the AlphaFold server (Jumper et al, 2021; Kelley et al, 2015). Further idealization of the geometry was achieved by five cycles of minimization with Refmac5. The model of full-length USE1 was downloaded from the AlphaFold server (Jumper et al, 2021; Kelley et al, 2015). Structure visualization and figures preparations were performed with PyMOL (Molecular Graphics System, http://www.pymol.org). We employed the continuum solvation method APBS (Adaptive Poisson–Boltzmann Solver) with the CHARMM force field to calculate the electro-potential surface of UBA1 and UBA6 (Baker et al, 2001). Available protein structures were from PDB code 4II2, 6DC6, 1Y8Q, 2NVU, and 7PYV.

## Biophysical interaction studies with polyalanine peptides

Microscale thermophoresis (MST) analyses of the SCCH domain:polyalanine peptide interaction were carried out using our Monolith NT.115 (Nano Temper Technologies). A peptide containing a tandem of 7 alanine residues was labeled with Cy5 (GL Biochem Shanghai Ltd.), and was used at constant final concentration of 100 nM. The labeled peptide was mixed with the ligand, a purified recombinant unlabeled UBA6 (623-889) or UBA1 (624-891) SCCH domains. 1:2 serial dilutions of the ligands from 200 μM to 1.56 μM final concentration were assayed. After a short incubation, the samples were loaded into premium glass capillaries before data collection. All MSTs were performed twice at Excitation Power 30% of the LED and MST power of 20% and 40%.

## Microscopy

The cells were grown on coverslips, and then washed and fixed in 4% Paraformaldehyde for 10–15 min before being permeabilized with 0.1% Triton X-100. A solution of, 1% BSA in PBS was used to block both primary and secondary antibodies. The primary antibody was added at a ratio of 1:100 and incubated for at least 1 h at room temperature while the secondary antibody (1:300 Invitrogen) was allowed to incubate with the sample for 30 min at room temperature. Neurons were permeabilized in 2% BSA + 0.1% Triton, and blocked with 2% BSA and the primary antibodies were incubated overnight at 4 °C, at a ratio of 1:150, with the secondary antibody incubated for 2 h at room temperature at 1:500. A Zeiss 710 confocal microscope was utilized for confocal imaging with a ×63 oil-immersion lens. Nuclear staining was detected by staining with DRAQ5. For quantification, the operator was blinded to the outcome of the experiment when selecting suitably similar fields to image for subsequent computerized analysis.

For the colocalization experiments, the association of PHOX2B and UBA6 outside the nucleus was measured by selecting PHOX2B positive cells manually using Fiji, and excluding the nuclei by segmenting and removing the nuclear channel. The colocalization between the channels of interest was then measured using the JACops plugin in Fiji with the default parameters, and Pearson's correlation coefficient was calculated. The cytoplasmic intensity was measured for the channel of interest and the mean gray value was recorded after excluding the nuclei as already described. For the localization experiments related to UBA6 in neurons, the association of UBA6 with the cell body was examined by selecting the neuronal cell bodies manually using Fiji (Schindelin et al, 2012). For the localization of UBA6 with neurites, the cell bodies were selected manually and removed. Arc intensity in the neurons was measured by circulating the neuronal cell body together with the first neurite junction manually using Fiji. The mean integrated value and the area for each neuron were recorded. The value of the mean divided by the area was used for statistical analysis.

For UBA6 intensity measurements in aggregated vs. non-aggregated forms of mutant GFP-PHOX2B in primary cortical neurons, cell bodies were manually selected, and clusters were detected in the GFP signal with a minimum diameter of 0.25 μm². The "Red" intensity (UBA6) was then measured in the area of the clusters (cluster intensity). For cytoplasmic intensity, the "red" intensity was measured in the cell body area excluding the clusters

and the nucleus (segmented from the nuclei signal). The intensity was recorded as integrated density.

FLIM imaging was performed using two-photon FLIM microscopy. The cells were immunostained by mouse anti-USE1 and rabbit anti-UBA6 primary antibodies, with secondary antibodies anti-mouse Alexa Fluor 488 and anti-rabbit Alexa Fluor 555, respectively. The donor antibody (Alexa Fluor 488) was excited with a Ti-sapphire laser (Chameleon, Coherent) at a wavelength of 920 nm and a power of 1.0–2.0 mW. The images were acquired by Bergamo two-photon microscope (Thorlabs) equipped with a Time-Correlated Single Photon Counting board (Time Harp 260, Picoquant), thorough a ×18 0.8 na objective (Nikon). Images were acquired at $128 \times 128$ or $256 \times 256$ pixel size, and averaged across 24 frames. For FLIM analysis, we calculated the mean lifetime of multiple ROIs at each image using double exponent fitting. For each group, we determined the lifetime of donor only samples (prepared with staining for USE1 only) and compared them with the mean lifetime of cells stained for donor and acceptor (USE1 + UBA6). Then, we subtracted donor-(donor/acceptor) lifetime, to compare the change in lifetime between groups. The analysis was performed using a custom software written with C+ (FLIMage, Florida Lifetime Imaging).

## mRNA analysis by qRT-PCR

RNA purification was performed using the total RNA purification Micro kit (Norgen, 35300) according to manufacturer's protocol. The concentration and quality of the RNA were measured by NanoDrop one (Thermo Fisher). The cDNA generation was done by using a High capacity cDNA reverse transcription kit (Applied Biosystem, 4368814) with RNase inhibitor (Applied Biosystem, N8080119) as described in the kit protocol. Then, the cDNA was diluted in ultra-pure water in a ratio of 1:5 (each biological replicate was assessed in triplicates). Reaction solutions were prepared for each set of primers including the genes E6AP/UBE3A and two different sets of primers for β-Actin. The reaction volume contained Fast SYBER green master mix (Applied Biosystem, 43856120), forward and reverse primers, ultra-pure water, and the cDNA template. To calculate relative RNA expression, the mean of two sets of primers for β-Actin was included as an endogenous control. The data were analyzed using the ΔΔCT method. The sequence of the primers used:

| Gene name | Forward | Reverse |
|---|---|---|
| E6AP/UBE3A | AACTACAGAATATGAC GGTGGC | TGTCTGTGCCCGTTGTAAAC |
| β-Actin set 1 | ACCCAGCACAATGAAG ATCAA | ACATCTGCTGGAAGGTGGAC |
| β-Actin set 2 | GAGCACAGAGCCT CGCCTTT | ACATGCCGGAGCCGTTGTC |

## Analysis of detergent-insoluble aggregates in cells

For extraction of soluble protein fraction, cells were lysed with buffer 1 (25 mM Tris pH 7.4, 150 mM NaCl, 1 mM EDTA, 1 mM EGTA, supplemented with Protease Inhibitor Cocktail, 1 mM PMSF, phosphatase inhibitor cocktail II & III) and ultracentrifuged for 15 min. The supernatant was kept as the soluble fraction and the pellet was re-homogenize in high salt/sucrose buffer (10 mM Tris pH7.4, 0.8 M NaCl, 10% Sucrose, 1 mM EGTA supplemented with 1 mM PMSF), and ultracentrifuge for 15 min. The supernatant was adjusted to 1% sarkosyl and incubated for 1 h at 37 °C on orbital shaker followed by ultracentrifugation for 1 h at 4 °C. The pellet (sarkosyl-insoluble fraction) was resuspended 50 µl TBSX1 for further analysis.

## Bioinformatics analysis

USE1/UBE2Z human homologs were searched against the Uniprot (Pubmed id 29425356) and NCBI databases using BLAST (Pubmed id 2231712). Prosite (Pubmed id 23161676) was used to scan for alanine residues motifs with between 6 and 10 continuous alanine residues in the BLAST results (a search for proteins containing polyalanine stretches in the ubiquitin cascades). Alignments of the E2 family in vertebrates and across all databases were calculated using MAFFT (Pubmed id 28968734). The figures were generated using Jalview (Pubmed id 19151095).

## Statistics

Basic data handling was performed in Microsoft Excel. For single comparisons, the statistical significance of the difference between experimental groups was determined using two-tailed Student's $t$ test with the Prism GraphPad software v.9. Comparisons of multiple means were made by one-way or two-way analysis of variance (ANOVA) followed by the Tukey's, Dunnett's and Sidak's post hoc tests to determine statistical significance. Differences were considered statistically significant for $P < 0.05$. Sample sizes were chosen based on extensive experience with the assays we have performed. For primary neurons transduced with lentiviral vectors, we used independent cultures prepared from brains of mouse embryos taken from different females. For iPSC-derived neurons, three independent cultures from different differentiation days were considered for analysis, and for cell-line-based experiments, we considered replicates performed in different days. Errors bars shown in the figures are standard errors of the mean (s.e.m.).

# Data availability

This study includes no data deposited in external repositories.

# Peer review information

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

## Acknowledgements

We are grateful for funding from Azrieli Foundation (grant to AA), CCHS Foundation/CCHS Network (grant to AA and GDV), Israel Science Foundation 923/22 (grant to GDV), Israel Science Foundation 1440/21 (grant to GP), Acceleration Grant Program of the Israel Cancer Research Fund 940283 (grant to GP), and Yoran Institute for Human Genome Research (scholarship to FA-S). We thank Yad Laneshima for assistance in patient recruitment; A Yeheskel and H Benyamini for bioinformatics analysis; M Groysman for assistance in viral vector production; H Grobe for assistance in image analysis; D Fornasari, DC Rubinsztein, A Aichem, M Groettrup, Y Tokita, and D Hnisz for contributing reagents.

## Author contributions

**Fatima Amer-Sarsour**: Conceptualization; Data curation; Formal analysis; Writing—original draft; Writing—review and editing. **Daniel Falik**: Conceptualization; Data curation; Formal analysis; Writing—original draft; Writing—review and editing. **Yevgeny Berdichevsky**: Conceptualization; Data curation; Formal analysis; Writing—original draft; Writing—review and editing. **Alina Kordonsky**: Data curation; Formal analysis. **Sharbel Eid**: Data curation; Formal analysis. **Tatiana Rabinski**: Data curation; Formal analysis. **Hasan Ishtayeh**: Data curation; Formal analysis. **Stav Cohen-Adiv**: Data curation; Formal analysis. **Itzhak Braverman**: Data curation. **Sergiu C Blumen**: Data curation. **Tal Laviv**: Data curation; Formal analysis. **Gali Prag**: Conceptualization; Formal analysis; Writing—original draft; Writing—review and editing. **Gad D Vatine**: Conceptualization; Formal analysis; Supervision; Funding acquisition; Writing—original draft; Writing—review and editing. **Avraham Ashkenazi**: Conceptualization; Formal analysis; Supervision; Funding acquisition; Writing—original draft; Writing—review and editing.

## Disclosure and competing interests statement

The authors declare no competing interests.

# Expanded View Figures

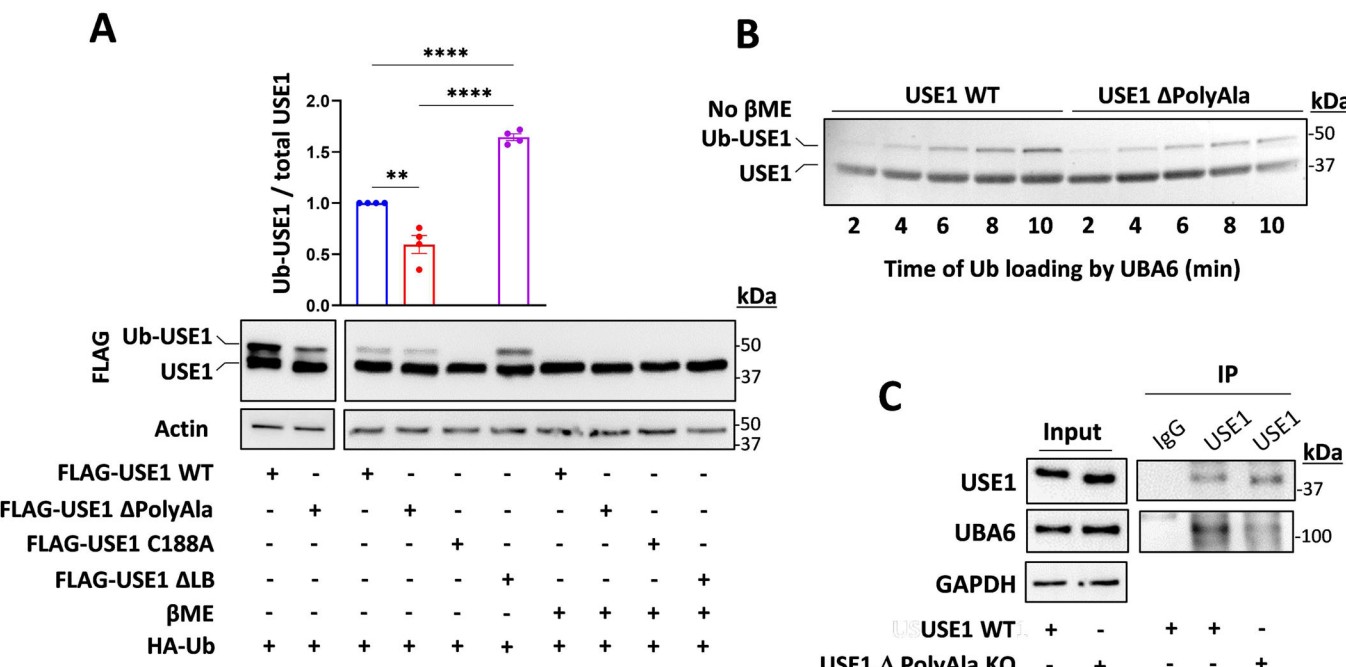

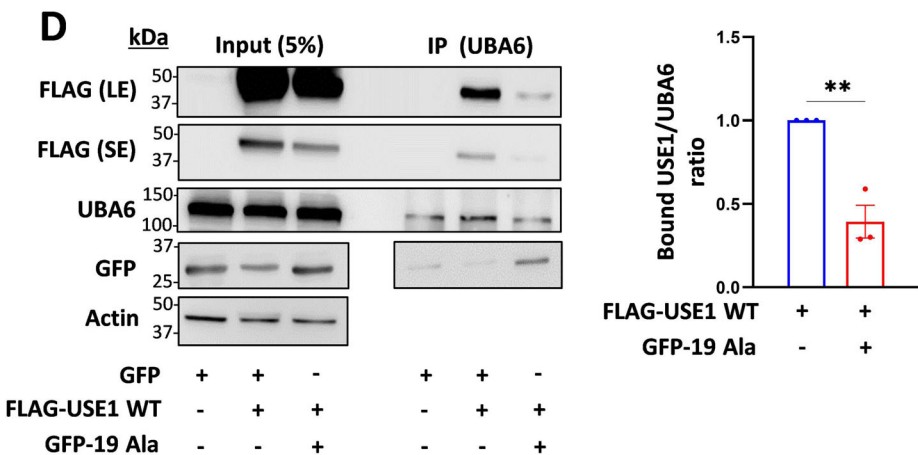

**Figure EV1. Polyalanine stretches regulate UBA6-USE1 interaction and ubiquitin transfer.**

(A) FLAG-WT USE1, FLAG-USE1 ΔPolyAla, FLAG-USE1 C188A, and FLAG-USE1 ΔLB were co-expressed with HA-Ub in HEK293T cells. Cell lysates were incubated with or without β mercaptoethanol (βME) and analyzed for ubiquitin loading. Results are normalized to control WT USE1. $n = 4$ biological replicates. (B) A representative blot for time-dependent in vitro ubiquitin loading of WT and ΔPolyAla USE1 by UBA6 (quantification is presented in Fig. 1C). (C) WT and ΔPolyAla KO cells were treated with the cross-linker formaldehyde and cell lysates were immunoprecipitated with anti-USE1 or control IgG antibodies. Immunocomplexes were analyzed with anti-USE1 and anti-UBA6 antibodies. (D) FLAG-WT USE1, empty GFP and GFP-polyAla (19Ala) constructs were transfected into HEK293T cells. Cell lysates were immunoprecipitated with anti-UBA6 antibodies and the immunocomplexes were analyzed with anti-FLAG, anti-UBA6 and anti-GFP antibodies. The bound USE1/UBA6 ratio is shown. $n = 3$ biological replicates. Data information: Data points in (A, D) represent mean ± s.e.m. *P* values were calculated by one-way ANOVA Tukey's test (A) or paired 2-tailed *t* test (D). **$P < 0.01$, ****$P < 0.0001$. Source data are available online for this figure.

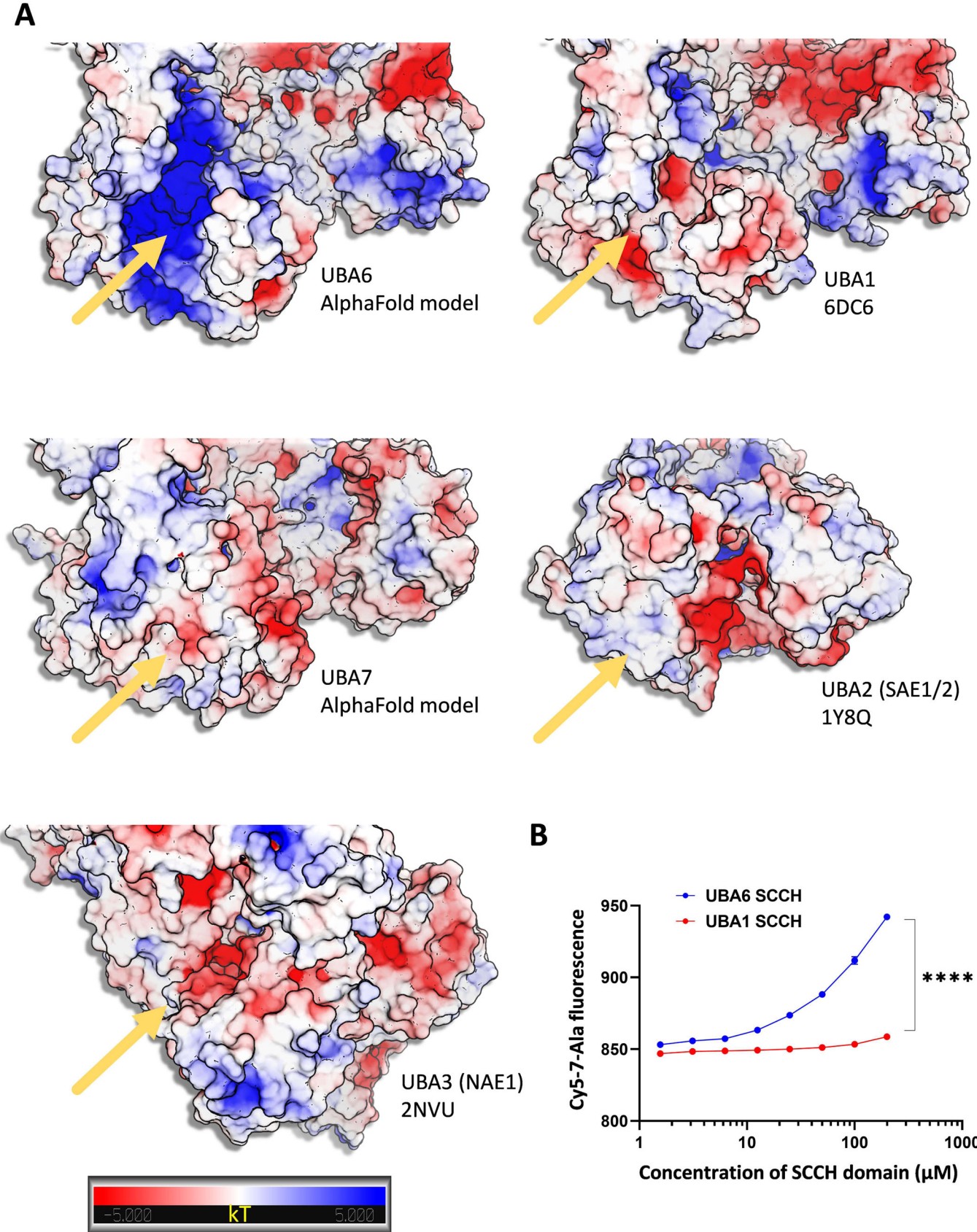

◀

**Figure EV2.** **Biophysical analysis of the interaction between a polyalanine peptide and the SCCH domains of the canonical E1 ubiquitin-like activating enzymes.**

(A) AlphaFold models of UBA6 and UBA7 and the crystal structures of UBA1, UBA2 and UBA3 are shown. The structures were aligned and electrostatic potential was calculated as described in the methods. The yellow arrows indicate the location of the groove within the SCCH domains. UBA6, UBA1 and UBA7 form an extended lobe within the SCCH, which is missing in UBA2 and UBA3. The groove in UBA7 is covered and do not exist in UBA2 and UBA3. The grooves in UBA1 and UBA6 are highly similar in terms of structure but present significantly different electro potential surfaces. The gradient from negative (red) to positive (blue) charge is shown. The figure was prepared by PyMol. (B) Microscale thermophoresis interaction analysis of cy5-7Ala-peptide against UBA6 or UBA1 SCCH domain. The dose-response curve of cy5-7Ala-peptide titrated against increasing concentrations of the SCCH domain is presented. Results are mean ± s.e.m. $n = 4$ biological replicates, Two-way ANOVA, Sidak's test. ****$P < 0.0001$.

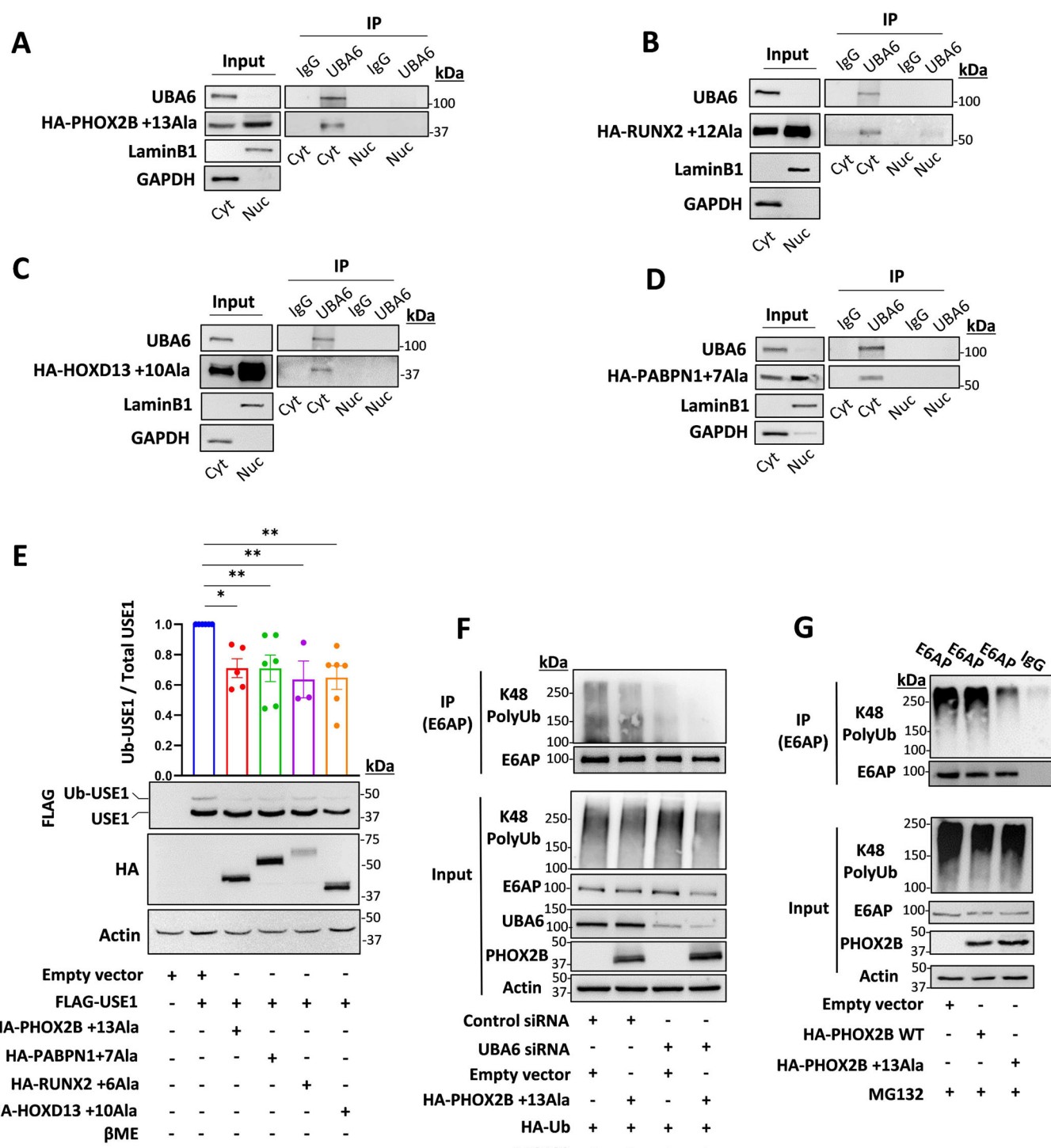

◀ **Figure EV3.  Cytoplasmic polyalanine-expanded disease proteins interact with UBA6 and decrease USE1 ubiquitin loading and E6AP polyubiquitination.**

(A–D) HEK293T cells were transfected with the indicated constructs: (**A**) HA-mutant PHOX2B (+13Ala). (**B**) HA-mutant RUNX2 (+12Ala). (**C**) HA-mutant HOXD13 (+10Ala). (**D**) HA-mutant PABPN1 (+7 Ala). Endogenous UBA6 was immunoprecipitated from the nuclear fraction (Nuc, LaminB1 enriched) or the cytoplasmic (Cyt, GAPDH enriched) fraction (unrelated IgG was used as a control). The immunocomplexes were analyzed with anti-HA antibodies. (**E**) HEK293T cells were transfected with constructs expressing different polyalanine-expanded disease proteins (mutant PHOX2B, mutant RUNX2, mutant HOXD13, and mutant PABPN1) together with FLAG-USE1. Cell lysates were incubated without β mercaptoethanol and analyzed for ubiquitin loading. Results are mean ± s.e.m. normalized to control (empty vector, no disease protein). Paired 2-tailed t test. $n = 3$–6 biological replicates. (**F**) Control and UBA6-depleted HEK293T cells were transfected with HA-Ub, mutant PHOX2B or empty vector and incubated for the last 6 h with the proteasome inhibitor MG132 (10 μM). Endogenous E6AP was immunoprecipitated from cell lysates for ubiquitination analysis. (**G**) HEK293T cells were transfected with WT PHOX2B, mutant PHOX2B or empty vector and incubated for the last 6 h with the proteasome inhibitor MG132 (10 μM). Endogenous E6AP was immunoprecipitated from cell lysates for ubiquitination analysis (unrelated IgG was used as a control). ns non-significant, $*P < 0.05$, $**P < 0.01$.

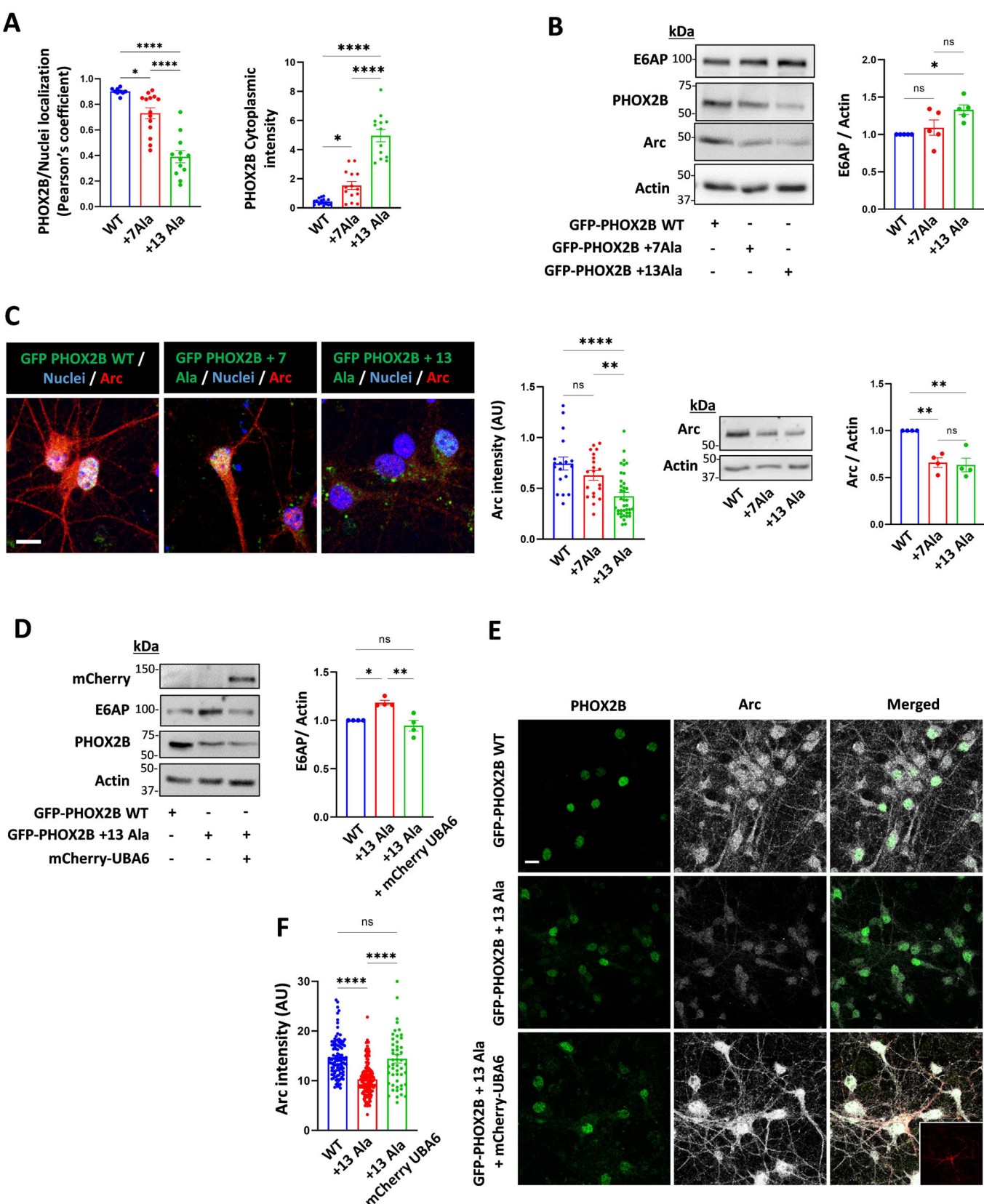

◀   **Figure EV4.   UBA6 overexpression affects E6AP and Arc levels in mutant PHOX2B transduced neurons.**

(A–C) Mouse primary cortical neurons were transduced with lentiviral vectors expressing GFP-tagged WT PHOX2B or mutant PHOX2B (+7Ala or +13Ala). (A) Quantification of the association of GFP-PHOX2B with the nucleus (Pearson's coefficient) is presented as well as GFP-PHOX2B cytoplasmic intensity, related to Fig. 4C. (B) Analysis of E6AP and Arc levels in the WT and mutant PHOX2B-expressing neurons ($n = 5$ and $n = 4$ biological replicates, respectively). (C) Quantification of Arc intensity in the GFP-PHOX2B expressing neurons is presented (image scale bar 10 μm) as well as a blot showing Arc protein levels. Results represent the average values from neurons in different imaged fields. $n = 20$, $n = 30$ and $n = 100$ neurons analyzed for WT PHOX2B, mutant PHOX2B +7Ala and +13Ala, respectively. (D–F) The neurons were transduced with lentiviral vectors expressing GFP-WT PHOX2B, GFP-mutant PHOX2B (+13Ala) with or without lentiviruses encoding for mCherry-UBA6. (D), E6AP levels were analyzed in cell lysates. Results are normalized to WT PHOX2B. $n = 4$ biological replicates. (E, F) Quantification of Arc intensity in cycloheximide-treated GFP-PHOX2B expressing neurons that were positive or negative to mCherry. $n = 110$, $n = 150$ and $n = 50$ neuronal cell bodies analyzed for WT PHOX2B, mutant PHOX2B and mutant PHOX2B + UBA6, respectively. Inset shows mCherry signal. Scale bar 10 μm. AU arbitrary units. Data information: Data points in (A–D, F) represent mean ± s.e.m. $P$ values were calculated by one-way ANOVA Tukey's test (A–D, F). *$P < 0.05$, **$P < 0.01$, ****$P < 0.0001$, ns non-significant.

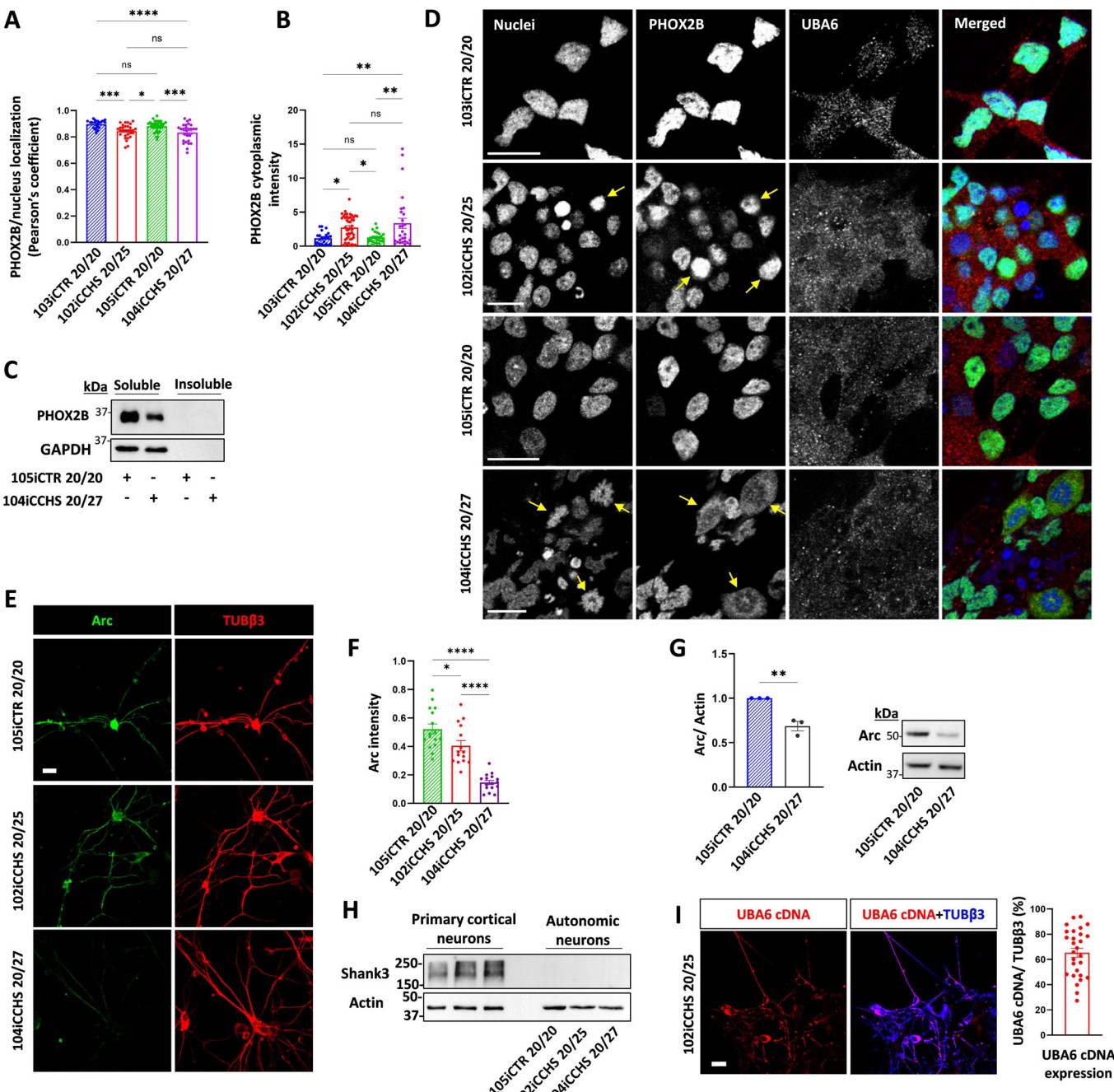

**Figure EV5.  Patient-derived autonomic neurons showing cytoplasmic mislocalized soluble PHOX2B, and decreased Arc levels.**

(A, B) Quantification of the association of endogenous PHOX2B with the nucleus (Pearson's coefficient) in autonomic neurons from control and CCHS patients. Quantification is shown also for PHOX2B cytoplasmic intensity. Total number of neurons analyzed was 160 for 103iCTR 20/20, 350 for 102iCCHS 20/25, 380 for 105iCTR 20/20 and 550 for 104iCCHS 20/27. The results represent additional analysis from the same neurons analyzed in Fig. 5B. (C) Autonomic neurons from control and CCHS patients were analyzed for the levels of PHOX2B in the soluble and sarkosyl-insoluble fractions. (D) iPSC-derived human autonomic neurons from control and CCHS patients were labeled by nuclear staining (colored blue, abnormal nuclear morphology marked with arrows), and for endogenous PHOX2B (colored green) and endogenous UBA6 (colored red). Images indicate events of severe cytoplasmic mislocalization of PHOX2B (marked with arrows). Scale bar 10 μm. (E, F) Immunostaining of Arc (colored green) and TUBβ3 (colored red) in autonomic neurons from control and CCHS patients. Scale bar 20μm. For quantification, Arc intensity was normalized to TUBβ3 in different image fields. Number of neurons analyzed 105iCTR 20/20 $n = 105$, 102iCCHS 20/25 $n = 220$, 104iCCHS 20/27 $n = 166$. (G) Analysis of Arc levels in the autonomic neurons from control and CCHS patient-derived neurons. Results are normalized to control line. $n = 3$ biological replicates. (H) Analysis of shank3 levels in mouse primary cortical neurons and in autonomic neurons from control and CCHS patients. (I) Quantification of the abundance of UBA6 cDNA in CCHS patient-derived neurons transduced with mCherry tagged-UBA6 lentiviral vectors. Representative images are presented for mCherry (colored red) and TUBβ3 staining (colored blue) in the patient neurons (scale bar 20 μm). For quantification, the percentage of mCherry coverage from the TUBβ3 staining was calculated in different image fields ($n = 80$ neurons). Data information: Data points in (A, B, F, G, I) represent mean ± s.e.m. $P$ values were calculated by one-way ANOVA Tukey's test (A, B, F) or paired 2-tailed $t$ test (G). *$P < 0.05$, **$P < 0.01$, ***$P < 0.001$ ****$P < 0.0001$, ns non-significant.

