## [Peer Review File · The EMBO Journal]

Disease-associated polyalanine expansion mutations impair UBA6-dependent ubiquitination

Avraham Ashkenazi, Fatima Amer-Sarsour, Daniel Falik, Yevgeny Berdichevsky, Alina Kordonsky, Sharbel Eid, Tatiana Rabinski, Hasan Ishtayeh, Stav Cohen-Adiv, Itzhak Braverman, Sergiu Blumen, Tal Laviv, Gali Prag, and Gad Vatine
DOI: 10.15252/emj.2023114934

Corresponding author(s): Avraham Ashkenazi (ashkenaziavi@tauex.tau.ac.il) , Gad Vatine (vatineg@bgu.ac.il)

Review Timeline:

Submission Date:	6th Jul 23
Editorial Decision:	14th Aug 23
Revision Received:	29th Oct 23
Editorial Decision:	22nd Nov 23
Revision Received:	23rd Nov 23
Accepted:	1st Dec 23

Editor: Ioannis Papaioannou

Transaction Report:

(Note: With the exception of the correction of typographical or spelling errors that could be a source of ambiguity, letters and reports are not edited. Referee reports are anonymous unless the Referee chooses to sign their reports.)

Please note that this manuscript was transferred from another journal where it was originally reviewed. Since the original reviews are not subject to EMBO's transparent review process policy, they cannot be published.

Response to reviewers (Amer-Sarsour et al.)

Referee #1

Comment:

Poly-Ala repeat expansions in a variety of proteins, including transcription factors, have been seen to cause a series of developmental diseases. How expansion of a poly-Ala repeat causes aberrant protein behavior and disease is not fully understood. Recent work had suggested that poly-Ala repeats in transcription factors lead to an “unblending” of transcription factor condensates and thereby alters the output of gene expression networks (Basu et al., Cell 2020; this paper is surprisingly not cited by the authors). Whether other mechanisms contribute to disease is not known.

Answer:

We thank the reviewer for this comment. We did not intend to skip this important work by Basu et al. In fact, we pointed in the Discussion the literature supporting the notion that there should be additional mechanisms contributing to polyalanine-expansion diseases besides dysregulation of transcriptional programs. The search for such mechanisms was the focus of our work. We have now revised this to make it clearer and included the Basu et al., Cell 2020 paper in the Discussion.

Comment:

In this manuscript, Amer-Sarsour and colleagues suggest that proteins with poly-Ala repeat expansions might interfere with the activity of a poorly understood E1 enzyme, UBA6, and its E2, USE1. They propose that USE1 binding to UBA6 requires a stretch of six Ala residues in the N-terminal extension of the E2 enzyme. Overexpression of proteins with extended poly-Ala regions, as seen in disease, was then shown to reduce the interaction between overexpressed UBA6 and USE1. The authors proceed by proposing that the resulting reduced activity of USE1 leads to a stabilization of the E3 E6AP and subsequent degradation of Arc, a potential substrate of E6AP. These relationships have also been observed in iPSC-derived neurons obtained from patients with polyA-expansion disease.

Answer:

We thank the reviewer for describing and highlighting novel aspects from our work. In the current version of the manuscript we performed additional mechanistic experiments and discussions to further support this model including new data in **Fig. 1 d-g, Fig. 3 f-h, Fig. 4c, Fig. 4 e-g, Extended Data Fig. 3b, Extended Data Fig. 5b, Extended Data Fig. 6 a-g, Extended Data Fig. 7c, e, Extended Data Fig. 8 a-d, Extended Data Fig. 10 c,h,g,i.**

Comment:

Most interaction studies, as for example in Fig. 1b, are using overexpressed FLAG-USE1 and HA-Ub. To judge the physiological relevance of the observations documented in this work, it would be important to confirm such data without overexpression, i.e. mutate the endogenous Ala stretch in USE1. As the authors later show that a positive region on UBA6 might be important for USE1 binding, it is also kind of unfair to introduce Arg residues into the stretch of 6 Ala residues in USE1 (one could expect electrostatic repulsion, even if the poly-A doesn't specifically contribute to binding). I would strongly recommend more conservative mutations (Leu, Val) and see whether the interaction is disturbed, using a quantitative in vitro assay.

Answer:

We thank the Reviewer for this important comment. In agreement with this view, we have now generated cells with mutation in the endogenous polyalanine domain of USE1. A more conservative mutation to address the contribution of this domain would be, in our opinion, a deletion mutation in which the polyalanine was removed, since this will avoid introducing potential non-relevant effects of alternative residues. We now used a quantitative FLIM-FRET approach to measure the USE1-UBA6 interaction. The new data is presented in **Fig. 1d**. This is now addressed in the Results, page 5: “*We next mutated the endogenous polyalanine stretch of USE1 by generating a knockout HEK293T cell line harboring a deletion of the polyalanine stretch in the UBE2Z alleles (USE1 ΔPolyAla KO). The endogenous USE1-UBA6 interaction in these cells was detected with Förster resonance energy transfer (FRET) based fluorescence lifetime imaging microscopy (FLIM). The results indicated less binding between UBA6 and USE1 in the USE1 ΔPolyAla KO cells compared to control cells (Fig. 1d)*”.

Comment:

The mechanistic studies investigating the importance of Ala residues in USE1 and a positive patch in UBA6 lack required depth. For example, it is not clear how significant certain interactions are. In Fig. 1e, please show input and IP in one blot (that should be done generally) and explicitly state how much lysate is shown compared to IP. Moreover, effects of mutations should be analyzed quantitatively, an established approach for E1/E2 interactions. For example, the authors could investigate the effects of USE1 or UBA6 mutations with recombinant proteins by fluorescence polarization or FRET experiments. Knowing a KD for the USE1/UBA6 interaction and the impact of mutations would be very informative. In a similar manner, the KD of poly-Ala peptides for UBA6 and the effect on the UBA6/USE1 interactions needs to be determined. This is really critical to determine whether any of the overexpression experiments documented in this work can at least be important in cells.

Answer:

The Reviewer is correct and the input and IP for UBA6 and USE1 were run on one blot. We have now included it in **Extended Data Fig. 2c** and indicated how much lysate is shown.

For the second part, we thank the Reviewer for this valuable comment, which was also suggested by Reviewer 3, because now we can show direct biophysical interaction of the peptide with 7 alanine residues (7xAla peptide) with the SCCH domain (known to determine E2s specificities). The Reviewer correctly requested to show in a biophysically manner the direct interaction of the USE1 poly-Ala with the SCCH domain. To address this, we purified the SCCH domain from *E. coli* and used a 7xAla peptide (unfortunately the company could not synthesize longer polyAla peptides), which were tested in MST binding assay. In this assay, the peptide was fluorescently labeled. Different E1s bind the E2 core and present specificity via other regions. It is important to appreciate that the specificity is achieved by weak interactions that are difficult to measure and quantify in terms of KD. Indeed, we could show, and now add to the paper, the binding by these measurements. However, we could not determine the KD as affinity for this interaction is ultra-weak. Careful examination of the MST binding shows that we are far from saturation even at 200 micromolar of the domain. Also, it worth to mention that the 7xAla peptide is insoluble at high concentrations, thus it was kept at low concentrations to allow solubility. Most importantly, our data clearly shows a significant binding differences between UBA1 and UBA6. The new data was added in **Extended Data Fig.3b**. Please see Results page 7: *“Indeed, a biophysical analysis revealed that a peptide of 7 alanine residues interacts directly with the SCCH domain of UBA6, but has significantly weaker binding to the SCCH domain of UBA1 (Extended Data Fig. 3b)”*.

In addition, we pointed here and in the text of the new manuscript that during the revision of our paper, the crystal structures of UBA6 were published and further support the validity of our structural model. The papers do not investigate the polyaniline domain of USE1 and do not mention the polyaniline at all.

We have now related to this in the text of the Results page 6-7: *“Using AlphaFold we constructed a model of UBA6, and intriguingly, during the revision of the paper, the crystal structures of UBA6 have been published^{23, 24}. These experimental data confirmed the AlphaFold model with very minor differences. Comparison of the models for the general structure containing 698 C α atoms yielded an RMSD of 1.2Å. For the second catalytic cysteine half (SCCH) domain alone (168 C α), which assumed a slightly different orientation compared with the enzyme core, the RMSD was 0.78Å”*.

Comment:

Although the authors focus on E6AP as a critically affected cellular protein, effects are really minor. See, for example, Ext. Data Fig. 4. It would be very important to establish whether such a small increase can result in different substrate degradation. The phenotype of UBA6

overexpression is also minor – how could such a minute difference in enzyme levels lead to a different output?

Answer:

E6AP is tightly regulated in the nervous system by various mechanisms including imprinting, PKA-dependent phosphorylation, neurotransmitter-dependent transcription and ubiquitin-dependent proteolysis. Delicate activity changes result in severe neurological effects, such as autism and Angelman syndrome. This explains the tight regulation of E6AP. We would like to point out that the effects we are observing regarding E6AP increased levels in our systems are compatible with physiological induction rates of E6AP protein levels which were documented previously. For example, please see Figure S1 B, E in Greer, P. L. et al. *Cell*, 2010 (PMID: 20211139). In addition, we have now provided new data in **Fig. 1e**, **Fig. 1f**, **Fig. 3g**, and **Extended Data Fig. 6 f, g** on the degradation rates of E6AP by the polyalanine stretches. Please see Results, page 5: “*We therefore examined the stability of E6AP in the USE1 ΔPolyAla KO cells. In control HEK293T cells, a cycloheximide chase experiment revealed that E6AP has an apparent half-life of approximately 10-14 hours (Fig. 1e), which is consistent with previous reports in cultured cells¹³. In contrast, E6AP was stabilized in the USE1 ΔPolyAla KO cells and showed a decrease in Lys48-linked polyubiquitination (Fig. 1e, Fig. 1f), suggesting that proteasome-mediated degradation of E6AP is inhibited in the USE1 ΔPolyAla KO cells*”.

Later on in the Results, page 8-9: “*To test the effects on E6AP degradation, we measured the stability and ubiquitination of E6AP in mutant PHOX2B expressing HEK293T cells. Mutant PHOX2B increases the levels of E6AP due to its stabilization in HEK293T cells (Fig. 3g, Extended Data Fig. 5c, Extended Data Fig. 6f). However, mutant PHOX2B does not increase E6AP mRNA levels, indicating that the increased E6AP levels are not related to transcriptional effects (Extended Data Fig. 6g)*”.

Likewise, in light of the Reviewer comment, we have now discussed in the paper that the increased levels of E6AP are sufficient to perturb E6AP targets. Results, page 10: “*Moreover, expression of mutant PHOX2B with +7 Ala and +13 Ala increase the levels of E6AP in the primary neurons by 1.3 fold (Fig. 3j), which is consistent with physiological induction rates of E6AP protein levels in cultured neurons²⁸. The synaptic activity-regulated cytoskeleton-associated protein (Arc) is negatively regulated by E6AP^{28, 29}, and accordingly, the increase in E6AP levels due to ectopic expression of the PHOX2B mutants with +7 Ala and +13 Ala decrease Arc levels in primary neurons (Fig. 3j, Extended Data Fig. 7f)*”.

Comment:

Fig. 3f: change in apparent K48-ubiquitylation of E6AP is similar to decrease in these conjugates in input sample. Experiment would need to be performed with endogenous ubiquitin; both tagging and overexpression of ubiquitin can lead to aberrant modification of substrates and enzymes. If phenotypes are so small, as seen here for E6AP, overexpression of tagged ubiquitin has to be avoided at all costs.

Answer:

We thank the reviewer for pointing on this issue, which we fully agree with. We repeated the experiment with endogenous ubiquitin and the new data is now presented in **Fig. 3h**.

Comment:

Fig. 3h: again, the effects are minimal, and at least the Arc Western does not look like being developed properly. There are controls missing regarding specificity – could the authors, for example, perform quantitative mass spec on cell lysates and thus provide a more unbiased and more quantitative analysis of changes in protein levels? If Arc is the only, or the major, protein affected in this pathway, then their data would be greatly strengthened. If, by contrast, Arc was the only protein they investigated, then the data is weak (especially given the minor effects seen here).

Answer:

For the first part of the comment, we have now included Arc Western from an additional experiment in **Extended Data Fig. 7f** to support the significant reduction in Arc levels.

As for the second part, we monitored the E6AP and Arc levels as reliable physiological readouts for perturbations in the UBA6-USE1 interaction. Indeed, Arc is not the only and probably not the major component in this cascade as the Reviewer commented. Instead of choosing mass spec experiment to pinpoint on major affected specific protein levels, we chose to tackle UBA6 effects on the survival of neurons from patients with polyalanine expansion mutations (new results in **Fig. 4 f,g**). This demonstrates the pathological relevance of our findings in an unbiased manner, which we believe is aligned with the Reviewer views on unbiased outputs.

Comment:

One of the major shortcomings of this paper remains that mechanistic insight is limited – how do Ala-stretches modulate E1 binding (they are not in the domain normally used for it)? Why are proteins with extended poly-Ala repeats accumulating in the cytoplasm? Are they caught right off the ribosome by quality control pathways or are they exported out of the nucleus? Are they captured by other cytoplasmic proteins? As UBA6 seems to be mainly cytoplasmic, and because the authors suggest that polyA proteins exert their effect onto UBA6, these are important questions to answer. It is currently very difficult to determine whether apparent changes in UBA6 activity or E6AP ubiquitylation are direct or indirect.

Answer:

We have now included mechanistic studies and discussions to address the reviewer questions:

1. how do Ala-stretches modulate E1 binding (they are not in the domain normally used for it)

We can now answer this question and show in Extended Data Fig.3b that the SCCH domain of UBA6, which is involved in E2 specificity, can directly interact with Ala-stretches (please also see our previous elaborated answer on this point).

2. Why are proteins with extended poly-Ala repeats accumulating in the cytoplasm? Are they caught right off the ribosome by quality control pathways or are they exported out of the nucleus? Are they captured by other cytoplasmic proteins?

This is a relevant question to answer. We have now included a reference in the Discussion supporting the nuclear export mechanism for the polyalanine expansion mutation. Please see Discussion, page 13: “*Indeed, the expanded polyalanine domain itself can serve as a nuclear export signal*⁴³”. While nuclear export could explain the mislocalization behavior of the mutant protein, it was unclear how cytoplasmic mislocalization is associated with toxicity and potential harmful gain of function. Our model suggests that sequestering the enzyme UBA6 into soluble cytoplasmic polyalanine expansions may represent a deleterious mechanism for such mutations. This is now further supported by the new data in **Fig. 4** showing that overexpression of UBA6 in the patient neurons rescues cell death.

3. It is currently very difficult to determine whether apparent changes in UBA6 activity or E6AP ubiquitylation are direct or indirect.

To address this comment, we performed additional new mechanistic experiments (**Fig. 1g**, **Fig. 3f**, **Extended Data Fig. 3b**, and **Extended Data Fig. 6 a-e**) supporting direct effects of wild-type and expanded polyalanine stretches on UBA6 binding, UBA6 activity and E6AP ubiquitination. We addressed this in Results page 5-6: “*Since E6AP is monoubiquitinated and polyubiquitinated by the UBA6-USE1 cascade*¹³, we further investigated the role of the polyalanine stretch of USE1 in E6AP ubiquitination by incubating purified E6AP with UBA6, USE1 WT and USE1 mutants (Δ PolyAla and C188A) *in vitro*. The results indicated that the incubation with the USE1 Δ PolyAla decreased the polyubiquitination of E6AP but not the monoubiquitin conjugate, similarly to the effects of the USE1 C188A catalytically dead mutant (Fig. 1g)”.

Results page 7: “*Indeed, a biophysical analysis revealed that a peptide of 7 alanine residues interacts directly with the SCCH domain of UBA6, but has significantly weaker binding to the SCCH domain of UBA1 (Extended Data Fig. 3b)”.*

Results page 8: “*Consequently, the interactions between UBA6 and the mutant proteins (PHOX2B +13 Ala, RUNX2 +12 Ala, HOXD13 +10 Ala, and PABPN1 +7 Ala) are measurable in the cytoplasmic but not in the nuclear fraction of the cells (Extended Data Fig. 6 a-d), which reduces the USE1 ubiquitin loading (Extended Data Fig. 6 e)”.*

Later on in Results page 8: “*Our results indicate that recombinant mutant PHOX2B (+ 13 Ala) competes with USE1 for binding to UBA6 in vitro (Fig. 3f)”.*

Comment:

It is also unclear whether E6AP is an important target in this pathway, and whether the link to UBA6 is related to disease. To address these questions, the authors would need to implement a functional assay that monitors neuronal output – similar to what Marius Wernig and Tom Suedhoff have done for APP processing in induced neurons, for example. If they see defects in their iPSC derived neurons from patients, they could then ask whether these are rescued by overexpression of UBA6. Alternatively, they could test whether reducing E6AP levels could rescue a neuronal phenotype – although I would be surprised that the minor changes in the levels of an enzyme really have a devastating consequence. Without such mechanistic insight, much of this paper shows correlation, but does not provide evidence for causation.

Answer:

We thank the reviewer for suggesting this experiment, which was also suggested by Reviewer 2. We have now provided new data supporting that the overexpression of UBA6 increases the resilience of the patient neurons to cell death. The new results are presented in **Fig. 4f**, **Fig. 4g** and **Extended Data Fig. 10i**. We addressed this in Results, page 12: *“we asked how does UBA6 affect polyalanine expanded proteins-associated toxicity? Because we observed apoptotic nuclear morphology in the CCHS patient neurons (Extended Data Fig. 10d), we next attempted to rescue the cell death in CCHS neurons by their transduction with lentiviruses expressing the cDNA of UBA6 (Fig. 4f, g), which resulted in robust UBA6 expression in both cell body and neurites (Extended Data Fig. 10i). Interestingly, the overexpression of UBA6 significantly reduced the number of CCHS neurons positive to terminal deoxynucleotidyl transferase dUTP nick end labeling (TUNEL) (Fig. 4g), and also decreased the amounts of phosphatidylserine on the extracellular surface as detected by Annexin V binding (Fig. 4f). We can therefore conclude that overexpression of UBA6 in the CCHS neurons rescues them from neuronal death”*.

Referee #2

Comment:

In this study, Amer-Sarsour et al. used HEK293T to report that the ubiquitin-activating enzyme 6 (UBA6) binds to the polyalanine domain of the ubiquitin E2 conjugating enzyme 1 (USE1), and that polyalanine-expanded proteins disrupt this interaction. Focusing on one polyalanine expansion disorder - congenital central hypoventilation syndrome (CCHS), the authors then showed an increased association between UBA6 and mutant PHOX2B, and higher levels E3 ubiquitin ligase E6AP in mouse primary neurons and in CCHS patient derived iPSCs. The results seem valid and interesting but not sufficient for publishing in [the journal where the manuscript was originally submitted].

Answer:

We thank the reviewer for the positive comment, highlighting novel aspects from our work. In the current version of the manuscript we performed additional mechanistic experiments and

discussions to further support this model including new data in **Fig. 1 d-g**, **Fig. 3 f-h**, **Fig. 4c**, **Fig. 4 e-g**, **Extended Data Fig. 3b**, **Extended Data Fig. 5b**, **Extended Data Fig. 6 a-g**, **Extended Data Fig. 7c, e**, **Extended Data Fig. 8 a-d**, **Extended Data Fig. 10 c,h,g,i**.

Comment:

The finding of UBA6 and USE1 interaction is not original. Previous studies (cited in this manuscript) have shown that USE1 interacts specifically with the UBA6 enzyme rather than the conventional E1 enzyme, and that Uba6 is required for charging USE1:

Jin, J., Li, X., Gygi, S. P., & Harper, J. W. (2007). Dual E1 activation systems for ubiquitin differentially regulate E2 enzyme charging. *Nature*, 447(7148), 1135–1138.

<https://doi.org/10.1038/nature05902>

Lee, P. C., Sowa, M. E., Gygi, S. P., & Harper, J. W. (2011). Alternative ubiquitin activation/conjugation cascades interact with N-end rule ubiquitin ligases to control degradation of RGS proteins. *Molecular cell*, 43(3), 392–405. <https://doi.org/10.1016/j.molcel.2011.05.034>

However, the link between UBA6 and polyalanine tracts in normal proteins and mutant proteins presents new results. This is a well-written and systematically constructed paper. Nevertheless, there are specific issues that need to be addressed as listed below:

Answer:

We agree with the reviewer that the link and mechanism of UBA6 regulation by polyalanine tracts in normal proteins and mutant proteins present new results, and we appreciate the enthusiasm of the reviewer for our work. We have addressed the reviewer's comments below by performing the requested experiments, which further strengthened our model.

Major comments

Comment:

The authors claim that disease-causing proteins with polyalanine expansions interact with UBA6. However, they state that isolated polyalanine stretches with NLS did not bind UBA6 and had no apparent effect on USE1 loading. All polyalanine disease-causing proteins are nuclear proteins, and UBA6 is mainly localized in the cytoplasm. The authors also mention that only when these proteins with polyalanine expansions partially mislocalized to the cytoplasm they then could interact with endogenous UBA6. The authors should determine and present the interaction between UBA6 and the nuclear/cytoplasmic fraction for every mutant polyalanine protein.

Answer:

As suggested by the Reviewer, we determined the interaction between UBA6 and the nuclear/cytoplasmic fraction for every mutant polyalanine protein, and the new results are now presented in **Extended Data Fig. 6 a-d**. Please see also Results page 8: “*Consequently, the interactions between UBA6 and the mutant proteins (PHOX2B +13 Ala, RUNX2 +12 Ala,*

HOXD13 +10 Ala, and PABPN1 +7 Ala) are measurable in the cytoplasmic but not in the nuclear fraction of the cells (Extended Data Fig. 6 a-d)”.

Comment:

It is important to mention that expanded PABPN1 has not been found to mislocalize in the cytoplasm. Hence, it is not appropriate to generalize that these polyalanine mutant proteins interact with endogenous UBA6 in the cytoplasm.

Answer:

To address this issue we have now obtained OPMD patient-derived biopsy that is highly vulnerable to the PABPN1 polyalanine expansion mutation. We could demonstrate that patient-derived cricopharyngeal myotubes and primary fibroblasts showed the cytoplasmic presence of PABPN1. We have also added a reference supporting the nuclear-cytoplasmic shuttling ability of PABPN1. Moreover, our analysis of the OPMD-patient-derived cells further supports our model and the data was now added to **Extended Data Fig. 8**. Please see Results, page 10: “*The cricopharyngeal muscle of OPMD patients is vulnerable to the polyalanine expansion mutations in the PABPN1 gene*³⁰. *OPMD patient-derived cricopharyngeal myotubes and primary fibroblasts as well as non-affected control fibroblasts, express PABPN1 both in the cytoplasm and in the nucleus (Extended Data Fig. 8a, b), which could be related to the ability of PABPN1 to shuttle between the nucleus and the cytoplasm*³¹. *Accordingly, OPMD patient-derived fibroblasts have less interaction between UBA6 and USE1 than control fibroblasts (Extended Data Fig. 8 c-d)”.*

Comment:

These polyalanine-expanded proteins can be present in either a soluble or insoluble fraction. When studying the ubiquitination of these mutant proteins it is of importance to separate the soluble and insoluble fractions efficiently. Aggregated proteins are ubiquitinated differently when compared to their soluble counterparts. The authors should perform the soluble/insoluble assay and compare the interaction between UBA6 with soluble polyalanine mutant proteins versus insoluble counterparts. Ubiquitin is involved in aggregate formation of expanded polyalanine. It is important to test whether UBA6-USE1 co-localize with polyalanine aggregates or not.

Answer:

We thank the Reviewer for this point and thus we performed soluble or sarkosyl-insoluble fractionation assays for various cell types expressing wild-type and expanded polyalanine proteins to address this issue. In addition, we also tested if UBA6 co-localizes with polyalanine aggregates. Our results suggest that the soluble fraction of the polyalanine expansion is sufficient to exert effects on UBA6. The new data was now added in **Extended Data 5b, Extended 7c, and Extended Data 10c**. Please see results page 8: “*Biochemical analysis of the GFP-19Ala*

could not detect a sarkosyl-insoluble fraction, suggesting that the soluble polyalanine stretches can interact with UBA6 (Extended Data Fig. 5b)”.

Results page 9 regarding ectopic expression of expanded polyalanine proteins with lentiviruses in primary neurons: *“While most of the wild type PHOX2B could be detected in the nucleus, mutant PHOX2B (+13 Ala) is also present in the cell body and in neurites, where the GFP fluorescence imaging and biochemical analysis revealed both non-aggregated and aggregated patterns (Extended Data Fig. 7b, c)”*. Later in Results page 9: *“An analysis of UBA6 abundance revealed that UBA6 is predominantly associated with the non-aggregated forms of PHOX2B (Extended Data Fig. 7e)”*.

Results page 11 related to endogenous expression of polyalanine-expanded proteins in patient neurons: *“Analysis of the PHOX2B localization revealed a decrease in the nuclear fraction of PHOX2B in the mutant 20/25 and 20/27 neurons, as compared to healthy control neurons (Extended Data Fig. 10 a,b). The cytoplasmic PHOX2B is soluble, and predominantly perinuclear without visible aggregates (Extended Data Fig. 10 c,d). In addition, there was an increased association between UBA6 and PHOX2B in the CCHS patient neurons (Fig. 4b), accompanied by a decrease in the interaction between UBA6 and USE1 (Fig. 4c)”*.

Comment:

The authors used one biochemical method (i.e. co-immunoprecipitation) to determine the interaction between UBA6 and USE1 and the interaction between UBA6 and various disease-causing proteins with polyalanine expansion mutations. It is important to confirm this interaction using another method (such as two-hybrid assays) where the proteins are more likely to be in their native conformation.

Answer:

To address this comment, we performed quantitative FRET-FLIM experiments to determine the endogenous cellular interaction in the native conformation between USE1 and UBA6 in different cell types, either wild-type cells or from patients with polyalanine expansion mutations. The new Data is presented in **Fig. 1d, Fig. 4c, Extended Data Fig. 8 c-d**.

Please see results, page 5: *“We next mutated the endogenous polyalanine stretch of USE1 by generating a knockout HEK293T cell line harboring a deletion of the polyalanine stretch in the UBE2Z alleles (USE1 Δ PolyAla KO). The endogenous USE1-UBA6 interaction in these cells was detected with Förster resonance energy transfer (FRET) based fluorescence lifetime imaging microscopy (FLIM). The results indicated less binding between UBA6 and USE1 in the USE1 Δ PolyAla KO cells compared to control cells (Fig. 1d)”*.

Results, page 10 (using FRET-FLIM): *“Accordingly, OPMD patient-derived fibroblasts have less interaction between UBA6 and USE1 than control fibroblasts (Extended Data Fig. 8 c-d)”*.

Results page 11 (using FRET-FLIM in Fig. 4c): *“In addition, there was an increased association between UBA6 and PHOX2B in the CCHS patient neurons (Fig. 4b), accompanied by a decrease in the interaction between UBA6 and USE1 (Fig. 4c)”*.

Comment:

How does UBA6 affect polyalanine expanded proteins-associated toxicity? In the conclusion's section, the authors mention that the inhibition of UBA6 by sequestering the enzyme into cytoplasmic mislocalized polyalanine expansions represent a deleterious mechanism for such mutations. The authors should perform should investigate whether overexpression of recombinant UBA6 would restore polyubiquitylation activity by USE1 for polyalanine expanded proteins and rescue cell death.

Answer:

We highly appreciate this comment to overexpress UBA6 and investigate effects on cellular defects caused by the mutant proteins, which was also suggested by Reviewer 1. We addressed the pathological relevance of our findings by focusing on the analysis of cell death. We have now provided new data supporting that the overexpression of UBA6 increases the resilience of the patient neurons to cell death. The new results are presented in **Fig. 4f, Fig. 4g** and **Extended Data Fig. 10i**. Please see results, page 12: *“we asked how does UBA6 affect polyalanine expanded proteins-associated toxicity? Because we observed apoptotic nuclear morphology in the CCHS patient neurons (Extended Data Fig. 10d), we next attempted to rescue the cell death in CCHS neurons by their transduction with lentiviruses expressing the cDNA of UBA6 (Fig. 4f, g), which resulted in robust UBA6 expression in both cell body and neurites (Extended Data Fig. 10i). Interestingly, the overexpression of UBA6 significantly reduced the number of CCHS neurons positive to terminal deoxynucleotidyl transferase dUTP nick end labeling (TUNEL) (Fig. 4g), and also decreased the amounts of phosphatidylserine on the extracellular surface as detected by Annexin V binding (Fig. 4f). We can therefore conclude that overexpression of UBA6 in the CCHS neurons rescues them from neuronal death”*.

Comment:

The authors should investigate the interaction between polyalanine expanded proteins and UBA6 in other patients' cells to demonstrate if the same phenomenon occurs as in CCHS iPSCs.

Answer:

We have now included the analysis of cellular material derived from OPMD patients in Extended Data Fig. 8, which shows effects on UBA6 with a similar trend as in the CCHS-derived cells.

Minor comments

Comment:

In Figure 1C, the decrease level of ub-USE of the analyzed proteins following USE1 Δ PolyAla mutant is not really evident and does not match the reported quantification. There is no difference between lane 1 and lane 2. The authors must provide better quality immunoblotting and quantification results in line with the images.

Answer:

We thank the reviewer for spotting this. We have now included an additional blot analysis from another experiment, which better reflect the quantification. The data is now presented in **Extended Data Fig.2a**.

Comment:

In the Extended Data Fig. 4., the title needs to be changed from: Polyalanine expansion mutations regulate ... to isolated polyalanine stretches regulate....

The level of E6AP was not measured in cells expressing expanded polyalanine proteins.

Answer:

We thank the Reviewer for this comment. In Extended Data Fig. 4 in previous version, which is now Extended Data Fig. 5, we also presented analysis of solubility of the polyalanine stretches. Therefore, as suggested by the Reviewer we now changed the title to read: “Soluble isolated polyalanine stretches...”

Comment:

For some of the confocal microscopy images, the authors need to provide micrographs that are at lower magnification so that the reader would see more cells in the field that would be consistent with the quantifications (e.g., Extended Data Fig. 3 and Figure 4d).

Answer:

We thank the Reviewer for this comment. We have now provided lower magnification micrographs to capture more cells. Please see the micrographs that the Reviewer requested as well as additional ones in **Fig. 1d, Fig. 4c, Fig. 4 f,g, Extended Data Fig. 4, Extended Data Fig. 8c, Extended Data Fig. 10 e, i**.

Comment:

In Figure 3h, the authors should explain the reduced level of PHOX2B that is observed in lane 3 (primary neurons expressing GFP-PHOX2B+13Ala). This could produce the increased levels of E6AP observed in this blot.

Answer:

We appreciate this comment. The reduced levels of the PHOX2B+13Ala observed in lane 3 (primary neurons expressing GFP-PHOX2B+13Ala) is probably because of an aggregated fraction which could not be resolved properly in the lysis buffer. We have now included data to support this in the insoluble fractionation biochemical assay in **Extended Data Fig.7c**.

Comment:

In Figure 4d, the authors should count more than 40 neurons for immunostaining of Arc and TUB β 3 in autonomic neurons from control and CCHS patients.

Answer:

As suggested by the reviewer, we have now included additional analysis of more neurons, which supports the trend we observed. The Data is presented in **Extended Data Fig. 10 e-f**.

Comment:

In the Methods (Bioinformatics analysis), the authors wrote: Prosite (ref pubmed id 23161676) was used to scan for alanine residues motifs with between 6 and 10 continuous alanine residues in the BLAST results. They should add: A search for proteins containing polyalanine stretches in the ubiquitin cascades. They performed Blast only in the ubiquitin pathway.

Answer:

The sentence regarding the Bioinformatics analysis was revised accordingly.

Comment:

In the Extended Data Fig. 1, generated from the Blast, there are 13 ubiquitin enzymes all contain alanine residues (e.g., ZFP91 contains 9 alanine residues). The authors should explain the rationale behind choosing USE1 and not another ubiquitin enzyme?

Answer:

In our Bioinformatics analysis, USE1 was the only E2 enzyme containing polyalanine while the others have E3-ubiquitin ligase activities. Since E2s are likely to work with more than one E3 enzyme and there are currently nine different diseases with polyalanine expansion mutation in different proteins, we reasoned that the investigation of upstream regulation of the ubiquitin cascade might have a broader effect.

Referee #3

Comment:

Amer-Sarsour et al. investigated the mechanism of polyalanine expansion diseases. Their hypothesis is that an E1 ubiquitin-activating enzyme UBA6, recognizes a polyalanine stretch in the ubiquitin-conjugating E2 enzyme UBE2Z/USE1, and the result of various polyalanine

expansion gene products can compete for the binding and therefore negatively affects protein degradation. The authors propose that the competition for UBA6 provides a shared mechanism that mediates the neurodegeneration of polyalanine expansion diseases. While authors presented an interesting hypothesis, almost all of the experimental evidence are from overexpression studies and some key data are very weak at best. The evidence presented are very far from sufficient to prove this model.

Answer:

We thank the Reviewer for appreciating our suggested mechanism, and the comments to strengthen our model. Through additional mechanistic experiments we now provide stronger evidence at the endogenous levels in patient cells and non-affected material. The new data is now presented in **Fig. 1 d-g, Fig. 3 f-h, Fig. 4c, Fig. 4 e-g, Extended Data Fig. 3b, Extended Data Fig. 5b, Extended Data Fig. 6 a-g, Extended Data Fig. 7c, e, Extended Data Fig. 8 a-d, Extended Data Fig. 10 c,h,g,i.**

Major issues

Comment:

The data that support this model came from overexpression of 293T cells which cannot differentiate direct vs. indirect effects. The computational derived model (Figure 2) predicts the direct interaction of polyalanine with the binding site in polyalanine length specific manner. This prediction should be tested biochemically using in vitro binding studies including the measurement of binding affinities of various polyalanine stretches.

Answer:

We thank the reviewer for the comment, which was also suggested by Reviewer 1, because now we can show direct biophysical interaction of the peptide with 7 alanine residues (7xAla peptide) with the SCCH domain (known to determine E2s specificities). The Reviewers correctly requested to show in a biophysically manner the direct interaction of the USE1 poly-Ala with the SCCH domain. To address this, we purified the SCCH domain from *E. coli* and used a 7xAla peptide (unfortunately the company could not synthesize longer polyAla peptides), which were tested in MST binding assay. In this assay, the peptide was fluorescently labeled. Different E1s bind the E2 core and present specificity via other regions. It is important to appreciate that the specificity is achieved by weak interactions that are difficult to measure and quantify in terms of KD. Indeed, we could show, and now add to the paper, the binding by these measurements. However, we could not determine the KD as affinity for this interaction is ultra-weak. Careful examination of the MST binding shows that we are far from saturation even at 200 micromolar of the domain. Also, it worth to mention that the 7xAla peptide is insoluble at high concentrations, thus it was kept at low concentrations to allow solubility. Most importantly, our data clearly shows a significant binding differences between UBA1 and UBA6. The new data was added in **Extended Data Fig.3b**. Please see Results page 7: “*Indeed, a biophysical analysis revealed that a peptide of 7 alanine residues interacts directly with the SCCH domain of UBA6, but has significantly weaker binding to the SCCH domain of UBA1 (Extended Data Fig. 3b)*”.

Comment:

The authors present data that suggest USE1 2A-2R mutation affecting its ubiquitin-loading which is partially UBA6 dependent (Figure 1). Does this USE1 2A-2R mutation affect its interaction with UBA6?

Answer:

We have now generated cells with mutation in the endogenous polyalanine domain of USE1. Analyzing the data at the endogenous level with no overexpression was suggested by Reviewer 3 in the first comment and also by Reviewer 1. While we introduced different alterations in the polyalanine stretch of USE1 (2A-2R and a deletion mutation), all of which presented reduced USE1 ubiquitin loading, Reviewer 1 requested in depth analysis of a more conservative mutation. This is why we chose to further investigate the polyalanine deletion mutation, since this will avoid introducing potential non-relevant effects of alternative residues. We used a quantitative FLIM-FRET approach to measure the USE1-UBA6 interaction, which is more physiologically relevant and can preserve the native conformation as requested by Reviewer 2. The new data is presented in **Fig. 1d**. Please see results, page 5: “*We next mutated the endogenous polyalanine stretch of USE1 by generating a knockout HEK293T cell line harboring a deletion of the polyalanine stretch in the UBE2Z alleles (USE1 ΔPolyAla KO). The endogenous USE1-UBA6 interaction in these cells was detected with Förster resonance energy transfer (FRET) based fluorescence lifetime imaging microscopy (FLIM). The results indicated less binding between UBA6 and USE1 in the USE1 ΔPolyAla KO cells compared to control cells (Fig. 1d)*”.

Comment:

Can polyalanine and other polyalanine proteins, e.g. PHOX2B, RUNX2 and HOXD13, PABPN1 etc, affect the ubiquitin-loading of USE1? In Extended Data Figure 4a, GFP-19Ala only marginally, if there is any, decreased the loading of ubiquitin onto USE1.

Answer:

We appreciate this comment. We can now show that the other mutant polyalanine proteins decreased the ubiquitin loading of USE1. The new results are in Extended Data Fig. 6e. We addressed this in the Results page 8: “*Consequently, the interactions between UBA6 and the mutant proteins (PHOX2B +13 Ala, RUNX2 +12 Ala, HOXD13 +10 Ala, and PABPN1 +7 Ala) are measurable in the cytoplasmic but not in the nuclear fraction of the cells (Extended Data Fig. 6 a-d), which reduces the USE1 ubiquitin loading (Extended Data Fig. 6 e)*”.

Comment:

Does overexpression of polyalanine-expanded proteins disrupt interaction between endogenous UBA6 and USE1? In addition, in patient-derived cells or tissues which contain endogenous polyalanine-expanded proteins, is the interaction between endogenous UBA6 and USE1 destroyed or reduced? Co-IP assays should be performed in these cell or tissue models.

Answer:

We have now performed endogenous interaction analysis between UBA6 and USE1 in patient-derived cells. We analyzed autonomic neurons with polyalanine expansion mutation in PHOX2B from CCHS patients, and also included the analysis of OPMD patient-derived primary fibroblasts with the polyalanine expansion mutation in PABPN1. We used FRET-FLIM to measure the interaction in a quantitative way. The new data is in **Fig. 4c**, and **Extended Data Fig. 8 c-e**. Please see also Results, page 10: “*Accordingly, OPMD patient-derived fibroblasts have less interaction between UBA6 and USE1 than control fibroblasts (Extended Data Fig. 8 c-d)*”.

Results page 11: “*In addition, there was an increased association between UBA6 and PHOX2B in the CCHS patient neurons (Fig. 4b), accompanied by a decrease in the interaction between UBA6 and USE1 (Fig. 4c)*”.

Comment:

The authors observed a correlation between overexpression of PHOX2B-13Ala and upregulation of E6AP (and downregulation of Arc). However, the underlying mechanism is unclear.

Answer:

We thank the reviewer for this comment and we have now performed additional mechanistic experiments to support this correlation. We provided new data in **Fig. 3g**, **Fig. 4e** and **Extended Data Fig. 6 f, g** on the degradation rates of E6AP by PHOX2B-13Ala. Please see Results, page 8-9: “*To test the effects on E6AP degradation, we measured the stability and ubiquitination of E6AP in mutant PHOX2B expressing HEK293T cells. Mutant PHOX2B increases the levels of E6AP due to its stabilization in HEK293T cells (Fig. 3g, Extended Data Fig. 5c, Extended Data Fig. 6f). However, mutant PHOX2B does not increase E6AP mRNA levels, indicating that the increased E6AP levels are not related to transcriptional effects (Extended Data Fig. 6g). This correlates with a decrease in Lys48-linked polyubiquitination of E6AP (Fig. 3h, Extended Data Fig. 6h), suggesting that proteasome-mediated degradation of E6AP is inhibited by mutant PHOX2B in the cells*”.

Later on in Results page 12: “In accordance with the mouse data, the levels of E6AP protein are higher in the patient neurons (Fig. 4d), although no change in E6AP mRNA was detected (Fig. 4e)”.

Since Arc levels are negatively regulated by E6AP at the transcriptional and post-translational levels (PMID: 20211139, 23671107), we could not perform a proper degradation assays for Arc, and instead we showed Arc steady state levels by western blots and immunostaining in the primary and autonomic neurons.

Comment:

In Figure 3h and Extended Data Figure 4a, only marginal or mild upregulation of E6AP is observed when PHOX2B+13Ala is overexpressed, and the authors claimed that PHOX2b+13Ala inhibits E6AP degradation. To obtain such a conclusion, one should compare the half-lives of E6AP in the presence and absence of polyalanine expansion. Thus, other experiments, such as cycloheximide chase or pulse chase assays, are needed. In addition, to rule out transcriptional effects, the authors could also examine E6AP mRNA level both in overexpressed and patient-derived cell models. Similar experiments should be performed for Arc as well.

Answer:

Please see our answer above. We performed the requested cycloheximide chase experiments for E6AP and ruled out transcriptional effects (new data in **Fig. 3g**, **Fig. 4e** and **Extended Data Fig. 6 f, g**). Please also note our answer above regarding the difficulties in performing such chase assays for Arc. In addition, we added new results in **Fig. 1e,f** determining the half-life of E6AP in cells mutated in the endogenous polyalanine domain of USE1. Please see Results, page 5: “*In control HEK293T cells, a cycloheximide chase experiment revealed that E6AP has an apparent half-life of approximately 10-14 hours (Fig. 1e), which is consistent with previous reports in cultured cells¹³. In contrast, E6AP was stabilized in the USE1 ΔPolyAla KO cells and showed a decrease in Lys48-linked polyubiquitination (Fig. 1e, Fig. 1f), suggesting that proteasome-mediated degradation of E6AP is inhibited in the USE1 ΔPolyAla KO cells*”. We believe these additional experiments at the endogenous level with no overexpression further support the mechanisms we are proposing.

Comment:

In Page 2, the authors claimed that “Aberrations in this polyalanine stretch reduced ubiquitin transfer to ... downstream target, the E3 ubiquitin ligase, E6AP.” However, result in Fig. 3f is not sufficient to prove this conclusion. It has been shown that UBA6-USE1 ubiquitinates E6AP in vitro (PMC3640669). The authors should reconstitute the reaction and test if the polyalanine stretch in USE1 is required for in vitro ubiquitination activity. In addition, the authors should test if the recombinant polyalanine-stretch containing proteins (e.g., PHOX2B+13Ala) directly inhibits E6AP ubiquitination using the in vitro assay.

Answer:

We agree with the Reviewer. We reconstituted the UBA6-USE1 ubiquitination reaction of E6AP and tested the effect of the polyalanine stretch in USE1. We added the new result in **Fig.1g**. Please see also results page 5-6: “*Since E6AP is monoubiquitinated and polyubiquitinated by the UBA6-USE1 cascade¹³, we further investigated the role of the polyalanine stretch of USE1 in E6AP ubiquitination by incubating purified E6AP with UBA6, USE1 WT and USE1 mutants (ΔPolyAla and C188A) in vitro. The results indicated that the incubation with the USE1 ΔPolyAla decreased the polyubiquitination of E6AP but not the monoubiquitin conjugate, similarly to the effects of the USE1 C188A catalytically dead mutant (Fig. 1g)*”.

Moreover, instead of performing the same ubiquitination experiment but with PHOX2B+13Ala, we designed a competition experiment with recombinant PHOX2B+13Ala to demonstrate that the mutant polyalanine expanded protein directly interacts with UBA6 and interfere with USE1 binding. We believe this would better answer the Reviewer's comment regarding the underlying mechanism. The new data is presented in **Fig. 3f**. Please see also Results page 9: "*Our results indicate that recombinant mutant PHOX2B (+ 13 Ala) competes with USE1 for binding to UBA6 in vitro (Fig. 3f)*".

Comment:

In addition to E6AP, Shank3 has been also reported to enrich in UBA6-deficient cells. Did the expression of polyalanine-expanded proteins lead to upregulation of Shank3? In patient-derived cells, is the Shank3 upregulated?

Answer:

We highly appreciate this suggestion and thus we have attempted to analyze Shank3 levels in the patient autonomic neurons to see effects at the endogenous level. Unfortunately, we could not detect Shank3 expression in the human iPSC-derived autonomic neurons. We added Data addressing this issue in **Extended Data Fig. 10h**. Please see also Results page 12: "*In addition to E6AP, the UBA6-USE1 cascade has been shown to regulate the levels of the synaptic protein shank3¹³. We could not detect endogenous expression of shank3 in the human autonomic neurons (Extended Data Fig. 10 h), thereby precluding the analysis of UBA6-USE1 effects on shank3 in these neurons*". In order to address other UBA6 outputs relevant to the pathology of polyalanine expansion diseases, we have now provided new data supporting that the overexpression of UBA6 increases the resilience of the patient neurons to cell death. The new results are presented in **Fig. 4f, Fig. 4g** and **Extended Data Fig. 10i**. Please see results, page 12: "*we asked how does UBA6 affect polyalanine expanded proteins-associated toxicity? Because we observed apoptotic nuclear morphology in the CCHS patient neurons (Extended Data Fig. 10d), we next attempted to rescue the cell death in CCHS neurons by their transduction with lentiviruses expressing the cDNA of UBA6 (Fig. 4f, g), which resulted in robust UBA6 expression in both cell body and neurites (Extended Data Fig. 10i). Interestingly, the overexpression of UBA6 significantly reduced the number of CCHS neurons positive to terminal deoxynucleotidyl transferase dUTP nick end labeling (TUNEL) (Fig. 4g), and also decreased the amounts of phosphatidylserine on the extracellular surface as detected by Annexin V binding (Fig. 4f). We can therefore conclude that overexpression of UBA6 in the CCHS neurons rescues them from neuronal death*".

Other specific points

Comment:

Figure 1d, original immunoblots should be shown.

Answer:

We have now included the blots for the quantification in Extended Data Fig. 2b.

Comment:

Result in Figure 1e is not so reliable, because the expression of FLAG-USE1 is not even in all samples (input). The authors could first affinity purify the UBA6-USE1 complex by the anti-UBA6-immobilized agarose, then elute the USE1 from the UBA6 using 19-Ala peptide and control peptide (or recombinant GFP-19Ala).

Answer:

Thank you for this suggestion. We have now performed similar experiment to the one the Reviewer suggested but included recombinant PHOX2B+13 Ala instead of the isolated polyalanine stretch to better link it to disease. Indeed, the levels of the FLAG-USE1 and HA-UBA6 were even in the input, and the pulldown of UBA6 resulted in a significantly decreased amount of bound USE1 in the presence of the disease protein. The new data is presented in **Fig. 3f**.

Comment:

Figure 2b, the authors determined the interaction between overexpressed USE1 and UBA6. To further validate the result, endogenous interaction of the two proteins should be measured (i.e., in cells without transfection of plasmid).

Answer:

This is an important point. We have added new data analyzing the interaction of endogenous USE1 with UBA6 in cells with no overexpression. This includes USE1 delta Ala KO cells, CCHS patient autonomic neurons, and OPMD patient-derived fibroblasts. The new results are presented in **Fig. 1d, Fig. 4c, and Extended Data Fig. 8 c-d**.

Comment:

Figure 3a-3e, these co-IP experiments lack key controls. Mock IPs with IgG for samples expressing polyalanine stretch-containing preys are required.

Answer:

We have now performed additional co-IP experiments of endogenous UBA6 with IgG controls from nuclear and cytoplasmic fractions of cells expressing different polyalanine-expanded proteins. The new data is presented in **Extended Data Fig. 6 a-d**.

Comment:

Figure 3f, it seems that the IP experiment was performed under native condition, according to the Methods (page 21). To determine the ubiquitination status of E6AP, samples should be denatured by SDS (plus boiling) before IP to disrupt any non-covalent binding.

Answer:

We apologize for this confusion since all experiments determining the ubiquitination status of E6AP were analyzed under SDS conditions plus boiling with reducing agents. We made it clearer in the Methods section.

Comment:

Figure 4c, immunoblots for Arc in control and patient-derived cells should be shown.

Answer:

We have now included blots for Arc levels in control and CCHS patient neurons in Extended Data Fig. 10 g.

Dear Avraham,

Thank you for submitting your revised manuscript for consideration by the EMBO Journal. Your original manuscript was previously reviewed at another journal, and you substantially revised it before submitting it to us. We also obtained its review history from the previous journal (with your and the referees' consent). As per your request, we then asked two additional experts to assess whether the concerns of the original referees have been sufficiently addressed, and to advise us on the potential suitability of your study for publication in the EMBO Journal. We have now received the reports of both advisors, I have already sent you a copy of their comments (which are included again below), and I have asked you for a revision plan, which I have now read carefully and discussed with the other members of our editorial team.

Our advisors both recognize that the study has been thoroughly revised with the addition of new data, and they acknowledge that most of the initial referees' concerns have been successfully addressed in a strengthened and more convincing manuscript. However, advisor #1 also identifies a few points that have not been fully addressed. In particular, we agree that their points #2 (regarding the Arc half-life) and #3 (regarding E6-AP and Arc levels in rescue experiments) need to be addressed for publication in the EMBO Journal. In addition, advisor #2 points out that a better control (i.e. PHOX2B without the Ala tract) should be used in the experiment shown in Fig. 3G. Furthermore, both advisors provide a few more suggestions for minor textual improvements of the manuscript.

Given the advisors' positive comments and recommendations, I would like to invite you to submit a revised version of your manuscript, addressing the aforementioned concerns of both advisors, along the lines you suggested in your revision plan. I should add that it is EMBO Journal policy to allow only a single round of revision, and acceptance of your manuscript will therefore depend on the completeness of your responses in this revised version. If you have any questions or comments, we can also discuss the revisions in a video chat, if you like.

We generally allow three months as standard revision time (12th November 2023). As a matter of policy, competing manuscripts published during this period will not negatively impact our assessment of the conceptual advance presented by your study. However, we request that you contact the editor as soon as possible upon publication of any related work, to discuss how to proceed. Should you foresee a problem in meeting this three-month deadline, please let us know in advance and we may be able to grant an extension.

Thank you for the opportunity to consider your work for publication in the EMBO Journal. I look forward to your revision.

Best regards,

Ioannis

Instructions for preparing your revised manuscript

1. When you are ready to submit the revision, please upload:

- A Word file of the manuscript text (including legends of main Figures, EV Figures and Tables). Please make sure that changes are highlighted (or "tracked") to be clearly visible.

- Individual production-quality figure files (one file per figure). When assembling your figures, please refer to our figure preparation guidelines in order to ensure proper formatting and readability in print as well as on screen:

If the data shown in a figure are obtained from n {less than or equal to} 2, please use scatter plots showing the individual data points.

- i. the name of the statistical test used to generate error bars and P values
- ii. the number (n) of independent experiments (please specify technical or biological replicates) underlying each data point (discussion of statistical methodology can be reported in the Materials and Methods section, but figure legends should contain a basic description of n , P , and the test applied)

iii. the nature of the bars and error bars (s.d., s.e.m.).

- A point-by-point response to the referees' comments, with a detailed description of the changes made (as a word file). All referees' concerns must be fully addressed and their suggestions taken on board. When preparing your letter of response to the referees' comments, please bear in mind that this will form part of the Review Process File and will therefore be available online to the community. Please note that you have the possibility to opt out of the transparent process at any stage prior to publication by letting the editorial office know (contact@embojournal.org); if you do opt out, the Review Process File link will point to the following statement: "No Review Process File is available with this article, as the authors have chosen not to make the review process public in this case.". For more details on our Transparent Editorial Process, please visit our website: <https://www.embopress.org/page/journal/14602075/authorguide#transparentprocess>

- Expanded View (EV) files (replacing Supplementary Information) that are collapsible/expandable online. A maximum of 5 EV Figures can be typeset. EV Figures should be cited as "Figure EV1, Figure EV2" etc. in the text, and their respective legends should be included in the manuscript file after the legends of regular figures. See detailed instructions regarding Expanded View files here:

- For the figures that you do NOT wish to display as Expanded View figures, they should be bundled together with their legends in a single PDF file called "Appendix", which should start with a short Table of Contents (including page numbers). Appendix figures should be referred to in the main text as: "Appendix Figure S1, Appendix Figure S2" etc. Please see detailed instructions here: <https://www.embopress.org/page/journal/14602075/authorguide#expandedview>

- A complete author checklist, which you can download from our author guidelines (<https://www.embopress.org/page/journal/14602075/authorguide>). Please note that the checklist will also be part of the Review Process File.

2. Please note that no statistics should be calculated if $n=2$.

3. Please note that a "Data availability" section at the end of Materials and Methods is mandatory. In case you have no new data that require deposition in a public database, please state so: "Our study includes no data deposited in public repositories." under the heading "Data availability".

See also <https://www.embopress.org/page/journal/14602075/authorguide#dataavailability>). Please note that the "Data availability" statement is restricted to new primary data that are part of this study.

4. Please check that the title and the abstract of the manuscript are brief, yet explicit, even to non-specialists. The length of the title should not exceed 100 characters (including spaces), and the abstract should be a single paragraph not exceeding 175 words.

5. Please also note our reference format: <https://www.embopress.org/page/journal/14602075/authorguide#referencesformat>.

7. Please remember: digital image enhancement is acceptable practice, as long as it accurately represents the original data and conforms to community standards. If a figure has been subjected to significant electronic manipulation, this must be noted in the figure legend or in the "Materials and Methods" section. The editors reserve the right to request original versions of figures and the original images that were used to assemble the figure.

8. Our journal encourages inclusion of data citations in the reference list to directly cite datasets that were obtained from public databases. Data citations in the article text are distinct from normal bibliographical citations and should directly link to the database records from which the data can be accessed. In the main text, data citations are formatted as follows: "Data ref: Smith et al, 2001" or "Data ref: NCBI Sequence Read Archive PRJNA342805, 2017". In the Reference list, data citations must be labeled with "[DATASET]". A data reference must provide the database name, accession number/identifiers, and a resolvable link to the landing page from which the data can be accessed at the end of the reference. Further instructions are available at: <https://www.embopress.org/page/journal/14602075/authorguide#referencesformat>.

9. We request authors to consider both actual and perceived competing interests. Please review our policy (<https://www.embopress.org/page/journal/14602075/authorguide#conflictofinterest>) and update your competing interests statement if necessary. Please name this section 'Disclosure and competing interests statement' and place it after the Acknowledgements section.

10. Please note that all corresponding authors are required to provide an ORCID ID upon submission of a revised manuscript

(<https://orcid.org/>). Please find instructions on how to link your ORCID ID to your account in our manuscript tracking system in our Author guidelines (<https://www.embopress.org/page/journal/14602075/authorguide#authorshipguidelines>).

11. We use CRediT to specify the contributions of each author in the journal submission system. CRediT replaces the author contribution section, which should be removed from the manuscript. Please use the free text box to provide more detailed descriptions. See also guide to authors: <https://www.embopress.org/page/journal/14602075/authorguide#authorshipguidelines>.

13. We would also welcome the submission of cover suggestions or motifs to be used by our Graphics Illustrator in designing a cover.

14. Please use the link below to submit your revision:
<https://emboj.msubmit.net/cgi-bin/main.plex>

Referee #1:

Below you find my comments on each of the issues in red font. I have limited myself to only analyzing whether the concerns of the reviewers have been addressed and refrained from bringing up other issues. Overall, I think that the authors have done a thorough job in addressing the concerns. However, there are a few points that haven't been fully addressed.

1: Two reviewers brought up concerns about the alanine -> arginine substitutions that the authors use to support their model that the alanine stretch in USE1 is responsible for the binding to UBA6. Reviewer #1 proposes to test this with conservative mutations as the charge of the arginine residues could have been a problem. Instead, the authors opted for using mutants in which the entire alanine stretch was deleted (not sure why, I don't agree with their argument that would be a more conservative mutation that substituting the residues). However, such a deletion is also quite dramatic and may indirectly affect the structural integrity and compromise binding of UBA6 to another part of the USE1. I think that given the overall picture of the provided data, it is likely, though not conclusively shown, that the alanine repeat is responsible for the binding. Having said that even performing these experiments with conserved mutations in USE1 will not exclude indirect effects. I personally don't think that additional experimentation is required but am a bit surprised by their argumentation.

2. My interpretation of the comments of Reviewer #1 is that this reviewer is wondering if the small changes in E6-AP are sufficient to have an effect on the half-life of its substrates. The authors address this by analyzing in detail the half-life of E6-AP, which they do properly but which I don't think is the actual point. Instead, they should have looked at the half-life of Arc. In response to comments of Reviewer #3, they state that the effect on Arc half-life cannot be assessed because E6-AP levels have an effect on both transcription and degradation of Arc. I cannot follow their reasoning. If E6-AP has an effect on transcription and translation, this would argue that conclusions about effects on Arc degradation can only be based on the half-life of Arc and not on the steady-state levels (which the authors do). Besides that, I don't understand why an effect of E6-AP on transcription would hinder the authors from analyzing the half-life in a turnover experiment. The initial levels at the start point may be different but one can compensate for that in a turnover experiment.

3. The authors included rescue experiments in which they show that ectopic expression of UBA6 can improve survival of CCHS neurons. This is interesting but also a very indirect readout for the model. Overexpression of UBA6 may improve cell viability by very different mechanisms than the model proposed by the authors. I don't understand why they didn't use the occasion to analyze in these rescue experiments if it restored E6-AP and Arc levels. I am even more surprised as this seems to be what Reviewer #3 is asking for. It would be a matter of taking the lysates of the neurons from the rescue experiments and probe for E6-AP and Arc.

See below for my comments on each of the issues brought up by the reviewers.

Response to reviewers (Amer-Sarsour et al.)

Referee #1

Comment:

Poly-Ala repeat expansions in a variety of proteins, including transcription factors, have been seen to cause a series of developmental diseases. How expansion of a poly-Ala repeat causes aberrant protein behavior and disease is not fully understood. Recent work had suggested that poly-Ala repeats in transcription factors lead to an "unblending" of transcription factor condensates and thereby alters the output of gene expression networks (Basu et al., Cell 2020; this paper is surprisingly not cited by the authors). Whether other mechanisms contribute to disease is not known.

Answer:

We thank the reviewer for this comment. We did not intend to skip this important work by Basu et al. In fact, we pointed in the Discussion the literature supporting the notion that there should be additional mechanisms contributing to polyalanine-expansion diseases besides dysregulation of transcriptional programs. The search for such mechanisms was the focus of our work. We have now revised this to make it clearer and included the Basu et al., Cell 2020 paper in the Discussion.

OK. It is a bit confusing though that they cite it in relation to PHOX2B whereas the cited paper deals with HOXD13.

Comment:

In this manuscript, Amer-Sarsour and colleagues suggest that proteins with poly-Ala repeat expansions might interfere with the activity of a poorly understood E1 enzyme, UBA6, and its E2, USE1. They propose that USE1 binding to UBA6 requires a stretch of six Ala residues in the N-terminal extension of the E2 enzyme. Overexpression of proteins with extended poly-Ala regions, as seen in disease, was then shown to reduce the interaction between overexpressed UBA6 and USE1. The authors proceed by proposing that the resulting reduced activity of USE1 leads to a stabilization of the E3 E6AP and subsequent degradation of Arc, a potential substrate of E6AP. These relationships have also been observed in iPSC-derived neurons obtained from patients with polyA-expansion disease.

Answer:

We thank the reviewer for describing and highlighting novel aspects from our work. In the current version of the manuscript we performed additional mechanistic experiments and discussions to further support this model including new data in Fig. 1 d-g, Fig. 3 f-h, Fig. 4c, Fig. 4 e-g, Extended Data Fig. 3b, Extended Data Fig. 5b, Extended Data Fig. 6 a-g, Extended Data Fig. 7c, e, Extended Data Fig. 8 a-d, Extended Data Fig. 10 c,h,g,i.

Comment:

Most interaction studies, as for example in Fig. 1b, are using overexpressed FLAG-USE1 and HA-Ub. To judge the physiological relevance of the observations documented in this work, it would be important to confirm such data without overexpression, i.e. mutate the endogenous Ala stretch in USE1. As the authors later show that a positive region on UBA6 might be important for USE1 binding, it is also kind of unfair to introduce Arg residues into the stretch of 6 Ala residues in USE1 (one could expect electrostatic repulsion, even if the poly-A doesn't specifically contribute to binding). I would strongly recommend more conservative mutations (Leu, Val) and see whether the interaction is disturbed, using a quantitative in vitro assay.

Answer:

We thank the Reviewer for this important comment. In agreement with this view, we have now generated cells with mutation in the endogenous polyalanine domain of USE1. A more conservative mutation to address the contribution of this domain would be, in our opinion, a deletion mutation in which the polyalanine was removed, since this will avoid introducing potential non-relevant effects of alternative residues. We now used a quantitative FLIM-FRET approach to measure the USE1-UBA6 interaction. The new data is presented in Fig. 1d. This is now addressed in the Results, page 5: "We next mutated the endogenous polyalanine stretch of USE1 by generating a knockout HEK293T cell line harboring a deletion of the polyalanine stretch in the UBE2Z alleles (USE1 PolyAla KO). The endogenous USE1-UBA6 interaction in these cells was detected with Förster resonance energy transfer (FRET) based fluorescence lifetime imaging microscopy (FLIM). The results indicated less binding between UBA6 and USE1 in the USE1 PolyAla KO cells compared to control cells (Fig. 1d)".

A problem with introducing dramatic changes either in amino acid composition (being it introduction of arginine residues or deletion of the alanine stretch) is that this may affect the overall structure of the protein and may have indirect effects on the interaction. From this experiment it is difficult to conclude that the alanine repeat in the endogenous protein is responsible for the interaction. One could argue that in combination with the other data a coherent picture emerges that supports that the alanine stretch is responsible for the binding.

Comment:

The mechanistic studies investigating the importance of Ala residues in USE1 and a positive patch in UBA6 lack required depth. For example, it is not clear how significant certain interactions are. In Fig. 1e, please show input and IP in one blot (that should be done generally) and explicitly state how much lysate is shown compared to IP. Moreover, effects of mutations should be analyzed quantitatively, an established approach for E1/E2 interactions. For example, the authors could investigate the effects of USE1 or UBA6 mutations with recombinant proteins by fluorescence polarization or FRET experiments. Knowing a KD for the USE1/UBA6 interaction and the impact of mutations would be very informative. In a similar manner, the KD of poly-Ala peptides for UBA6 and the effect on the UBA6/USE1 interactions needs to be determined. This is really critical to determine whether any of the overexpression experiments documented in this work can at least be important in cells.

Answer:

The Reviewer is correct and the input and IP for UBA6 and USE1 were run on one blot. We have now included it in Extended Data Fig. 2c and indicated how much lysate is shown.

For the second part, we thank the Reviewer for this valuable comment, which was also suggested by Reviewer 3, because now we can show direct biophysical interaction of the peptide with 7 alanine residues (7xAla peptide) with the SCCH domain (known to determine E2s specificities). The Reviewer correctly requested to show in a biophysically manner the direct interaction of the USE1 poly-Ala with the SCCH domain. To address this, we purified the SCCH domain from E. coli and used a 7xAla peptide (unfortunately the company could not synthesize longer polyAla peptides), which were tested in MST binding assay. In this assay, the peptide was fluorescently labeled. Different E1s bind the E2 core and present specificity via other regions. It is

important to appreciate that the specificity is achieved by weak interactions that are difficult to measure and quantify in terms of KD. Indeed, we could show, and now add to the paper, the binding by these measurements. However, we could not determine the KD as affinity for this interaction is ultra-weak. Careful examination of the MST binding shows that we are far from saturation even at 200 micromolar of the domain. Also, it worth to mention that the 7xAla peptide is insoluble at high concentrations, thus it was kept at low concentrations to allow solubility. Most importantly, our data clearly shows a significant binding differences between UBA1 and UBA6. The new data was added in Extended Data Fig.3b. Please see Results page 7: "Indeed, a biophysical analysis revealed that a peptide of 7 alanine residues interacts directly with the SCCH domain of UBA6, but has significantly weaker binding to the SCCH domain of UBA1 (Extended Data Fig. 3b)".

In addition, we pointed here and in the text of the new manuscript that during the revision of our paper, the crystal structures of UBA6 were published and further support the validity of our structural model. The papers do not investigate the polyalanine domain of USE1 and do not mention the polyalanine at all.

We have now related to this in the text of the Results page 6-7: "Using AlphaFold we constructed a model of UBA6, and intriguingly, during the revision of the paper, the crystal structures of UBA6 have been published 23, 24. These experimental data confirmed the AlphaFold model with very minor differences. Comparison of the models for the general structure containing 698 C atoms yielded an RMSD of 1.2Å. For the second catalytic cysteine half (SCCH) domain alone (168 C), which assumed a slightly different orientation compared with the enzyme core, the RMSD was 0.78Å".

Both of these points have been addressed. The authors have been unable to determine a Kd but they convincingly show that the binding differs between UBA6 and UBA1. I think that suffices.

Comment:

Although the authors focus on E6AP as a critically affected cellular protein, effects are really minor. See, for example, Ext. Data Fig. 4. It would be very important to establish whether such a small increase can result in different substrate degradation. The phenotype of UBA6 overexpression is also minor - how could such a minute difference in enzyme levels lead to a different output?

Answer:

E6AP is tightly regulated in the nervous system by various mechanisms including imprinting, PKA-dependent phosphorylation, neurotransmitter-dependent transcription and ubiquitin-dependent proteolysis. Delicate activity changes result in severe neurological effects, such as autism and Angelman syndrome. This explains the tight regulation of E6AP. We would like to point out that the effects we are observing regarding E6AP increased levels in our systems are compatible with physiological induction rates of E6AP protein levels which were documented previously. For example, please see Figure S1 B, E in Greer, P. L. et al. Cell, 2010 (PMID: 20211139). In addition, we have now provided new data in Fig. 1e, Fig. 1f, Fig. 3g, and Extended Data Fig. 6 f, g on the degradation rates of E6AP by the polyalanine stretches. Please see Results, page 5: "We therefore examined the stability of E6AP in the USE1 PolyAla KO cells. In control HEK293T cells, a cycloheximide chase experiment revealed that E6AP has an apparent half-life of approximately 10-14 hours (Fig. 1e), which is consistent with previous reports in cultured cells 13. In contrast, E6AP was stabilized in the USE1 PolyAla KO cells and showed a decrease in Lys48-linked polyubiquitination (Fig. 1e, Fig. 1f), suggesting that proteasome-mediated degradation of E6AP is inhibited in the USE1 PolyAla KO cells".

Later on in the Results, page 8-9: "To test the effects on E6AP degradation, we measured the stability and ubiquitination of E6AP in mutant PHOX2B expressing HEK293T cells. Mutant PHOX2B increases the levels of E6AP due to its stabilization in HEK293T cells (Fig. 3g, Extended Data Fig. 5c, Extended Data Fig. 6f). However, mutant PHOX2B does not increase E6AP mRNA levels, indicating that the increased E6AP levels are not related to transcriptional effects (Extended Data Fig. 6g)". Likewise, in light of the Reviewer comment, we have now discussed in the paper that the increased levels of E6AP are sufficient to perturb E6AP targets. Results, page 10: "Moreover, expression of mutant PHOX2B with +7 Ala and +13 Ala increase the levels of E6AP in the primary neurons by 1.3 fold (Fig. 3j), which is consistent with physiological induction rates of E6AP protein levels in cultured neurons 28. The synaptic activity-regulated cytoskeleton-associated protein (Arc) is negatively regulated by E6AP 28, 29, and accordingly, the increase in E6AP levels due to ectopic expression of the PHOX2B mutants with +7 Ala and +13 Ala decrease Arc levels in primary neurons (Fig. 3j, Extended Data Fig. 7f)".

As I interpret it the reviewer is wondering if the small changes in E6-AP will result in a change of the half-life of any of its substrates. This is a valid argument. Instead, the authors focus on the half-life of E6-AP itself. It seems to me that the authors have misunderstood the valid concern of the reviewer. Later they argue that the effect of E6-AP changes on the half-life cannot be determined because it will affect both transcription and degradation but I don't understand that argument (see also below).

Comment:

Fig. 3f: change in apparent K48-ubiquitylation of E6AP is similar to decrease in these conjugates in input sample. Experiment would need to be performed with endogenous ubiquitin; both tagging and overexpression of ubiquitin can lead to aberrant modification of substrates and enzymes. If phenotypes are so small, as seen here for E6AP, overexpression of tagged ubiquitin has to be avoided at all costs.

Answer:

We thank the reviewer for pointing on this issue, which we fully agree with. We repeated the experiment with endogenous ubiquitin and the new data is now presented in Fig. 3h.

OK.

Comment:

Fig. 3h: again, the effects are minimal, and at least the Arc Western does not look like being developed properly. There are controls missing regarding specificity - could the authors, for example, perform quantitative mass spec on cell lysates and thus provide a more unbiased and more quantitative analysis of changes in protein levels? If Arc is the only, or the major, protein affected in this pathway, then their data would be greatly strengthened. If, by contrast, Arc was the only protein they investigated, then the data is weak (especially given the minor effects seen here).

Answer:

For the first part of the comment, we have now included Arc Western from an additional experiment in Extended Data Fig. 7f to support the significant reduction in Arc levels.

As for the second part, we monitored the E6AP and Arc levels as reliable physiological readouts for perturbations in the UBA6-USE1 interaction. Indeed, Arc is not the only and probably not the major component in this cascade as the Reviewer commented. Instead of choosing mass spec experiment to pinpoint on major affected specific protein levels, we chose to tackle UBA6 effects on the survival of neurons from patients with polyalanine expansion mutations (new results in Fig. 4 f,g). This demonstrates the pathological relevance of our findings in an unbiased manner, which we believe is aligned with the Reviewer views on unbiased outputs.

The Arc western blot in Extended Data Fig 7f indeed shows a clear decrease in the Arc levels, which is consistent with the significant decrease observed by microscopy. The phrasing is a bit misleading as the western blot does not show a significant increase as no quantifications are included. The fact that UBA6 overexpression rescues CCHS neurons is interesting but does not show that a reduction Arc steady-state levels is responsible for the reduced viability of CCHS neurons. Does UBA6 overexpression also increase the Arc levels in CCHS neurons? To me it is not clear why they didn't look at this obvious read-out when they performed the rescue experiments anyway. In Fig 10g they show a very clear decrease in Arc levels in CCHS neurons. Are these levels reversed when UBA6 is overexpressed?

Comment:

One of the major shortcomings of this paper remains that mechanistic insight is limited - how do Ala-stretches modulate E1 binding (they are not in the domain normally used for it)? Why are proteins with extended poly-Ala repeats accumulating in the cytoplasm? Are they caught right off the ribosome by quality control pathways or are they exported out of the nucleus? Are they captured by other cytoplasmic proteins? As UBA6 seems to be mainly cytoplasmic, and because the authors suggest that polyA proteins exert their effect onto UBA6, these are important questions to answer. It is currently very difficult to determine whether apparent changes in UBA6 activity or E6AP ubiquitylation are direct or indirect.

Answer:

We have now included mechanistic studies and discussions to address the reviewer questions:

1. how do Ala-stretches modulate E1 binding (they are not in the domain normally used for it)

We can now answer this question and show in Extended Data Fig.3b that the SCCH domain of UBA6, which is involved in E2 specificity, can directly interact with Ala-stretches (please also see our previous elaborated answer on this point).

OK.

2. Why are proteins with extended poly-Ala repeats accumulating in the cytoplasm? Are they caught right off the ribosome by quality control pathways or are they exported out of the nucleus? Are they captured by other cytoplasmic proteins?

This is a relevant question to answer. We have now included a reference in the Discussion supporting the nuclear export mechanism for the polyalanine expansion mutation. Please see Discussion, page 13: "Indeed, the expanded polyalanine domain itself can serve as a nuclear export signal 43". While nuclear export could explain the mislocalization behavior of the mutant protein, it was unclear how cytoplasmic mislocalization is associated with toxicity and potential harmful gain of function. Our model suggests that sequestering the enzyme UBA6 into soluble cytoplasmic polyalanine expansions may represent a deleterious mechanism for such mutations. This is now further supported by the new data in Fig. 4 showing that overexpression of UBA6 in the patient neurons rescues cell death.

I have no concerns here. I think that the authors have convincingly shown that cytosolic mislocalization of polyalanine proteins occurs. Why this happens is indeed interesting but can be addressed in detail in follow-up studies.

3. It is currently very difficult to determine whether apparent changes in UBA6 activity or E6AP ubiquitylation are direct or indirect.

To address this comment, we performed additional new mechanistic experiments (Fig. 1g, Fig. 3f, Extended Data Fig. 3b, and Extended Data Fig. 6 a-e) supporting direct effects of wild-type and expanded polyalanine stretches on UBA6 binding, UBA6 activity and E6AP ubiquitination. We addressed this in Results page 5-6: "Since E6AP is monoubiquitinated and polyubiquitinated by the UBA6-USE1 cascade 13, we further investigated the role of the polyalanine stretch of USE1 in E6AP

ubiquitination by incubating purified E6AP with UBA6, USE1 WT and USE1 mutants (PolyAla and C188A) in vitro. The results indicated that the incubation with the USE1 PolyAla decreased the polyubiquitination of E6AP but not the monoubiquitin conjugate, similarly to the effects of the USE1 C188A catalytically dead mutant (Fig. 1g)".

Results page 7: "Indeed, a biophysical analysis revealed that a peptide of 7 alanine residues interacts directly with the SCCH domain of UBA6, but has significantly weaker binding to the SCCH domain of UBA1 (Extended Data Fig. 3b)".

Results page 8: "Consequently, the interactions between UBA6 and the mutant proteins (PHOX2B +13 Ala, RUNX2 +12 Ala, HOXD13 +10 Ala, and PABPN1 +7 Ala) are measurable in the cytoplasmic but not in the nuclear fraction of the cells (Extended Data Fig. 6 a-d), which reduces the USE1 ubiquitin loading (Extended Data Fig. 6 e)".

Later on in Results page 8: "Our results indicate that recombinant mutant PHOX2B (+ 13 Ala) competes with USE1 for binding to UBA6 in vitro (Fig. 3f)".

OK.

Comment:

It is also unclear whether E6AP is an important target in this pathway, and whether the link to UBA6 is related to disease. To address these questions, the authors would need to implement a functional assay that monitors neuronal output - similar to what Marius Wernig and Tom Suedhoff have done for APP processing in induced neurons, for example. If they see defects in their iPSC derived neurons from patients, they could then ask whether these are rescued by overexpression of UBA6. Alternatively, they could test whether reducing E6AP levels could rescue a neuronal phenotype - although I would be surprised that the minor changes in the levels of an enzyme really have a devastating consequence. Without such mechanistic insight, much of this paper shows correlation, but does not provide evidence for causation.

Answer:

We thank the reviewer for suggesting this experiment, which was also suggested by Reviewer 2. We have now provided new data supporting that the overexpression of UBA6 increases the resilience of the patient neurons to cell death. The new results are presented in Fig. 4f, Fig. 4g and Extended Data Fig. 10i. We addressed this in Results, page 12: "we asked how does UBA6 affect polyalanine expanded proteins-associated toxicity? Because we observed apoptotic nuclear morphology in the CCHS patient neurons (Extended Data Fig. 10d), we next attempted to rescue the cell death in CCHS neurons by their transduction with lentiviruses expressing the cDNA of UBA6 (Fig. 4f, g), which resulted in robust UBA6 expression in both cell body and neurites (Extended Data Fig. 10i). Interestingly, the overexpression of UBA6 significantly reduced the number of CCHS neurons positive to terminal deoxynucleotidyl transferase dUTP nick end labeling (TUNEL) (Fig. 4g), and also decreased the amounts of phosphatidylserine on the extracellular surface as detected by Annexin V binding (Fig. 4f). We can therefore conclude that overexpression of UBA6 in the CCHS neurons rescues them from neuronal death".

OK.

Referee #2

Comment:

In this study, Amer-Sarsour et al. used HEK293T to report that the ubiquitin-activating enzyme 6 (UBA6) binds to the polyalanine domain of the ubiquitin E2 conjugating enzyme 1 (USE1), and that polyalanine-expanded proteins disrupt this interaction. Focusing on one polyalanine expansion disorder - congenital central hypoventilation syndrome (CCHS), the authors then showed an increased association between UBA6 and mutant PHOX2B, and higher levels E3 ubiquitin ligase E6AP in mouse primary neurons and in CCHS patient derived iPSCs. The results seem valid and interesting but not sufficient for publishing in [the journal where the manuscript was originally submitted].

Answer:

We thank the reviewer for the positive comment, highlighting novel aspects from our work. In the current version of the manuscript we performed additional mechanistic experiments and discussions to further support this model including new data in Fig. 1 d-g, Fig. 3 f-h, Fig. 4c, Fig. 4 e-g, Extended Data Fig. 3b, Extended Data Fig. 5b, Extended Data Fig. 6 a-g, Extended Data Fig. 7c, e, Extended Data Fig. 8 a-d, Extended Data Fig. 10 c,h,g,i.

Comment:

The finding of UBA6 and USE1 interaction is not original. Previous studies (cited in this manuscript) have shown that USE1 interacts specifically with the UBA6 enzyme rather than the conventional E1 enzyme, and that Uba6 is required for charging USE1:

Jin, J., Li, X., Gygi, S. P., & Harper, J. W. (2007). Dual E1 activation systems for ubiquitin differentially regulate E2 enzyme charging. *Nature*, 447(7148), 1135-1138. <https://doi.org/10.1038/nature05902>

Lee, P. C., Sowa, M. E., Gygi, S. P., & Harper, J. W. (2011). Alternative ubiquitin activation/conjugation cascades interact with N-end rule ubiquitin ligases to control degradation of RGS proteins. *Molecular cell*, 43(3), 392-405. <https://doi.org/10.1016/j.molcel.2011.05.034>

However, the link between UBA6 and polyalanine tracts in normal proteins and mutant proteins presents new results. This is a well-written and systematically constructed paper. Nevertheless, there are specific issues that need to be addressed as listed below:

Answer:

We agree with the reviewer that the link and mechanism of UBA6 regulation by polyalanine tracts in normal proteins and mutant proteins present new results, and we appreciate the enthusiasm of the reviewer for our work. We have addressed the reviewer's comments below by performing the requested experiments, which further strengthened our model.

The publication Lee et al (2011) is not cited. Not critical.

Major comments

Comment:

The authors claim that disease-causing proteins with polyalanine expansions interact with UBA6. However, they state that isolated polyalanine stretches with NLS did not bind UBA6 and had no apparent effect on USE1 loading. All polyalanine disease-causing proteins are nuclear proteins, and UBA6 is mainly localized in the cytoplasm. The authors also mention that only when these proteins with polyalanine expansions partially mislocalized to the cytoplasm they then could interact with endogenous UBA6. The authors should determine and present the interaction between UBA6 and the nuclear/cytoplasmic fraction for every mutant polyalanine protein.

Answer:

As suggested by the Reviewer, we determined the interaction between UBA6 and the nuclear/cytoplasmic fraction for every mutant polyalanine protein, and the new results are now presented in Extended Data Fig. 6 a-d. Please see also Results page 8: "Consequently, the interactions between UBA6 and the mutant proteins (PHOX2B +13 Ala, RUNX2 +12 Ala, HOXD13 +10 Ala, and PABPN1 +7 Ala) are measurable in the cytoplasmic but not in the nuclear fraction of the cells (Extended Data Fig. 6 a-d)".

OK.

Comment:

It is important to mention that expanded PABPN1 has not been found to mislocalize in the cytoplasm. Hence, it is not appropriate to generalize that these polyalanine mutant proteins interact with endogenous UBA6 in the cytoplasm.

Answer:

To address this issue we have now obtained OPMD patient-derived biopsy that is highly vulnerable to the PABPN1 polyalanine expansion mutation. We could demonstrate that patient-derived cricopharyngeal myotubes and primary fibroblasts showed the cytoplasmic presence of PABPN1. We have also added a reference supporting the nuclear-cytoplasmic shuttling ability of PABPN1. Moreover, our analysis of the OPMD-patient-derived cells further supports our model and the data was now added to Extended Data Fig. 8. Please see Results, page 10: "The cricopharyngeal muscle of OPMD patients is vulnerable to the polyalanine expansion mutations in the PABPN1 gene 30. OPMD patient-derived cricopharyngeal myotubes and primary fibroblasts as well as non-affected control fibroblasts, express PABPN1 both in the cytoplasm and in the nucleus (Extended Data Fig. 8a, b), which could be related to the ability of PABPN1 to shuttle between the nucleus and the cytoplasm 31. Accordingly, OPMD patient-derived fibroblasts have less interaction between UBA6 and USE1 than control fibroblasts (Extended Data Fig. 8 c-d)".

OK.

Comment:

These polyalanine-expanded proteins can be present in either a soluble or insoluble fraction. When studying the ubiquitination of these mutant proteins it is of importance to separate the soluble and insoluble fractions efficiently. Aggregated proteins are ubiquitinated differently when compared to their soluble counterparts. The authors should perform the soluble/insoluble assay and compare the interaction between UBA6 with soluble polyalanine mutant proteins versus insoluble counterparts. Ubiquitin is involved in aggregate formation of expanded polyalanine. It is important to test whether UBA6-USE1 co-localize with polyalanine aggregates or not.

Answer:

We thank the Reviewer for this point and thus we performed soluble or sarkosyl-insoluble fractionation assays for various cell types expressing wild-type and expanded polyalanine proteins to address this issue. In addition, we also tested if UBA6 co-localizes with polyalanine aggregates. Our results suggest that the soluble fraction of the polyalanine expansion is sufficient to exert effects on UBA6. The new data was now added in Extended Data 5b, Extended 7c, and Extended Data 10c. Please see results page 8: "Biochemical analysis of the GFP-19Ala could not detect a sarkosyl-insoluble fraction, suggesting that the soluble polyalanine stretches can interact with UBA6 (Extended Data Fig. 5b)".

Results page 9 regarding ectopic expression of expanded polyalanine proteins with lentiviruses in primary neurons: "While most of the wild type PHOX2B could be detected in the nucleus, mutant PHOX2B (+13 Ala) is also present in the cell body and in neurites, where the GFP fluorescence imaging and biochemical analysis revealed both non-aggregated and aggregated patterns (Extended Data Fig. 7b, c)". Later in Results page 9: "An analysis of UBA6 abundance revealed that UBA6 is predominantly associated with the non-aggregated forms of PHOX2B (Extended Data Fig. 7e)".

Results page 11 related to endogenous expression of polyalanine-expanded proteins in patient neurons: "Analysis of the PHOX2B localization revealed a decrease in the nuclear fraction of PHOX2B in the mutant 20/25 and 20/27 neurons, as compared to healthy control neurons (Extended Data Fig. 10 a,b). The cytoplasmic PHOX2B is soluble, and predominantly perinuclear without visible aggregates (Extended Data Fig. 10 c,d). In addition, there was an increased association between UBA6 and PHOX2B in the CCHS patient neurons (Fig. 4b), accompanied by a decrease in the interaction between UBA6 and USE1 (Fig. 4c)".

OK

Comment:

The authors used one biochemical method (i.e. co-immunoprecipitation) to determine the interaction between UBA6 and USE1 and the interaction between UBA6 and various disease-causing proteins with polyalanine expansion mutations. It is important to confirm this interaction using another method (such as two-hybrid assays) where the proteins are more likely to be in their native confirmation.

Answer:

To address this comment, we performed quantitative FRET-FLIM experiments to determine the endogenous cellular interaction in the native conformation between USE1 and UBA6 in different cell types, either wild-type cells or from patients with polyalanine expansion mutations. The new Data is presented in Fig. 1d, Fig. 4c, Extended Data Fig. 8 c-d.

Please see results, page 5: "We next mutated the endogenous polyalanine stretch of USE1 by generating a knockout HEK293T cell line harboring a deletion of the polyalanine stretch in the UBE2Z alleles (USE1 PolyAla KO). The endogenous USE1-UBA6 interaction in these cells was detected with Förster resonance energy transfer (FRET) based fluorescence lifetime imaging microscopy (FLIM). The results indicated less binding between UBA6 and USE1 in the USE1 PolyAla KO cells compared to control cells (Fig. 1d)".

Results, page 10 (using FRET-FLIM): "Accordingly, OPMD patient-derived fibroblasts have less interaction between UBA6 and USE1 than control fibroblasts (Extended Data Fig. 8 c-d)". Results page 11 (using FRET-FLIM in Fig. 4c): "In addition, there was an increased association between UBA6 and PHOX2B in the CCHS patient neurons (Fig. 4b), accompanied by a decrease in the interaction between UBA6 and USE1 (Fig. 4c)".

OK.

Comment:

How does UBA6 affect polyalanine expanded proteins-associated toxicity? In the conclusion's section, the authors mention that the inhibition of UBA6 by sequestering the enzyme into cytoplasmic mislocalized polyalanine expansions represent a deleterious mechanism for such mutations. The authors should perform should investigate whether overexpression of recombinant UBA6 would restore polyubiquitylation activity by USE1 for polyalanine expanded proteins and rescue cell death.

Answer:

We highly appreciate this comment to overexpress UBA6 and investigate effects on cellular defects caused by the mutant proteins, which was also suggested by Reviewer 1. We addressed the pathological relevance of our findings by focusing on the analysis of cell death. We have now provided new data supporting that the overexpression of UBA6 increases the resilience of the patient neurons to cell death. The new results are presented in Fig. 4f, Fig. 4g and Extended Data Fig. 10i. Please see results, page 12: "we asked how does UBA6 affect polyalanine expanded proteins-associated toxicity? Because we observed apoptotic nuclear morphology in the CCHS patient neurons (Extended Data Fig. 10d), we next attempted to rescue the cell death in CCHS neurons by their transduction with lentiviruses expressing the cDNA of UBA6 (Fig. 4f, g), which resulted in robust UBA6 expression in both cell body and neurites (Extended Data Fig. 10i). Interestingly, the overexpression of UBA6 significantly reduced the number of CCHS neurons positive to terminal deoxynucleotidyl transferase dUTP nick end labeling (TUNEL) (Fig. 4g), and also decreased the amounts of phosphatidylserine on the extracellular surface as detected by Annexin V binding (Fig. 4f). We can therefore conclude that overexpression of UBA6 in the CCHS neurons rescues them from neuronal death".

A rescue of cell viability is quite indirect and could mechanistically be very different. Analysis of the E6-AP levels and/or Arc levels would be more convincing.

Comment:

The authors should investigate the interaction between polyalanine expanded proteins and UBA6 in other patients' cells to demonstrate if the same phenomenon occurs as in CCHS iPSCs.

Answer:

We have now included the analysis of cellular material derived from OPMD patients in Extended Data Fig. 8, which shows effects on UBA6 with a similar trend as in the CCHS-derived cells.

OK.

Minor comments

Comment:

In Figure 1C, the decrease level of ub-USE of the analyzed proteins following USE1 Δ PolyAla mutant is not really evident and does not match the reported quantification. There is no difference between lane 1 and lane 2. The authors must provide better quality immunoblotting and quantification results in line with the images.

Answer:

We thank the reviewer for spotting this. We have now included an additional blot analysis from another experiment, which better reflect the quantification. The data is now presented in Extended Data Fig.2a.

OK.

Comment:

In the Extended Data Fig. 4., the title needs to be changed from: Polyalanine expansion mutations regulate ... to isolated polyalanine stretches regulate....

The level of E6AP was not measured in cells expressing expanded polyalanine proteins.

Answer:

We thank the Reviewer for this comment. In Extended Data Fig. 4 in previous version, which is now Extended Data Fig. 5, we also presented analysis of solubility of the polyalanine stretches. Therefore, as suggested by the Reviewer we now changed the title to read: "Soluble isolated polyalanine stretches..."

OK.

Comment:

For some of the confocal microscopy images, the authors need to provide micrographs that are at lower magnification so that the reader would see more cells in the field that would be consistent with the quantifications (e.g., Extended Data Fig. 3 and Figure 4d).

Answer:

We thank the Reviewer for this comment. We have now provided lower magnification micrographs to capture more cells. Please see the micrographs that the Reviewer requested as well as additional ones in Fig. 1d, Fig. 4c, Fig. 4 f,g, Extended Data Fig. 4, Extended Data Fig. 8c, Extended Data Fig. 10 e, i.

OK

Comment:

In Figure 3h, the authors should explain the reduced level of PHOX2B that is observed in lane 3 (primary neurons expressing GFP-PHOX2B+13Ala). This could produce the increased levels of E6AP observed in this blot.

Answer:

We appreciate this comment. The reduced levels of the PHOX2B+13Ala observed in lane 3 (primary neurons expressing GFP-PHOX2B+13Ala) is probably because of an aggregated fraction which could not be resolved properly in the lysis buffer. We have now included data to support this in the insoluble fractionation biochemical assay in Extended Data Fig.7c.

OK.

Comment:

In Figure 4d, the authors should count more than 40 neurons for immunostaining of Arc and TUB β 3 in autonomic neurons from control and CCHS patients.

Answer:

As suggested by the reviewer, we have now included additional analysis of more neurons, which supports the trend we observed. The Data is presented in Extended Data Fig. 10 e-f.

OK.

Comment:

In the Methods (Bioinformatics analysis), the authors wrote: Prosite (ref pubmed id 23161676) was used to scan for alanine residues motifs with between 6 and 10 continuous alanine residues in the BLAST results. They should add: A search for proteins containing polyalanine stretches in the ubiquitin cascades. They performed Blast only in the ubiquitin pathway.

Answer:

The sentence regarding the Bioinformatics analysis was revised accordingly.

OK.

Comment:

In the Extended Data Fig. 1, generated from the Blast, there are 13 ubiquitin enzymes all contain alanine residues (e.g., ZFP91 contains 9 alanine residues). The authors should explain the rationale behind choosing USE1 and not another ubiquitin enzyme?

Answer:

In our Bioinformatics analysis, USE1 was the only E2 enzyme containing polyalanine while the others have E3-ubiquitin ligase activities. Since E2s are likely to work with more than one E3 enzyme and there are currently nine different diseases with polyalanine expansion mutation in different proteins, we reasoned that the investigation of upstream regulation of the ubiquitin cascade might have a broader effect.

OK.

Referee #3

Comment:

Amer-Sarsour et al. investigated the mechanism of polyalanine expansion diseases. Their hypothesis is that an E1 ubiquitin-activating enzyme UBA6, recognizes a polyalanine stretch in the ubiquitin-conjugating E2 enzyme UBE2Z/USE1, and the result of various polyalanine expansion gene products can compete for the binding and therefore negatively affects protein degradation. The authors propose that the competition for UBA6 provides a shared mechanism that mediates the neurodegeneration of polyalanine expansion diseases. While authors presented an interesting hypothesis, almost all of the experimental evidence are from overexpression studies and some key data are very weak at best. The evidence presented are very far from sufficient to prove this model.

Answer:

We thank the Reviewer for appreciating our suggested mechanism, and the comments to strengthen our model. Through additional mechanistic experiments we now provide stronger evidence at the endogenous levels in patient cells and non-affected material. The new data is now presented in Fig. 1 d-g, Fig. 3 f-h, Fig. 4c, Fig. 4 e-g, Extended Data Fig. 3b, Extended Data Fig. 5b, Extended Data Fig. 6 a-g, Extended Data Fig. 7c, e, Extended Data Fig. 8 a-d, Extended Data Fig. 10 c,h,g,i.

Major issues

Comment:

The data that support this model came from overexpression of 293T cells which cannot differentiate direct vs. indirect effects. The computational derived model (Figure 2) predicts the direct interaction of polyalanine with the binding site in polyalanine length specific manner. This prediction should be tested biochemically using in vitro binding studies including the measurement of binding affinities of various polyalanine stretches.

Answer:

We thank the reviewer for the comment, which was also suggested by Reviewer 1, because now we can show direct biophysical interaction of the peptide with 7 alanine residues (7xAla peptide) with the SCCH domain (known to determine E2s specificities). The Reviewers correctly requested to show in a biophysically manner the direct interaction of the USE1 poly-Ala with the SCCH domain. To address this, we purified the SCCH domain from E. coli and used a 7xAla peptide (unfortunately the company could not synthesize longer polyAla peptides), which were tested in MST binding assay. In this assay, the peptide was fluorescently labeled. Different E1s bind the E2 core and present specificity via other regions. It is important to appreciate that the specificity is achieved by weak interactions that are difficult to measure and quantify in terms of KD. Indeed, we could show, and now add to the paper, the binding by these measurements. However, we could not determine the KD as affinity for this interaction is ultra-weak. Careful examination of the MST binding shows that we are far from saturation even at 200 micromolar of the domain. Also, it worth to mention that the 7xAla peptide is insoluble at high concentrations, thus it was kept at low concentrations to allow solubility. Most importantly, our data clearly shows a significant binding differences between UBA1 and UBA6. The new data was added in Extended Data Fig.3b. Please see Results page 7: "Indeed, a biophysical analysis revealed that a peptide of 7 alanine residues interacts directly with the SCCH domain of UBA6, but has significantly weaker binding to the SCCH domain of UBA1 (Extended Data Fig. 3b)".

OK.

Comment:

The authors present data that suggest USE1 2A-2R mutation affecting its ubiquitin-loading which is partially UBA6 dependent (Figure 1). Does this USE1 2A-2R mutation affect its interaction with UBA6?

Answer:

We have now generated cells with mutation in the endogenous polyalanine domain of USE1. Analyzing the data at the endogenous level with no overexpression was suggested by Reviewer 3 in the first comment and also by Reviewer 1. While we introduced different alterations in the polyalanine stretch of USE1 (2A-2R and a deletion mutation), all of which presented reduced USE1 ubiquitin loading, Reviewer 1 requested in depth analysis of a more conservative mutation. This is why we chose to further investigate the polyalanine deletion mutation, since this will avoid introducing potential non-relevant effects of alternative residues. We used a quantitative FLIM-FRET approach to measure the USE1-UBA6 interaction, which is more physiologically relevant and can preserve the native conformation as requested by Reviewer 2. The new data is presented in Fig. 1d. Please see results, page 5: "We next mutated the endogenous polyalanine stretch of USE1 by generating a knockout HEK293T cell line harboring a deletion of the polyalanine stretch in the UBE2Z alleles (USE1 PolyAla KO). The endogenous USE1-UBA6 interaction in these cells was detected with Förster resonance energy transfer (FRET) based fluorescence lifetime imaging microscopy (FLIM). The results indicated less binding between UBA6 and USE1 in the USE1 PolyAla KO cells compared to control cells (Fig. 1d)".

As I pointed out earlier, the deletion of the alanine stretch may have indirect effects on the structure of USE1 and therefore it cannot be concluded that it directly binds. I am a bit surprised that they did not do what the reviewer asked. They have the expression constructs to do a co-IP and the reviewer does not request any in vitro experiments. It would be relatively easy to do it following the same experimental setup as for Fig 1b and pulldown the FLAG-USE1 and FLAG-USE1 2A>2R. Having said that the 2A>2R mutations could also have indirect effects and does not conclusively show that the alanine stretch is responsible for the interaction.

Comment:

Can polyalanine and other polyalanine proteins, e.g. PHOX2B, RUNX2 and HOXD13, PABPN1 etc, affect the ubiquitin-loading of USE1? In Extended Data Figure 4a, GFP-19Ala only marginally, if there is any, decreased the loading of ubiquitin onto USE1.

Answer:

We appreciate this comment. We can now show that the other mutant polyalanine proteins decreased the ubiquitin loading of USE1. The new results are in Extended Data Fig. 6e. We addressed this in the Results page 8: "Consequently, the interactions between UBA6 and the mutant proteins (PHOX2B +13 Ala, RUNX2 +12 Ala, HOXD13 +10 Ala, and PABPN1 +7 Ala) are measurable in the cytoplasmic but not in the nuclear fraction of the cells (Extended Data Fig. 6 a-d), which reduces the USE1 ubiquitin loading (Extended Data Fig. 6 e)".

OK.

Comment:

Does overexpression of polyalanine-expanded proteins disrupt interaction between endogenous UBA6 and USE1? In addition, in patient-derived cells or tissues which contain endogenous polyalanine-expanded proteins, is the interaction between endogenous UBA6 and USE1 destroyed or reduced? Co-IP assays should be performed in these cell or tissue models.

Answer:

We have now performed endogenous interaction analysis between UBA6 and USE1 in patient-derived cells. We analyzed autonomic neurons with polyalanine expansion mutation in PHOX2B from CCHS patients, and also included the analysis of OPMD patient-derived primary fibroblasts with the polyalanine expansion mutation in PABPN1. We used FRET-FLIM to measure the interaction in a quantitative way. The new data is in Fig. 4c, and Extended Data Fig. 8 c-e. Please see also Results, page 10: "Accordingly, OPMD patient-derived fibroblasts have less interaction between UBA6 and USE1 than control fibroblasts (Extended Data Fig. 8 c-d)".

Results page 11: "In addition, there was an increased association between UBA6 and PHOX2B in the CCHS patient neurons (Fig. 4b), accompanied by a decrease in the interaction between UBA6 and USE1 (Fig. 4c)".

OK.

Comment:

The authors observed a correlation between overexpression of PHOX2B-13Ala and upregulation of E6AP (and downregulation of Arc). However, the underlying mechanism is unclear.

Answer:

We thank the reviewer for this comment and we have now performed additional mechanistic experiments to support this correlation. We provided new data in Fig. 3g, Fig. 4e and Extended Data Fig. 6 f, g on the degradation rates of E6AP by PHOX2B-13Ala. Please see Results, page 8-9: "To test the effects on E6AP degradation, we measured the stability and ubiquitination of E6AP in mutant PHOX2B expressing HEK293T cells. Mutant PHOX2B increases the levels of E6AP due to its stabilization in HEK293T cells (Fig. 3g, Extended Data Fig. 5c, Extended Data Fig. 6f). However, mutant PHOX2B does not increase E6AP mRNA levels, indicating that the increased E6AP levels are not related to transcriptional effects (Extended Data Fig. 6g). This correlates with a decrease in Lys48-linked polyubiquitination of E6AP (Fig. 3h, Extended Data Fig. 6h), suggesting that proteasome-mediated degradation of E6AP is inhibited by mutant PHOX2B in the cells".

Later on in Results page 12: "In accordance with the mouse data, the levels of E6AP protein are higher in the patient neurons (Fig. 4d), although no change in E6AP mRNA was detected (Fig. 4e)".

Since Arc levels are negatively regulated by E6AP at the transcriptional and post-translational levels (PMID: 20211139, 23671107), we could not perform a proper degradation assays for Arc, and instead we showed Arc steady state levels by western blots and immunostaining in the primary and autonomic neurons.

I don't see why regulation of Arc levels at transcriptional and post-transcriptional level would preclude analysis of an effect of E6AP on the degradation of Arc. On the contrary, this would render conclusions based on steady-state levels inconclusive when it comes to a possible effect of E6-AP on Arc degradation and emphasizes the need to look at Arc turnover instead of steady-state levels.

Comment:

In Figure 3h and Extended Data Figure 4a, only marginal or mild upregulation of E6AP is observed when PHOX2B+13Ala is overexpressed, and the authors claimed that PHOX2b+13Ala inhibits E6AP degradation. To obtain such a conclusion, one should compare the half-lives of E6AP in the presence and absence of polyalanine expansion. Thus, other experiments, such as cycloheximide chase or pulse chase assays, are needed. In addition, to rule out transcriptional effects, the authors could also examine E6AP mRNA level both in overexpressed and patient-derived cell models. Similar experiments should be performed for Arc as well.

Answer:

Please see our answer above. We performed the requested cycloheximide chase experiments for E6AP and ruled out transcriptional effects (new data in Fig. 3g, Fig. 4e and Extended Data Fig. 6 f, g). Please also note our answer above regarding the difficulties in performing such chase assays for Arc. In addition, we added new results in Fig. 1e,f determining the half-life of E6AP in cells mutated in the endogenous polyalanine domain of USE1. Please see Results, page 5: "In control HEK293T cells, a cycloheximide chase experiment revealed that E6AP has an apparent half-life of approximately 10-14 hours (Fig. 1e), which is consistent with previous reports in cultured cells 13. In contrast, E6AP was stabilized in the USE1 PolyAla KO cells and showed a decrease in Lys48-linked polyubiquitination (Fig. 1e, Fig. 1f), suggesting that proteasome-mediated degradation of E6AP is inhibited in the USE1 PolyAla KO cells". We believe these additional experiments at the endogenous level with no overexpression further support the mechanisms we are proposing.

OK.

Comment:

In Page 2, the authors claimed that "Aberrations in this polyalanine stretch reduced ubiquitin transfer to ... downstream target, the E3 ubiquitin ligase, E6AP." However, result in Fig. 3f is not sufficient to prove this conclusion. It has been shown that UBA6-USE1 ubiquitinates E6AP in vitro (PMC3640669). The authors should reconstitute the reaction and test if the polyalanine stretch in USE1 is required for in vitro ubiquitination activity. In addition, the authors should test if the recombinant polyalanine-stretch containing proteins (e.g., PHOX2B+13Ala) directly inhibits E6AP ubiquitination using the in vitro assay.

Answer:

We agree with the Reviewer. We reconstituted the UBA6-USE1 ubiquitination reaction of E6AP and tested the effect of the polyalanine stretch in USE1. We added the new result in Fig.1g. Please see also results page 5-6: "Since E6AP is monoubiquitinated and polyubiquitinated by the UBA6-USE1 cascade 13, we further investigated the role of the polyalanine stretch of USE1 in E6AP ubiquitination by incubating purified E6AP with UBA6, USE1 WT and USE1 mutants (PolyAla and C188A) in vitro. The results indicated that the incubation with the USE1 PolyAla decreased the polyubiquitination of E6AP but not the monoubiquitin conjugate, similarly to the effects of the USE1 C188A catalytically dead mutant (Fig. 1g)". Moreover, instead of performing the same ubiquitination experiment but with PHOX2B+13Ala, we designed a competition experiment with recombinant PHOX2B+13Ala to demonstrate that the mutant polyalanine expanded protein directly interacts with UBA6 and interfere with USE1 binding. We believe this would better answer the Reviewer's comment regarding the underlying mechanism. The new data is presented in Fig. 3f. Please see also Results page 9: "Our results indicate that recombinant mutant PHOX2B (+ 13 Ala) competes with USE1 for binding to UBA6 in vitro (Fig. 3f)".

OK

Comment:

In addition to E6AP, Shank3 has been also reported to enrich in UBA6-deficient cells. Did the expression of polyalanine-expanded proteins lead to upregulation of Shank3? In patient-derived cells, is the Shank3 upregulated?

Answer:

We highly appreciate this suggestion and thus we have attempted to analyze Shank3 levels in the patient autonomic neurons to see effects at the endogenous level. Unfortunately, we could not detect Shank3 expression in the human iPSC-derived autonomic neurons. We added Data addressing this issue in Extended Data Fig. 10h. Please see also Results page 12: "In addition to E6AP, the UBA6-USE1 cascade has been shown to regulate the levels of the synaptic protein shank3 13. We could not detect endogenous expression of shank3 in the human autonomic neurons (Extended Data Fig. 10 h), thereby precluding the analysis of UBA6-USE1 effects on shank3 in these neurons". In order to address other UBA6 outputs relevant to the pathology of polyalanine expansion diseases, we have now provided new data supporting that the overexpression of UBA6 increases the resilience of the patient neurons to cell death. The new results are presented in Fig. 4f, Fig. 4g and Extended Data Fig. 10i. Please see results, page 12: "we asked how does UBA6 affect polyalanine expanded proteins-associated toxicity? Because we observed apoptotic nuclear morphology in the CCHS patient neurons (Extended Data Fig. 10d), we next attempted to rescue the cell death in CCHS neurons by their transduction with lentiviruses expressing the cDNA of UBA6 (Fig. 4f, g), which resulted in robust UBA6 expression in both cell body and neurites (Extended Data Fig. 10i). Interestingly, the overexpression of UBA6 significantly reduced the number of CCHS neurons positive to terminal deoxynucleotidyl transferase dUTP nick end labeling (TUNEL) (Fig. 4g), and also decreased the amounts of phosphatidylserine on the extracellular surface as detected by Annexin V binding (Fig. 4f). We can therefore conclude that overexpression of UBA6 in the CCHS neurons rescues them from neuronal death".

Valid explanation for not being able analysing Shank3. However, the rescue of cell viability does not address the question and is not really relevant in this respect.

Other specific points

Comment:

Figure 1d, original immunoblots should be shown.

Answer:

We have now included the blots for the quantification in Extended Data Fig. 2b.

OK.

Comment:

Result in Figure 1e is not so reliable, because the expression of FLAG-USE1 is not even in all samples (input). The authors could first affinity purify the UBA6-USE1 complex by the anti-UBA6-immobilized agarose, then elute the USE1 from the UBA6 using 19-Ala peptide and control peptide (or recombinant GFP-19Ala).

Answer:

Thank you for this suggestion. We have now performed similar experiment to the one the Reviewer suggested but included recombinant PHOX2B+13 Ala instead of the isolated polyalanine stretch to better link it to disease. Indeed, the levels of the FLAG-USE1 and HA-UBA6 were even in the input, and the pulldown of UBA6 resulted in a significantly decreased amount of

bound USE1 in the presence of the disease protein. The new data is presented in Fig. 3f.

OK

Comment:

Figure 2b, the authors determined the interaction between overexpressed USE1 and UBA6. To further validate the result, endogenous interaction of the two proteins should be measured (i.e., in cells without transfection of plasmid).

Answer:

This is an important point. We have added new data analyzing the interaction of endogenous USE1 with UBA6 in cells with no overexpression. This includes USE1 delta Ala KO cells, CCHS patient autonomic neurons, and OPMD patient-derived fibroblasts. The new results are presented in Fig. 1d, Fig. 4c, and Extended Data Fig. 8 c-d.

OK

Comment:

Figure 3a-3e, these co-IP experiments lack key controls. Mock IPs with IgG for samples expressing polyalanine stretch-containing preys are required.

Answer:

We have now performed additional co-IP experiments of endogenous UBA6 with IgG controls from nuclear and cytoplasmic fractions of cells expressing different polyalanine-expanded proteins. The new data is presented in Extended Data Fig. 6 a-d.

OK

Comment:

Figure 3f, it seems that the IP experiment was performed under native condition, according to the Methods (page 21). To determine the ubiquitination status of E6AP, samples should be denatured by SDS (plus boiling) before IP to disrupt any non-covalent binding.

Answer:

We apologize for this confusion since all experiments determining the ubiquitination status of E6AP were analyzed under SDS conditions plus boiling with reducing agents. We made it clearer in the Methods section.

OK.

Comment:

Figure 4c, immunoblots for Arc in control and patient-derived cells should be shown.

Answer:

We have now included blots for Arc levels in control and CCHS patient neurons in Extended Data Fig. 10 g.

OK.

Referee #2:

The initial reviewers from the original submission all found the preliminary data from Amer-Sarsour et al. to be intriguing but insufficient to make any strong conclusions about whether polyalanine expansions found in certain neurological disease proteins could interfere with UBA6 ubiquitin conjugation and thereby contribute to these diseases. The idea that a short poly-Ala tract in USE1, the E2 that works with the UBA6 E1, is needed for full binding to the E1 and is interfered with by the poly-Ala expansion proteins is attractive. However, I agree with the reviewers that the initial submission did not make a strong enough case for this.

I feel the authors have now done an excellent job in addressing the original reviewers' concerns with a number of new results. The effects are still often quite subtle but on the whole are more convincing. Amer-Sarsour et al. have now deleted the six-Ala tract in USE1 and shown that this modestly reduces E2 charging rate by UBA6 in vitro. They have confirmed a reduction in endogenous poly-Ala delta mutant USE1-UBA6 binding in cells using both co-immunoprecipitation and FLIM-FRET. While they could not measure a Kd for the interaction, they could show interference of USE1-UBA6 association in cells by excess poly-Ala (GFP-19Ala), and in vitro binding of the UBA6 SCCH domain with an Ala heptamer (UBA1 binding was much weaker). The new data on E6AP stability and K48 ubiquitination are now much stronger (I don't understand the source of the persistent E6AP monoubiquitination in Fig. 1g though; a sentence on this would be helpful). Other results support the relevance of disease proteins with poly-Ala tracts through interfering with UBA6-USE1 binding and charging, and the apoptotic cell death seen in CCHS patient neurons could be suppressed by overexpressed UBA6.

In summary, the current manuscript makes a compelling case that the short Ala tract in USE1 is significant for its binding to and charging by UBA6, that this binding can be inhibited by poly-Ala tracts directly, and that poly-Ala expansions in patient proteins can impair UBA6-USE1 function in neurons. E6AP does appear to be an important downstream target in this pathway as well. I

am in favor of publication in the EMBO Journal with just minor revisions:

-Should mention in the Intro that UBA6 also activates the ubiquitin-like protein FAT10.

-For Fig. 3g, the better control would seem to be PHOX2B w/o the 13 Ala tract, not the empty vector.

-There are typos here and there that will need to be corrected.

Response to Reviewers' comments

We highly appreciate the positive feedback of the Advisors. We addressed their comments experimentally with the following new results:

Referee 1 points #2 (regarding the Arc half-life): Fig. 4G (see our answer #3 to Referee 1).

Referee 1 points #3 (regarding E6-AP and Arc levels in rescue experiments): Fig EV4D-F (see our answer #4 to Referee 1).

Referee 2 points #2 (regarding better control i.e. PHOX2B without the Ala tract in cycloheximide experiment of E6AP): Fig. 4F (see our answer #4 to Referee 2).

Furthermore, we added to the paper additional new results in Fig EV1C and Fig EV3G we believe to supporting our model.

Referee 1

Comment 1:

Below you find my comments on each of the issues in red font. I have limited myself to only analyzing whether the concerns of the reviewers have been addressed and refrained from bringing up other issues. Overall, I think that the authors have done a thorough job in addressing the concerns. However, there are a few points that haven't been fully addressed.

Answer 1:

We thank the Referee for the positive feedback and highly appreciate his/her valuable suggestions to improve our paper, which we have now addressed both experimentally and by revising the text.

Comment 2:

Two reviewers brought up concerns about the alanine -> arginine substitutions that the authors use to support their model that the alanine stretch in USE1 is responsible for the binding to UBA6. Reviewer #1 proposes to test this with conservative mutations as the charge of the arginine residues could have been a problem. Instead, the authors opted for using mutants in which the entire alanine stretch was deleted (not sure why, I don't agree with their argument that would be a more conservative mutation that substituting the residues). However, such a deletion is also quite dramatic and may indirectly affect the structural integrity and compromise binding of UBA6 to another part of the USE1. I think that given the overall picture of the provided data, it is likely, though not conclusively shown, that the alanine repeat is responsible for the binding. Having said that even performing these experiments with conserved mutations in USE1 will not exclude indirect effects. I personally don't think that additional experimentation is required but am a bit surprised by their argumentation.

Answer 2:

We thank the referee for bringing this issue. We appreciate that the referee does not ask for further experiment here. According to this suggestion, we now highlight the limitations of the experimental approach of introducing mutations in the polyalanine domain of USE1. Results,

page 6: “Mutations in the polyalanine domain of USE1 may indirectly affect the structural integrity and compromise binding of UBA6 to another part of USE1. We therefore examined whether isolated polyalanine stretches (19 Ala residues) interact with UBA6. We transfected HEK293T cells with GFP-19Ala, and monitored the binding to UBA6, or the possible competition with USE1.”. This would fit with the referee views where he/she later wrote that one could argue that in combination with the other data a coherent picture emerges that supports that the alanine stretch is responsible for the binding. In addition, we can now show and present new data in **Fig EV1C** showing decreased USE1 binding to UBA6 in the USE1 Δ PolyAla KO cells via an immunoprecipitation experiment. This would strengthen the FLIM-FRET data at the endogenous protein level.

Comment 3:

My interpretation of the comments of Reviewer #1 is that this reviewer is wondering if the small changes in E6-AP are sufficient to have an effect on the half-life of its substrates. The authors address this by analyzing in detail the half-life of E6-AP, which they do properly but which I don't think is the actual point. Instead, they should have looked at the half-life of Arc. In response to comments of Reviewer #3, they state that the effect on Arc half-life cannot be assessed because E6-AP levels have an effect on both transcription and degradation of Arc. I cannot follow their reasoning. If E6-AP has an effect on transcription and translation, this would argue that conclusions about effects on Arc degradation can only be based on the half-life of Arc and not on the steady-state levels (which the authors do). Besides that, I don't understand why an effect of E6-AP on transcription would hinder the authors from analyzing the half-life in a turnover experiment. The initial levels at the start point may be different but one can compensate for that in a turnover experiment.

Answer 3:

We appreciate this comment, which we have now addressed experimentally. We performed the turnover experiment of Arc in primary neurons, which supports our model. Since E6AP has a longer half-life than Arc, we used different time points in the cycloheximide chase experiments. The new data was added to **Fig 4 F, G** and to the Results, page 10: “To measure the effects of mutant PHOX2B on the stability of E6AP and Arc, we performed cycloheximide chase experiments in transduced neurons expressing WT or mutant PHOX2B (+13Ala). While E6AP levels remained stable in the mutant PHOX2B-expressing neurons, Arc was more rapidly degraded (Fig 4F, G)”. Note that the initial levels of WT and mutant PHOX2B detected in the gel are different (mutant PHOX2B seems to be lower) possibly because of insoluble/aggregated fraction of the mutant protein, which could not be properly resolved. We have previously addressed this in a comment of the previous reviewers and showed supporting data for it in **Fig 4C**.

Comment 4:

The authors included rescue experiments in which they show that ectopic expression of UBA6 can improve survival of CCHS neurons. This is interesting but also a very indirect readout for the model. Overexpression of UBA6 may improve cell viability by very different mechanisms

than the model proposed by the authors. I don't understand why they didn't use the occasion to analyze in these rescue experiments if it restored E6-AP and Arc levels. I am even more surprised as this seems to be what Reviewer #3 is asking for. It would be a matter of taking the lysates of the neurons from the rescue experiments and probe for E6-AP and Arc.

Answer 4:

We agree with the Referee that analyzing E6AP and Arc levels in the UBA6 rescue experiment can provide additional support for our model. The referee assumed we have the lysates of the patient neurons from the cell-death rescue experiment that we can probe for E6AP and Arc levels. However, these experiments looking at cell-death markers were performed by immunohistochemistry and not by analyzing cell lysates. Thus, in order to probe for E6AP and Arc levels, we would have needed to generate new CCHS patient neurons: recover the iPSCs, new differentiations to sympathetic progenitors and to autonomic neurons following by UBA6 transduction. This can take significantly longer time to analyze compared to setting up the experiments in mutant PHOX2B expressing mouse primary neurons that are transduced with UBA6 for rescue. In order to do the requested experiments in a timely manner, and after discussing the revision plan with the Editor, we now provide these rescue experiments performed in the mouse primary neuron system. The new data is now shown in **Fig EV4D-F** and was added to the results, page 10-11: *“These alterations appear to be dependent on UBA6, because overexpression of UBA6 in the mutant PHOX2B-expressing neurons decreases E6AP and increases Arc levels (Fig EV4D-F)”*. Since Arc levels are indirectly regulated by UBA6 and can be affected by transcriptional regulation, we utilized immunofluorescence of cycloheximide-treated neurons to measure Arc directly in the UBA6 cDNA transduced neurons. We found this approach to be more accurate for Arc rescue experiment compared to analysis of cell lysates.

Comment 5:

This comment is regarding the Basu et al., Cell 2020 paper in the Discussion: “It is a bit confusing though that they cite it in relation to PHOX2B whereas the cited paper deals with HOXD13”.

Answer 5:

We thank the reviewer for detecting this confusion, and we have now revised the text accordingly in the Discussion, page 14: *“These genetic studies suggest the possibility of additional disease contributing mechanisms besides partial loss of PHOX2B function in the nucleus and alterations in transcriptional programs by polyalanine expansion mutations, as was recently described for HOXD13 (Basu, Mackowiak et al., 2020)”*.

Comment 6:

The publication Lee et al (2011) is not cited. Not critical.

Answer 6:

We thank the reviewer for noticing this and have now added the paper citation of Lee et al (2011).

Referee 2

Comment 1:

I feel the authors have now done an excellent job in addressing the original reviewers' concerns with a number of new results. The effects are still often quite subtle but on the whole are more convincing. Amer-Sarsour et al. have now deleted the six-Ala tract in USE1 and shown that this modestly reduces E2 charging rate by UBA6 in vitro. They have confirmed a reduction in endogenous poly-Ala delta mutant USE1-UBA6 binding in cells using both co-immunoprecipitation and FLIM-FRET. While they could not measure a Kd for the interaction, they could show interference of USE1-UBA6 association in cells by excess poly-Ala (GFP-19Ala), and in vitro binding of the UBA6 SCCH domain with an Ala heptamer (UBA1 binding was much weaker). The new data on E6AP stability and K48 ubiquitination are now much stronger (I don't understand the source of the persistent E6AP monoubiquitination in Fig. 1g though; a sentence on this would be helpful). Other results support the relevance of disease proteins with poly-Ala tracts through interfering with UBA6-USE1 binding and charging, and the apoptotic cell death seen in CCHS patient neurons could be suppressed by overexpressed UBA6.

Answer 1:

We thank the Referee for the positive feedback. As for the persistent E6AP monoubiquitination in Fig. 1G this may reflect an E2-independent mono-ubiquitination of E6AP. This is an interesting observation that can be further investigated in future research. We have found that ubiquitin binding domain (UBD) containing proteins, such as Rpn10 may be directly ubiquitinated by E1~Ub due to the non-covalent interaction of the loaded Ub with the UBD. Similar phenomenon was demonstrated for E3-independent ubiquitination of UBDs by Dikic and co-workers (E3-Independent Monoubiquitination of Ubiquitin-Binding Proteins Hoeller et al. Mol. Cell 2007). As suggested by the Referee, we now added the following sentence addressing this point in the manuscript. Results page 6: “*It is plausible that under these experimental conditions, E6AP undergoes direct monoubiquitination by E1 as demonstrated previously for other ubiquitin binding domain containing proteins (Hoeller, Hecker et al., 2007)*”.

Comment 2:

In summary, the current manuscript makes a compelling case that the short Ala tract in USE1 is significant for its binding to and charging by UBA6, that this binding can be inhibited by poly-Ala tracts directly, and that poly-Ala expansions in patient proteins can impair UBA6-USE1 function in neurons. E6AP does appear to be an important downstream target in this pathway as well. I am in favor of publication in the EMBO Journal with just minor revisions.

Answer 2:

We appreciate the referee's enthusiasm about our work and his/her valuable suggestions to improve our paper, which we have now addressed both experimentally and by revising the text.

Comment 3:

Should mention in the Intro that UBA6 also activates the ubiquitin-like protein FAT10.

Answer 3:

We have now added this information and reference to the text, page 4: "*USE1 can be specifically loaded with ubiquitin or ubiquitin-like FAT10 by the E1 ubiquitin activating enzyme, UBA6, via a transthiolation reaction (Aichem, Pelzer et al., 2010, Jin et al., 2007, Lee, Sowa et al., 2011, Pelzer et al., 2007)*".

Comment 4:

For Fig. 3g, the better control would seem to be PHOX2B w/o the 13 Ala tract, not the empty vector.

Answer 4:

We appreciate this comment. We now added new data in **Fig 4 F, G** using the wild-type (WT) PHOX2B without the 13 Ala tract as a control for the mutant PHOX2B + 13 Ala to detect E6AP and also Arc stability in primary neurons. Results, page 10: "*To measure the effects of mutant PHOX2B on the stability of E6AP and Arc, we performed cycloheximide chase experiments in transduced neurons expressing WT or mutant PHOX2B (+13Ala). While E6AP levels remained stable in the mutant PHOX2B-expressing neurons, Arc was more rapidly degraded (Fig 4F, G)*". As we wrote to Referee 1, the initial levels of WT and mutant PHOX2B detected in the gel are different (mutant PHOX2B seems to be lower) possibly because of insoluble/aggregated fraction of the mutant protein, which could not be properly resolved. We have previously addressed this in a comment of the previous reviewers and showed supporting data for it in Fig 4C. In addition, we performed immunoprecipitation experiment of E6AP in cells expressing the WT PHOX2B or mutant PHOX2B (+13Ala) for polyubiquitination analysis. We found that the mutant PHOX2B (but not WT) decreases E6AP polyubiquitination. The new data was added to **Fig EV3G**. Overall, the new data support our model.

Comment 5:

There are typos here and there that will need to be corrected.

Answer 5:

We thank the referee for spotting typos, which were now corrected.

Dear Avraham,

Thank you for the submission of your revised manuscript to The EMBO Journal. We have now received the comments of both referees that were asked to re-evaluate your study (included below). As you will see, both referees acknowledge that all of their previous concerns have been successfully addressed, and they now recommend publication.

From the editorial side, there are a few things that we need from you before we can proceed with publication of your manuscript:

- All corresponding authors are required to supply their ORCID IDs; please provide the ORCID ID of Prof. Gad D. Vatine.
- Please enter all relevant funding information in our online manuscript handling system. It should match exactly the information provided in the Acknowledgements section of your manuscript.
- Please provide up to 5 keywords in a revised version of the manuscript (after the abstract).
- Please rename "Methods" to "Materials and Methods".
- Please note that only the names of the first 10 co-authors followed by "et al." should be listed for publications with more than 10 authors.
- I would like to clarify that the "Data availability" section should contain access information (including specific URLs) of only those publicly available datasets that were generated in this study. Previously published datasets that were only (re-)analyzed in this study should be cited in the respective paragraphs of the Materials and Methods section rather than in the Data availability statement. If your study does not include any new datasets requiring deposition in a public database, please add the statement "This study includes no data deposited in external repositories." under the heading "Data availability".
- We noticed that no source data have been provided for Fig. 3J. Please included them in your re-submission.
- Please note that the final dimensions of the synopsis image are 550 pixels (width) x 300-600 pixels (height can be variable within this range). When your synopsis image is resized to these dimensions, text becomes illegible; we would therefore like to ask you to increase the font size of the text to make sure it will be easily readable when the image is resized to these dimensions.
- Although the sample size ("n") is provided, the nature of replicates (e.g. biological or technical) should also be defined in the legends of figures 1c, e; 2b, d; 3g-h; EV1a, d; EV5g.
- Please define the sample size ("n") and the nature of replicates in the legends of 4e; 5d-e, g; EV5a-b.
- Please define the error bars in the legends of figures 4a, b, d, e; 5b; EV1d; EV2b.
- Please consider adding a "Data information" section at the end of each figure legend when there is repetitive information regarding data, statistics, representation (e.g. error bars, scale bars) etc. referring to more than one panels of the figure.
- As soon as these issues are resolved, I might contact you again to discuss with you a few suggestions for minor improvements in the title, abstract and synopsis text.

Please also note that as part of the EMBO publications' Transparent Editorial Process, The EMBO Journal publishes online a Peer Review File along with each accepted manuscript. This File will be published in conjunction with your paper and will include the referee reports, your point-by-point response and all pertinent correspondence relating to the manuscript. You can opt out of this by letting the editorial office know (contact@embojournal.org). If you do opt out, the Peer Review File link will point to the following statement: "No Peer Review File is available with this article, as the authors have chosen not to make the review process public in this case."

We look forward to seeing a final version of your manuscript as soon as possible. Please use this link to submit your revision:
<https://emboj.msubmit.net/cgi-bin/main.plex>

Best regards,

Ioannis

Referee #1:

I have no further concerns.

Referee #2:

My initial review was already in favor of publication and essentially only asked for an additional control experiment, which has now been done. The response to reviewers was satisfactory.

All editorial and formatting issues were resolved by the authors.

Dear Avraham,

I am pleased to inform you that your manuscript has been accepted for publication in the EMBO Journal.

Best regards,

Ioannis
